# Variance-reduced accelerated methods for decentralized stochastic double-regularized nonconvex strongly-concave minimax problems

**Gabriel Mancino-Ball** *gabriel.mancino.ball@gmail.com*
*Department of Mathematical Sciences*
*Rensselaer Polytechnic Institute*
*Troy, NY 12180, USA*

**Muhammad Khan** *khanm7@rpi.edu*
*Department of Mathematical Sciences*
*Rensselaer Polytechnic Institute*
*Troy, NY 12180, USA*

**Yangyang Xu**[*] *xuy21@rpi.edu*
*Department of Mathematical Sciences*
*Rensselaer Polytechnic Institute*
*Troy, NY 12180, USA*

**Reviewed on OpenReview:** *https://openreview.net/forum?id=t1Nj3VTNzQ*

## Abstract

In this paper, we consider the decentralized, stochastic nonconvex strongly-concave (NCSC) minimax problem with nonsmooth regularization terms on both primal and dual variables, wherein a network of $m$ computing agents collaborate via peer-to-peer communications. We consider when the coupling function is in expectation or finite-sum form and the double regularizers are convex functions, applied separately to the primal and dual variables. Our algorithmic framework introduces a Lagrangian multiplier to eliminate the consensus constraint on the dual variable. Coupling this with variance-reduction (VR) techniques, our proposed method, entitled `VRLM`, by a single neighbor communication per iteration, is able to achieve an $\mathcal{O}(\kappa^3 \varepsilon^{-3})$ sample complexity under the general stochastic setting, with either a large-batch or small-batch VR option, where $\kappa$ is the condition number of the problem and $\varepsilon$ is the desired solution accuracy. With a large-batch VR, we can additionally achieve $\mathcal{O}(\kappa^2 \varepsilon^{-2})$ communication complexity. Under the special finite-sum setting, our method with a large-batch VR can achieve an $\mathcal{O}(n + \sqrt{n}\kappa^2 \varepsilon^{-2})$ sample complexity and $\mathcal{O}(\kappa^2 \varepsilon^{-2})$ communication complexity, where $n$ is the number of components in the finite sum. All complexity results match the best-known results achieved by a few existing methods for solving special cases of the problem we consider. To the best of our knowledge, this is the first work which provides convergence guarantees for NCSC minimax problems with general convex nonsmooth regularizers applied to both the primal and dual variables in the decentralized stochastic setting. Numerical experiments are conducted on two machine learning problems. Our code is available at `https://github.com/RPI-OPT/VRLM`.

---

[*]Corresponding Author

# 1 Introduction

In this paper, we consider a multi-agent minimax-structured problem:

$$\min_{\mathbf{x}\in\mathbb{R}^{d_1}} \max_{\mathbf{y}\in\mathbb{R}^{d_2}} \frac{1}{m}\sum_{i=1}^{m} f_i(\mathbf{x},\mathbf{y}) + g(\mathbf{x}) - h(\mathbf{y}), \text{ with } f_i(\mathbf{x},\mathbf{y}) := \mathbb{E}_{\xi\sim\mathcal{D}_i}\left[\tilde{f}_i(\mathbf{x},\mathbf{y};\xi)\right], \tag{1}$$

which are solved by $m$ computing agents collaboratively over a decentralized communication network. Here, $f_i$ is an $L$-smooth, nonconvex ($\mu$-)strongly-concave (NCSC) function known only to agent $i$, and $g,h$ are closed, convex functions common to all agents, often serving as regularizers. We refer to (1) as a *double-regularized minimax* problem which has received a lot of research attention recently due to its application in many machine learning settings such as adversarial training (Goodfellow et al., 2014; Liu et al., 2020), distributionally robust optimization (Namkoong & Duchi, 2016; Xian et al., 2021), and reinforcement learning (Zhang et al., 2021). It has also been used to study fairness in machine learning (Nouiehed et al., 2019) and improving generalization error (Foret et al., 2021).

We are interested in the case where each $f_i$ is governed by a *privately* owned dataset $\mathcal{D}_i$ and its objective and (potentially stochastic) gradient information is only accessible by the $i$-th agent for all $i = 1,\ldots,m$. The datasets $\{\mathcal{D}_i\}_{i=1}^{m}$ may be finite or infinite, corresponding to offline or online scenarios, respectively. In the special case where $\mathcal{D}_i$ is a discrete uniform distribution, we recover the finite-sum problem setting, and without lose of generality, we can assume each $\mathcal{D}_i$ involves the same number of scenarios, i.e.,

$$f_i(\mathbf{x},\mathbf{y}) := \frac{1}{n}\sum_{j=1}^{n} \tilde{f}_i(\mathbf{x},\mathbf{y};\xi_{ij}). \tag{2}$$

The agents are connected over a communication network $\mathcal{G} = (\mathcal{V},\mathcal{E})$ where $\mathcal{V} = \{1,\ldots,m\}$ denotes the set of agents and $\mathcal{E} \subseteq \mathcal{V}\times\mathcal{V}$ denotes the set of feasible communication links between the agents. We assume the network $\mathcal{G}$ is connected. Such a problem scenario is close to a real-world setting where agents may not have access to a global communication protocol (either due to privacy restrictions (Verbraeken et al., 2020) or high communication overhead (Lian et al., 2017)), nor access to the entire local gradient (e.g., when data arrives in a stream (Xu & Xu, 2023)).

For the $m$ agents to collaboratively solve (1), each agent $i \in \mathcal{V}$ will maintain a local copy of the primal-dual variable $(\mathbf{x},\mathbf{y})$, denoted as $(\mathbf{x}_i,\mathbf{y}_i)$. With the introduction of local variables $\{(\mathbf{x}_i,\mathbf{y}_i)\}_{i\in\mathcal{V}}$, we can reformulate (1) equivalently into the following decentralized consensus minimax problem

$$\min_{\mathbf{x}_1,\ldots,\mathbf{x}_m} \max_{\mathbf{y}_1,\ldots,\mathbf{y}_m} \frac{1}{m}\sum_{i=1}^{m} \left(f_i(\mathbf{x}_i,\mathbf{y}_i) + g(\mathbf{x}_i) - h(\mathbf{y}_i)\right), \text{ s.t. } (\mathbf{x}_i,\mathbf{y}_i) = (\mathbf{x}_j,\mathbf{y}_j), \forall i,j \in \mathcal{V}. \tag{3}$$

To ensure that the consensus constraint $(\mathbf{x}_i,\mathbf{y}_i) = (\mathbf{x}_j,\mathbf{y}_j)$ is satisfied for all $i, j \in \mathcal{V}$, agents exchange their information with their 1-hop neighbors via the communication links in $\mathcal{E}$. Mathematically, this communication is represented as multiplication with a *mixing matrix*, $\mathbf{W} \in \mathbb{R}^{m\times m}$, which serves as an inexact averaging operation among the agents. More details on $\mathbf{W}$ are provided in Assumption 3.

We are primarily interested in establishing the efficiency of a method to solve (3) by measuring the number of (stochastic) gradient calls and neighbor communications required to acheive $\varepsilon$-stationarity based on the primal function $p(\mathbf{x}) := \max_{\mathbf{y}}\{\frac{1}{m}\sum_{i\in\mathcal{V}} f_i(\mathbf{x},\mathbf{y}) - h(\mathbf{y})\}$. We refer to these quantities as *sample* and *communication* complexities, respectively. **In this work, we propose the Variance Reduced Lagrangian Multiplier method, VRLM, which is a novel single-loop (i.e., no inner subroutine needed) framework that leverages state-of-the-art variance-reduction (VR) techniques to achieve low sample *and* communication complexities when solving (3). Our framework is based on a reformulation of (3) which allows agents to take an aggressive step-size in the primal variable x, matching that of centralized methods. We acheive this by exploiting the strong-concavity of $f_i(\mathbf{x},\cdot)$ to remove the consensus constraint on the y-variable with the introduction of a per-agent Lagrangian multiplier.** We summarize our contributions below, followed by a brief literature review of relevant works.

## 1.1 Contributions

Our contributions are three-fold: the design of new algorithms, the establishment of improved complexity results, and the demonstration of promising numerical performance.

- *First*, we leverage a reformulation of problem (3) outlined in Xu (2024) to present the `Variance Reduced Lagrangian Multiplier` based framework, `VRLM`. `VRLM` is a single-loop (i.e., no inner subroutine needed) framework which allows for different VR gradient estimators to be "swapped" in for the local gradient estimators. While we leverage the probelm reformulation presented in the decentralized GDMax method from Xu (2024), we note that GDMax equipped with an optimal stochastic subroutine, will have worse complexity than our results, highlighting the contribution of our approach. Additionally, our framework allows for regularizers on both the primal and dual variables, enabling it to be applied to a broader class of problems than other methods which solve (3); see the related works in Section 1.2 for more details.

- *Second*, we provide rigorous convergence analysis of two VR options. One VR option uses large-batch sampling, while the other needs only $\mathcal{O}(1)$ samples for each update per agent. We prove that both versions of our method can achieve a sample complexity of $\mathcal{O}\left(\kappa^3\varepsilon^{-3}\right)$ in the general stochastic setting where $\kappa$ is the problem's condition number. The small-batch version attains a communication complexity of $\mathcal{O}\left(\kappa^3\varepsilon^{-3}\right)$, while the large-batch one only needs $\mathcal{O}\left(\kappa^2\varepsilon^{-2}\right)$ communication rounds. In addition, for the finite-sum structured problem (2), our method with the large-batch VR option can achieve a sample complexity of $\mathcal{O}(n + \sqrt{n}\kappa^2\varepsilon^{-2})$ and communication complexity of $\mathcal{O}(\kappa^2\varepsilon^{-2})$ per agent. These complexity results are optimal in terms of the dependence on $\varepsilon$ (Arjevani et al., 2023; Lu & De Sa, 2021) and match the best-known results of decentralized methods for solving special cases of (1).

- *Third*, we verify the performance of our proposed method on two machine learning problems in a real decentralized computing environment using NVIDIA Telsa V100 GPUs. Additionally, we make our code open source at `https://github.com/RPI-OPT/VRLM`.

## 1.2 Related work

A few decentralized methods have been proposed for solving minimax problems. However, most of them focus on deterministic or finite-sum structured problems. We provide additional related work in Appendix A.

The recent D-GDMax method (Xu, 2024) is very closely related to our proposed method, as both rely on a reformulation of (3) by the introduction of a Lagrangian multiplier to facilitate convergence. A key difference is that D-GDMax requires exact gradients. Thus it is not applicable to the general stochastic problem setting that we consider. While D-GDMax could be equipped with an optimal stochastic subroutine, it will have higher complexity than our results. In the special finite-sum case, D-GDMax' complexity will be significantly higher than ours if $n$ is big and the desired accuracy $\varepsilon$ is small. GT-DA (Tsaknakis et al., 2020) is also a deterministic gradient method. It solves a variant of (3) by only enforcing consensus on either the $\{\mathbf{x}_i\}_{i\in\mathcal{V}}$ or $\{\mathbf{y}_i\}_{i\in\mathcal{V}}$ variables, but not both simultaneously.

The PRECISION method in Liu et al. (2023) is designed for solving (3) under the finite-sum setting, and in addition, it assumes $h(\cdot) = \mathbb{I}_{\mathcal{Y}}(\cdot)$ for some convex set $\mathcal{Y}$. Similar to one VR option of our method, it utilizes the SPIDER-type (Fang et al., 2018) variance reduction. In order to produce an $\varepsilon$-stationary point, it needs $\mathcal{O}(n + \sqrt{n}\varepsilon^{-2})$ sample gradients and $\mathcal{O}(\varepsilon^{-2})$ communications rounds per agent without giving an explicit dependence on the problem's condition number $\kappa$; the dependence on $\varepsilon$ and $n$ is the best-known for the finite-sum setting. A preceding method of PRECISION, called GT-SRVR in Zhang et al. (2021), considers a more special case of (3) under the finite-sum setting. It assumes $g \equiv 0$ and $h \equiv 0$ and achieves the same-order complexity results by the SPIDER-type variance reduction. DSGDA (Gao, 2022) solves the same-structured problem as GT-SRVR and also enjoys the same-order complexity results. Different from GT-SRVR, DSGDA uses a SAGA-type acceleration technique (Defazio et al., 2014). It does not need to take large-batch samples for each update but requires a large memory to maintain $n$ component gradients.

Like our method, DREAM (Chen et al., 2024) can be applied to both the stochastic and finite-sum settings of (3). It employs a loopless version of the SPIDER-type VR method. However, it assumes $g \equiv 0$ and $h = \mathbb{I}_\mathcal{Y}$. In addition, it performs multiple communications per update, and its convergence results rely on the multi-communication trick. This is fundamentally different from our method. With a multi-communication trick, our results can have a lower-order dependence on the spectral of the communication network; see Remark 4.1. However, we can have guaranteed convergence with a single communication per update, while the analysis of DREAM requires the multi-communication trick to prove convergence. The requirement of multiple communications can be too restrictive and even impractical in a real computing setting, as it necessitates more coordination between agents (Lian et al., 2017). All the aforementioned methods require either large-batch samplings or a large number of maintained component gradients. In contrast, DM-HSGD (Xian et al., 2021), by the STORM-type (Cutkosky & Orabona, 2019) VR technique, only needs $\mathcal{O}(1)$ samples per update per agent, except for a possible large-batch sampling at the initial step. However, there is one critical error with its convergence analysis. It turns out that Eqn. (28) in Xian et al. (2021) does not hold with the given choice of $\theta$. In fact, $\theta$ must be $\Theta(1/L)$ instead of the given $\Theta(1/\mu)$, where $L$ is the smoothness constant of $\{f_i\}$ and $\mu$ the strong-concavity constant of $\{f_i(\mathbf{x}, \cdot)\}$. With the correct $\theta$, it is unclear if its final claimed convergence results will hold[1].

## 1.3 Notation

We denote $[m]$ as the set $\{1, \ldots, m\}$. We let $\mathbf{z} := [\mathbf{x}; \mathbf{y}] \in \mathbb{R}^{d_1 + d_2}$ and $\mathcal{Z} = \text{dom}(g) \times \text{dom}(h)$ be the joint variable and domain. For each $i \in [m]$, let $\mathbf{z}_i := [\mathbf{x}_i; \mathbf{y}_i]$ be a local copy of $\mathbf{z}$ on agent $i$. Then we denote

$$\mathbf{X} = \begin{bmatrix} \mathbf{x}_1, \ldots, \mathbf{x}_m \end{bmatrix}^\top, \ \mathbf{Y} = \begin{bmatrix} \mathbf{y}_1, \ldots, \mathbf{y}_m \end{bmatrix}^\top, \ \mathbf{Z} = \begin{bmatrix} \mathbf{z}_1, \ldots, \mathbf{z}_m \end{bmatrix}^\top,$$
$$\bar{\mathbf{x}} = \tfrac{1}{m} \sum_{i=1}^m \mathbf{x}_i, \ \mathbf{X}_\perp = \mathbf{X} - \mathbf{1}\bar{\mathbf{x}}^\top, \tag{4a}$$
$$\nabla_\mathbf{x} F(\mathbf{Z}) = \begin{bmatrix} \nabla_\mathbf{x} f_1(\mathbf{z}_1), \ldots, \nabla_\mathbf{x} f_m(\mathbf{z}_m) \end{bmatrix}^\top,$$
$$\nabla_\mathbf{y} F(\mathbf{Z}) = \begin{bmatrix} \nabla_\mathbf{y} f_1(\mathbf{z}_1), \ldots, \nabla_\mathbf{y} f_m(\mathbf{z}_m) \end{bmatrix}^\top. \tag{4b}$$

We use the superscript $^{(t)}$ for the $t$-th iteration. For a set $\mathcal{X} \subseteq \mathbb{R}^d$, we denote its $0$-$\infty$ indicator function by $\mathbb{I}_\mathcal{X}(\cdot)$, i.e., $\mathbb{I}_\mathcal{X}(\mathbf{x}) = 0$ if $\mathbf{x} \in \mathcal{X}$ and $+\infty$ otherwise. For a closed convex function $r$, we define its proximal mapping as $\mathbf{prox}_r(\mathbf{x}) := \arg\min_\mathbf{y}\{r(\mathbf{y}) + \tfrac{1}{2} \|\mathbf{y} - \mathbf{x}\|^2\}$. Finally, given a set of random samples $\mathcal{B}$, we denote mini-batch gradient estimators as

$$G_i(\mathcal{B}) := \tfrac{1}{|\mathcal{B}|} \sum_{\xi \in \mathcal{B}} \tilde{\nabla} f_i(\mathbf{x}_i, \mathbf{y}_i; \xi), \quad G_i^{(t)}(\mathcal{B}) := \tfrac{1}{|\mathcal{B}|} \sum_{\xi \in \mathcal{B}} \tilde{\nabla} f_i(\mathbf{x}_i^{(t)}, \mathbf{y}_i^{(t)}; \xi). \tag{5}$$

## 1.4 Outline

The rest of this paper proceeds as follows. In Section 2 we state the reformulation of problem (3) and outline our key assumptions. In Section 3 we introduce our proposed method, VRLM, for solving the reforumlated problem. We provide convergence results in Section 4 and numerical experiments in Section 5. In Section 6 we make concluding remarks.

## 2 Assumptions and preliminaries

As stated in the introduction, our proposed method relies on a reformulation of (3) presented in Xu (2024) which removes the consensus constraint on the $\mathbf{y}$-variables by introducing a per-agent Langrangian multiplier, $\boldsymbol{\lambda}_i$ for all $i = 1, \ldots, m$. We define the following modified objective function

$$\Phi(\mathbf{X}, \boldsymbol{\Lambda}, \mathbf{Y}) := \tfrac{1}{m} \sum_{i=1}^m \left( f_i(\mathbf{x}_i, \mathbf{y}_i) - h(\mathbf{y}_i) \right) - \tfrac{L}{2\sqrt{m}} \langle \boldsymbol{\Lambda}, (\mathbf{W} - \mathbf{I})\mathbf{Y} \rangle, \tag{6}$$

---

[1]With $\theta = \Theta(1/L)$, the coefficient for $\mathbb{E} \|\bar{u}^t\|$ becomes positive but it is required to be negative in the analysis for the DM-HSGD method.

where the $i$-th row of $\mathbf{\Lambda}$ is $\boldsymbol{\lambda}_i^\top$. When $\mathrm{dom}(h)$ has nonempty relative interior, the problem (3) can be reformulated equivalently into

$$\min_{\mathbf{X},\mathbf{\Lambda}} \max_{\mathbf{Y}} \ \Phi(\mathbf{X},\mathbf{\Lambda},\mathbf{Y}) + \tfrac{1}{m}\sum_{i=1}^m g(\mathbf{x}_i), \ \text{s.t.} \ \mathbf{x}_i = \mathbf{x}_j, \forall\, i,j \in [m]. \tag{7}$$

In Section 3, we present a novel single loop framework for solving (7). First, we state our key assumptions and define a solution criteria for methods that solve (7). We define the following functions that will be used throughout our analysis

$$P(\mathbf{x},\mathbf{\Lambda}) := \max_{\mathbf{Y}} \Phi(\mathbf{1}\mathbf{x}^\top, \mathbf{\Lambda}, \mathbf{Y}), \quad Q(\mathbf{X},\mathbf{\Lambda}) := \max_{\mathbf{Y}} \Phi(\mathbf{X},\mathbf{\Lambda},\mathbf{Y}),$$
$$S_\Phi(\mathbf{X},\mathbf{\Lambda}) := \arg\max_{\mathbf{Y}} \Phi(\mathbf{X},\mathbf{\Lambda},\mathbf{Y}), \tag{8}$$

$$f(\mathbf{x},\mathbf{y}) := \tfrac{1}{m}\sum_{i=1}^m f_i(\mathbf{x},\mathbf{y}), \quad p(\mathbf{x}) := \max_{\mathbf{y}} f(\mathbf{x},\mathbf{y}) - h(\mathbf{y}), \quad \phi(\mathbf{x},\mathbf{\Lambda}) := P(\mathbf{x},\mathbf{\Lambda}) + g(\mathbf{x}). \tag{9}$$

We make the following assumptions on the objective function and communication network, which are standard in the minimax and decentralized optimization literature (Zhang et al., 2021; Shi et al., 2015).

**Assumption 1** *For each $i \in [m]$, $f_i$ is $L$-smooth. In addition, in the stochastic case, $\tilde{\nabla} f_i(\mathbf{x},\mathbf{y};\xi)$ satisfies* (i) $\mathbb{E}_\xi[\tilde{\nabla} f_i(\mathbf{x},\mathbf{y};\xi)] = \nabla f_i(\mathbf{x},\mathbf{y})$, (ii) *there exists $\sigma \geq 0$ such that $\mathbb{E}_\xi \left\| \tilde{\nabla} f_i(\mathbf{x},\mathbf{y};\xi) - \nabla f_i(\mathbf{x},\mathbf{y}) \right\|_2^2 \leq \sigma^2$, and* (iii) *for any $(\mathbf{x},\mathbf{y}),(\mathbf{x}',\mathbf{y}') \in \mathbb{R}^{d_1+d_2}$, it holds $\mathbb{E}_\xi \left\| \tilde{\nabla} f_i(\mathbf{x},\mathbf{y};\xi) - \tilde{\nabla} f_i(\mathbf{x}',\mathbf{y}';\xi) \right\|_2^2 \leq L^2 \left( \|\mathbf{x}-\mathbf{x}'\|_2^2 + \|\mathbf{y}-\mathbf{y}'\|_2^2 \right)$.*

**Assumption 2** *For each $i \in [m]$, $f_i(\mathbf{x},\cdot)$ is $\mu$-strongly concave with $\mu > 0$; $g$ and $h$ are closed, convex functions and $\mathrm{dom}(h)$ has a nonempty relative interior; there exists $\phi^* \in \mathbb{R}$ such that $\phi(\mathbf{x},\mathbf{\Lambda}) \geq \phi^*$ for all $\mathbf{x}, \mathbf{\Lambda}$ where $\phi$ is defined in* (9).

Assumption 1 (iii) is called the *mean-squared smoothness property* and is necessary for a stochastic gradient-based algorithm to achieve the optimal order $\mathcal{O}(\varepsilon^{-3})$ of complexity to produce an $\varepsilon$-stationary point, as defined below in Definition 2.1 (Arjevani et al., 2023). Additionally, we highlight Assumption 2 necessitates the strong concavity of the individual $f_i(\mathbf{x},\cdot)$ rather than a weaker assumption only requiring strong concavity of the global function $f(\mathbf{x},\cdot)$. Technically, we require this stronger assumption to ensure $P$ and $Q$ in (8) are smooth as $\Phi$ depends on all $\mathbf{y}_i$; it is an open question whether or not this assumption can be relaxed and the same convergence rates in Corollaries 4.1 and 4.2 still hold.

We denote the problem's condition number by

$$\kappa := \frac{L}{\mu}. \tag{10}$$

As stated in the introduction, the communication network $\mathcal{G}$ is assumed to be connected; we enforce this by making the following assumption on the mixing matrix $\mathbf{W}$.

**Assumption 3** *The mixing matrix $\mathbf{W} \in \mathbb{R}^{m \times m}$ satisfies the conditions:* (i) $\mathrm{Null}(\mathbf{W}-\mathbf{I}) = \mathrm{Span}\{\mathbf{1}\}$ *and* $\mathbf{W}^\top \mathbf{1} = \mathbf{1}$; (ii) $\rho := \|\mathbf{W} - \tfrac{1}{m}\mathbf{1}\mathbf{1}^\top\|_2 < 1$; (iii) $\|\mathbf{W}-\mathbf{I}\|_2 \leq 2$.

To quantify the sample and communication complexities, we are interested in finding an $\varepsilon$-stationary point of (7) for a given tolerance $\varepsilon > 0$, defined below.

**Definition 2.1** *For any $\varepsilon > 0$, a point $(\mathbf{X},\mathbf{\Lambda})$ is called an $\varepsilon$-stationary point in expectation of* (7) *if there is some $\eta_\mathbf{x} > 0$ such that for the primal function $P(\mathbf{x},\mathbf{\Lambda}) := \max_{\mathbf{Y}} \Phi(\mathbf{1}\mathbf{x}^\top, \mathbf{\Lambda}, \mathbf{Y})$, we have*

$$\mathbb{E} \left\| \frac{1}{\eta_\mathbf{x}} \left( \bar{\mathbf{x}} - \mathbf{prox}_{\eta_\mathbf{x} g} \left( \bar{\mathbf{x}} - \eta_\mathbf{x} \nabla_\mathbf{x} P(\bar{\mathbf{x}},\mathbf{\Lambda}) \right) \right) \right\|_2^2 + \frac{L^2}{m} \mathbb{E} \|\mathbf{X}_\perp\|_F^2 \leq \varepsilon^2 \ \text{and} \ \mathbb{E} \|\nabla_\mathbf{\Lambda} P(\bar{\mathbf{x}},\mathbf{\Lambda})\|_F^2 \leq \varepsilon^2. \tag{11}$$

Based on Definition 2.1, we define the **sample complexity** and **communication complexity** respectively as the **number of sample gradients** and the **number of communication rounds required to produce an $\varepsilon$-stationary point in expectation**. We make the following remark regarding the relationship between $\varepsilon$-stationarity for (7) and (3).

**Remark 2.1** *Notice that $P$ is the objective function of the primal problem of* (7). *The stationarity measure in Definition 2.1 is sometimes called optimization stationarity (Zheng et al., 2022). Another related notion is the so-called game stationarity (Li et al., 2025), which also involves the dual variable* $\mathbf{y}$. *In addition, Xu (2024) shows that if* $(\mathbf{X}, \mathbf{\Lambda})$ *is an $\varepsilon$-stationary point of* (7), *then* $\mathbf{X}$ *is an $\mathcal{O}(\varepsilon)$-stationary point of the original decentralized formulation* (3) *when $h$ is smooth. This claim can actually be extended to the case where the function $P(\mathbf{x}, \cdot)$ defined in* (8) *satisfies a quadratic-growth condition (Drusvyatskiy & Lewis, 2018) for all* $\mathbf{x} \in \mathrm{dom}(g)$. *Hence, we adopt the notion in Definition 2.1. Note that smoothness of $h$ is not a necessary condition for Definition 2.1 to apply to* (7), *but rather a sufficient condition to easily relate the stationarity of problems* (7) *and* (3).

## 3 Proposed method

It is not immediately obvious why stochastic objective functions would benefit from the problem reformulation of (7). As stated in Xu (2024), problem (7) allows agents to take an aggressive step-size in the primal $\mathbf{x}$-variables through local maximization of the $\mathbf{y}$-variables. Even in the less general finite-sum setting (2), such an approach incurs an additional $\sqrt{\kappa}$ gradient computations in the $\mathbf{y}$-variables compared to the number of required $\mathbf{x}$-gradient computations (see Table 1 in Xu (2024) for more details). In this work, we address two lingering questions from Xu (2024). First, we remove the need for agents to perform local $\mathbf{y}$-variable maximization, which reduces the overall algorithm complexity, while still maintaining the benefit of aggressive $\mathbf{x}$-variable step-sizes; in particular, we only require agents to compute a single (potentially stochastic) gradient during each algorithm iteration, which can significantly reduce algorithm run-time in practice. Second, our algorithm and corresponding analysis is applicable to more general expectation structured problems. We now state our proposed framework for solving (7).

We propose the Variance Reduced Lagrangian Multiplier method, VRLM, for decentralized double-regularized minimax problems. VRLM makes use of four key algorithmic components to solve (7). First, we incorporate **variance-reduction (VR)** techniques to facilitate a better estimate of the true local gradients; namely we utilize either the SPIDER (Fang et al., 2018; Nguyen et al., 2017) or STORM-type (Cutkosky & Orabona, 2019; Xu & Xu, 2023) VR technique. Second, we leverage **gradient tracking** in the $\mathbf{x}$-variable to make the least restrictive assumptions on the data distribution among the agents (Tang et al., 2018b), namely, heterogeneous data is allowed. Third, we communicate the $\mathbf{y}$-variables through **Lagrangian multiplier** updates, relaxing the need for exact consensus on these variables and allowing for large $\mathbf{x}$-variable step-sizes. Finally, VRLM is a **single loop (i.e., no subroutine needed), single communication per update** algorithm which requires less computation than double loop algorithms (Luo et al., 2020) and less communication than multi-communication algorithms (Chen et al., 2024). Compared to other gradient tracking and VR-based algorithms for decentralized minimax problems (Xian et al., 2021; Chen et al., 2024), we incur the cost of storing one additional variable, $\mathbf{\lambda}_i$, on each agent $i$, but we do not incur any additional variable communications as all algorithms in this category communicate four variables per iteration.

The SPIDER and STORM-type VR gradient estimators represent large-batch and small-batch gradient estimators, respectively. Fundamentally, VRLM is not limited to just these gradient estimators. Alternatives such as ZeroSARAH (Li et al., 2021b) or PAGE (Li et al., 2021a) could also be utilized as gradient estimators. We limit formal analysis of VRLM to the SPIDER and STORM-type gradient estimators to demonstrate the flexibility of our approach and leave analytical extensions to other gradient estimators to interested readers. Using equation (5), we denote the gradient estimator on the $i$-th agent as $\mathbf{d}_i^{(t)} \triangleq [\mathbf{d}_{\mathbf{x},i}^{(t)}; \mathbf{d}_{\mathbf{y},i}^{(t)}]$; for the SPIDER-type VR gradient estimator we have

$$\mathbf{d}_i^{(t)} = \begin{cases} G_i^{(t)}(\tilde{\mathcal{B}}_i^{(t)}), & \text{if } \mathrm{mod}(t,q) = 0, \text{ with } |\tilde{\mathcal{B}}_i^{(t)}| = \mathcal{S}_1, \\ G_i^{(t)}(\mathcal{B}_i^{(t)}) - G_i^{(t-1)}(\mathcal{B}_i^{(t)}) + \mathbf{d}_i^{(t-1)}, & \text{otherwise, with } |\mathcal{B}_i^{(t)}| = \mathcal{S}_2, \end{cases} \tag{12}$$

while for the STORM-type VR gradient estimator we have

$$\mathbf{d}_i^{(t)} = G_i^{(t)}(\mathcal{B}_i^{(t)}) + (1 - \beta)\left(\mathbf{d}_i^{(t-1)} - G_i^{(t-1)}(\mathcal{B}_i^{(t)})\right), \text{ with } |\mathcal{B}_i^{(t)}| = \mathcal{S}_t, \text{ for } t \geq 1. \tag{13}$$

The SPIDER-type VR technique requires large-batch sampling, meaning that the number of samples to compute a local gradient estimator at each iteration depends on a desired solution accuracy $\varepsilon$. As shown in the next section, the large batches can lead to an improved communication complexity result, but may lead to poor generalization on some machine learning tasks (Keskar et al., 2017). Furthermore, in scenarios where the data arrives in a stream, it may be impractical to wait for enough samples to compute a large-batch gradient estimator. Hence, we also analyze the STORM VR technique. By the STORM technique, agents only need $\mathcal{O}(1)$ samples to compute a stochastic gradient estimator, except for a possible large-batch sampling at the initial step. We find in practice (see Section 5) that the STORM estimator outperforms the SPIDER estimator on more complex tasks. Nevertheless, we will provide convergence analysis for both methods. Having presented (12) and (13), we give the pseudocode of VRLM in Algorithm 1.

---

**Algorithm 1:** Variance Reduced Lagrangian Multiplier (VRLM) method

**Input:** $\mathbf{x}_i^{(0)} = \mathbf{x}^{(0)} \in \mathrm{dom}(g)$, $\mathbf{y}_i^{(0)} = \mathbf{y}^{(0)} \in \mathrm{dom}(h)$, $\boldsymbol{\lambda}_i^{(0)} = \mathbf{0}$ for all $i \in [m]$; step-sizes $\eta_{\mathbf{x}}, \eta_{\mathbf{y}}, \eta_{\boldsymbol{\Lambda}} > 0$;
  VR-tag $\in \{\text{SPIDER, STORM}\}$

1 **Initial step:** set $\mathbf{v}_i^{(0)} = \mathbf{d}_i^{(0)} = G_i^{(0)}(\mathcal{B}_i^{(0)})$ where $|\mathcal{B}_i^{(0)}| = \mathcal{S}_0$ for all $i \in [m]$
2 **for** $t = 1, \dots, T$ **do**
3    **for agents** $i \in [m]$ **in parallel do**
4       **if** VR-tag $==$ SPIDER **then**
5          Update $\mathbf{d}_i^{(t)}$ by (12).
6       **if** VR-tag $==$ STORM **then**
7          Update $\mathbf{d}_i^{(t)}$ by (13).
8       Update the gradient tracking variables by $\mathbf{v}_{\mathbf{x},i}^{(t)} = \sum_{j=1}^m w_{ij} \left( \mathbf{v}_{\mathbf{x},j}^{(t-1)} + \mathbf{d}_{\mathbf{x},j}^{(t)} - \mathbf{d}_{\mathbf{x},j}^{(t-1)} \right)$ and
      $\mathbf{v}_{\mathbf{y},i}^{(t)} = \mathbf{d}_{\mathbf{y},i}^{(t)} - \frac{L\sqrt{m}}{2} \left( \sum_{j=1}^m w_{ji} \boldsymbol{\lambda}_j^{(t)} - \boldsymbol{\lambda}_i^{(t)} \right)$.
9       Update the Lagrangian multiplier by $\boldsymbol{\lambda}_i^{(t+1)} = \boldsymbol{\lambda}_i^{(t)} + \frac{L\eta_{\boldsymbol{\Lambda}}}{2\sqrt{m}} \left( \sum_{j=1}^m w_{ij} \mathbf{y}_j^{(t)} - \mathbf{y}_i^{(t)} \right)$.
10       Let $\mathbf{x}_i^{(t+1)} = \mathbf{prox}_{\eta_{\mathbf{x}} g} \left( \sum_{j=1}^m w_{ij} \mathbf{x}_j^{(t)} - \eta_{\mathbf{x}} \mathbf{v}_{\mathbf{x},i}^{(t)} \right)$ and $\mathbf{y}_i^{(t+1)} = \mathbf{prox}_{\eta_{\mathbf{y}} h} \left( \mathbf{y}_i^{(t)} + \eta_{\mathbf{y}} \mathbf{v}_{\mathbf{y},i}^{(t)} \right)$.

---

Algorithm 1 provides the VRLM updates from an agent point of view. To facilitate ease of expression and readability, we form $\mathbf{D}_{\mathbf{x}}, \mathbf{D}_{\mathbf{y}}$ and re-write the last three lines of Algorithm 1 in the following matrix form

$$\mathbf{D}_{\mathbf{x}}^{(t)} = [\mathbf{d}_{\mathbf{x},1}, \dots, \mathbf{d}_{\mathbf{x},m}]^\top, \quad \mathbf{D}_{\mathbf{y}}^{(t)} = [\mathbf{d}_{\mathbf{y},1}, \dots, \mathbf{d}_{\mathbf{y},m}]^\top, \tag{14}$$

$$\mathbf{V}_{\mathbf{x}}^{(t)} = \mathbf{W} \left( \mathbf{V}_{\mathbf{x}}^{(t-1)} + \mathbf{D}_{\mathbf{x}}^{(t)} - \mathbf{D}_{\mathbf{x}}^{(t-1)} \right), \quad \mathbf{V}_{\mathbf{y}}^{(t)} = \mathbf{D}_{\mathbf{y}}^{(t)} - \frac{L\sqrt{m}}{2} (\mathbf{W} - \mathbf{I})^\top \boldsymbol{\Lambda}^{(t)}, \tag{15}$$

$$\boldsymbol{\Lambda}^{(t+1)} = \boldsymbol{\Lambda}^{(t)} + \frac{L\eta_{\boldsymbol{\Lambda}}}{2\sqrt{m}} (\mathbf{W} - \mathbf{I}) \mathbf{Y}^{(t)}, \tag{16}$$

$$\mathbf{X}^{(t+1)} = \mathbf{prox}_{\eta_{\mathbf{x}} g} \left( \widetilde{\mathbf{X}}^{(t)} - \eta_{\mathbf{x}} \mathbf{V}_{\mathbf{x}}^{(t)} \right), \quad \widetilde{\mathbf{X}}^{(t)} = \mathbf{W} \mathbf{X}^{(t)}, \quad \mathbf{Y}^{(t+1)} = \mathbf{prox}_{\eta_{\mathbf{y}} h} \left( \mathbf{Y}^{(t)} + \eta_{\mathbf{y}} \mathbf{V}_{\mathbf{y}}^{(t)} \right), \tag{17}$$

where the **prox** operator acts row-wisely on the input.

## 4 Convergence results

In this section, we give the convergence analysis of Algorithm 1 for both VR options. The primary challenges with providing convergence guarantees are caused by the non-linearity of the proximal mapping associated with the non-smooth terms $g$ and $h$, the nonconvexity of $\{f_i(\cdot, \mathbf{y})\}$, and the stochasticity of gradient information. Our analysis carefully addresses each of these challenges. It adheres to the following logical flow. First, we provide a one-iteration progress inequality about $\phi$ from (9) without specifying the VR gradient estimator. Second, we present bounds on the consensus errors and local gradient estimators using Assumptions 1 and 3 coupled with the algorithm updates (15), (16), and (17). Finally, for the given VR estimators we construct relevant bounds which can used to establish a relationship between the one-iteration progress inequality and the stationarity violation in Definition 2.1. Our analysis builds

this relationship through defining new terms $\Gamma_t(\mathbf{Y}^{(t)}) :=:= \frac{1}{m} \sum_{i=1}^m f_i(\mathbf{x}_i^{(t)}, \mathbf{y}_i) - \frac{L}{2\sqrt{m}} \langle (\mathbf{W} - \mathbf{I}) \mathbf{Y}, \mathbf{\Lambda}^{(t)} \rangle$ and $\hat{\delta}_t := Q(\mathbf{X}^{(t)}, \mathbf{\Lambda}^{(t)}) - \left( \Gamma_t(\mathbf{Y}^{(t)}) - \frac{1}{m} \sum_{i=1}^m h(\mathbf{y}_i^{(t)}) \right)$ in (29) and (43) to facilitate the construction of novel Lyapunov functions which we use to show convergence. We defer details of the analysis to Appendix C and present our main results and remarks below.

## 4.1 SPIDER variance reduction

**Theorem 4.1** *Under Assumptions 1,2, and 3, let $\{(\mathbf{X}^{(t)}, \mathbf{\Lambda}^{(t)}, \mathbf{Y}^{(t)})\}_{t \geq 0}$ be generated from Algorithm 1 with VR-tag = SPIDER and $q = \mathcal{S}_2$, $\eta_\mathbf{y} = \frac{1}{4L}$, and*

$$\eta_\mathbf{x} = \min \left\{ \frac{(1-\rho)^2}{180 L_P}, \ \frac{1}{20(L+1)(12\kappa^2 + 2\kappa + 5)} \right\},$$

$$\eta_\mathbf{\Lambda} = \min \left\{ \frac{5L_P(1-\rho)^2}{24L^2(12\kappa^2 + \kappa + 1)}, \ \frac{1}{2L_P + 128L\kappa^2 + \frac{(L+1)(20\kappa^2 + \kappa + 1)}{2} + \frac{4L^2(12\kappa^2 + \kappa + 1)}{30 L_P}} \right\}.$$

*For any integer $T \geq 1$, select $\tau$ uniformly at random from $\{0, 1, \ldots, T-1\}$. Then*

$$
\mathbb{E} \left\| \frac{1}{\eta_\mathbf{x}} \left( \bar{\mathbf{x}}^{(\tau)} - \mathbf{prox}_{\eta_\mathbf{x} g} \left( \bar{\mathbf{x}}^{(\tau)} - \eta_\mathbf{x} \nabla_\mathbf{x} P(\bar{\mathbf{x}}^{(\tau)}, \mathbf{\Lambda}^{(\tau)}) \right) \right) \right\|_2^2 + \frac{L^2}{m} \mathbb{E} \left\| \mathbf{X}_\perp^{(\tau)} \right\|_F^2
$$
$$
\leq \frac{L^2(60\kappa + 3)}{mT} \left\| \widetilde{\mathbf{Y}}^{(0)} - \mathbf{Y}^{(0)} \right\|_F^2 + (80\kappa^2 + 10)\Upsilon + \frac{(80\kappa^2 + 10)L\hat{\delta}_0}{T}
$$
$$
+ \left( \frac{C_0}{T} + \left( \frac{1}{30 L_P} + \frac{1}{L_P(1-\rho)^2} + \frac{L+1}{8L^2} \right) \Upsilon \right) \cdot \left[ 10L^2 \left( 150\eta_\mathbf{x} \kappa^2 (20\kappa^2 + \kappa + 4) \right. \right. \tag{18}
$$
$$
\left. \left. + 16\eta_\mathbf{\Lambda} \kappa^2 (20\kappa^2 + \kappa + 3) \right) + \frac{20}{\eta_\mathbf{x}} + \left( 3L^2 \eta_\mathbf{x}(3 + 5\kappa^2) + \frac{7}{\eta_\mathbf{x}} \right) + 300 L_P \right]
$$

*and*

$$
\mathbb{E} \left\| \nabla_\mathbf{\Lambda} P(\bar{\mathbf{x}}^{(\tau)}, \mathbf{\Lambda}^{(\tau)}) \right\|_F^2 \leq \frac{24\kappa L^2}{mT} \left\| \widetilde{\mathbf{Y}}^{(0)} - \mathbf{Y}^{(0)} \right\|_F^2 + 32\kappa^2 \Upsilon + \frac{32 L \kappa^2 \hat{\delta}_0}{T}
$$
$$
+ \left( \frac{C_0}{T} + \left( \frac{1}{30 L_P} + \frac{1}{L_P(1-\rho)^2} + \frac{L+1}{8L^2} \right) \Upsilon \right) \cdot \left[ 4L^2 \left( 150\eta_\mathbf{x} \kappa^2 (20\kappa^2 + \kappa + 2) \right. \right. \tag{19}
$$
$$
\left. \left. + 16\eta_\mathbf{\Lambda} \kappa^2 (20\kappa^2 + \kappa + 1) \right) + \frac{4}{\eta_\mathbf{\Lambda}} + 6L^2 \kappa^2 \eta_\mathbf{x} \right],
$$

*where $\Upsilon = \frac{\sigma^2}{\mathcal{S}_1}$ for the case of a general distribution and $\Upsilon = 0$ for the finite-sum case in (2), $L_P = L\sqrt{4\kappa^2 + 1}$, and $C_0$ is a constant dependent on the initial point, defined in Theorem C.1.*

By Theorem 4.1, we give the total sample and communication complexity result of Algorithm 1 as follows with VR-tag == SPIDER. The proof is given in Appendix C.4.

**Corollary 4.1** *Let $\varepsilon > 0$ be given and assume $L \geq 1$. Under the assumptions in Theorem 4.1, suppose*

$$\left\| \mathbf{V}_{\perp, \mathbf{x}}^{(0)} \right\|_F^2 = \mathcal{O}\left( m L_P(1-\rho) \right), \quad \left\| \widetilde{\mathbf{Y}}^{(0)} - \mathbf{Y}^{(0)} \right\|_F^2 = \mathcal{O}\left( \min\left\{ \frac{m\kappa}{L}, \ \frac{m L_P \kappa (1-\rho)^2}{L^2} \right\} \right).$$

*Then Algorithm 1 with VR-tag = SPIDER, $\mathcal{S}_0 = \mathcal{S}_1 = \Theta\left( \frac{\sigma^2 \cdot \max\{\kappa^2, (1-\rho)^{-4}\}}{\varepsilon^2} \right)$ for the general stochastic case and $\mathcal{S}_0 = \mathcal{S}_1 = n$ for the special finite-sum case, and $\mathcal{S}_2 = q = \lceil \sqrt{\mathcal{S}_1} \rceil$ can find an $\varepsilon$-stationary point in expectation of (7) by $T_s$ stochastic gradients and $T_c$ local neighbor communications, where*

$$T_c = \Theta\left( \frac{L\kappa^2}{\varepsilon^2 \cdot \min\{1, \kappa(1-\rho)^2\}} \left( \phi(\bar{\mathbf{x}}^{(0)}, \mathbf{\Lambda}^{(0)}) - \phi^* + \frac{\hat{\delta}_0}{\min\{1, \kappa(1-\rho)^2\}} + 1 \right) \right)$$

*and $T_s = n + \sqrt{n} T_c$ for the finite-sum case and $T_s = \frac{\sigma}{\varepsilon} \cdot \max\{\kappa, (1-\rho)^{-2}\} T_c$ for the general stochastic case.*

**Remark 4.1** *The assumption on $\left\|\mathbf{V}_{\perp,\mathbf{x}}^{(0)}\right\|_F^2$ and $\left\|\widetilde{\mathbf{Y}}^{(0)} - \mathbf{Y}^{(0)}\right\|_F^2$ is mild and can easily hold if $\kappa(1-\rho)$ is not small. Otherwise, multiple communications can be performed at the initial step to satisfy the condition. In addition, when $\kappa(1-\rho)^2 \geq 1$, the dominant terms in the complexity results will be independent of the graph topology. In this case, our stepsize $\eta_{\mathbf{x}}$ and $\eta_{\mathbf{\Lambda}}$ in (52) are both in the order of $\frac{1}{L\kappa^2}$, matching to that used by a centralized method for NCSC minimax problems, e.g., the GDA in (Lin et al., 2020a). This can be significantly larger than the stepsize in the order of $\frac{1}{L\kappa^3}$ taken by state-of-the-art decentralized methods for minimax problems, e.g., PRECISION (Liu et al., 2023) and DSGDA (Gao, 2022). When $\kappa(1-\rho)^2 \ll 1$, then $T_c$ linearly depends on $(1-\rho)^{-4}$, which is worse than existing results with a dependence of $(1-\rho)^{-2}$ for decentralized methods with a single communication per iteration on solving composite nonconvex problems; see (Scutari & Sun, 2019) for example. To improve the dependence, we can again perform multiple communications in the initial step to have $\hat{\delta}_0 = \mathcal{O}(\kappa(1-\rho)^2)$ and obtain $T_c = \Theta\left(\frac{L\kappa}{\varepsilon^2(1-\rho)^2}\right)$, yielding a lower-order dependence on $\kappa$. Moreover, if the multi-communication trick, which needs more coordinations between agents, is applied for every update, we can change the mixing matrix $\mathbf{W}$ in our analysis to a polynomial in $\mathbf{W}$, denoted by $q(\mathbf{W})$. If $\mathbf{W}$ is symmetric, we can apply the Chebyshev polynomial (e.g., see (Mancino-Ball et al., 2023b)) to have an accelerated averaging. This way, the sample complexity will be independent of $\rho$ and the communication complexity will linearly depend on $(1-\rho)^{-\frac{1}{2}}$.*

## 4.2 STORM variance reduction

**Theorem 4.2** *Under Assumptions 1,2, and 3, let $\{(\mathbf{X}^{(t)}, \mathbf{\Lambda}^{(t)}, \mathbf{Y}^{(t)})\}_{t\geq 0}$ be generated from Algorithm 1 with VR-tag = STORM and $\beta \in (0,1)$, $\eta_{\mathbf{y}} = \frac{\sqrt{\beta}}{4\sqrt{2}L}$, and*

$$\eta_{\mathbf{x}} = \min\left\{\frac{\kappa(1-\rho)^2}{40L(24\kappa^2 + 8\kappa + 5)}, \frac{\sqrt{\beta}}{48(L+1)(24\kappa^2 + 7\kappa + 4)}\right\},$$

$$\eta_{\mathbf{\Lambda}} = \min\left\{\frac{(1-\rho)^2}{4L(20\kappa + 3)}, \frac{\sqrt{\beta}}{4(L+1)(52\kappa^2 + \kappa + 1)}\right\}.$$

*For any integer $T \geq 1$, select $\tau$ uniformly at random from $\{0, \ldots, T-1\}$. Then it holds that*

$$\mathbb{E}\left\|\frac{1}{\eta_{\mathbf{x}}}\left(\bar{\mathbf{x}}^{(\tau)} - \mathbf{prox}_{\eta_{\mathbf{x}}g}\left(\bar{\mathbf{x}}^{(\tau)} - \eta_{\mathbf{x}}\nabla P(\bar{\mathbf{x}}^{(\tau)}, \mathbf{\Lambda}^{(\tau)})\right)\right)\right\|_2^2 + \frac{L^2}{m}\mathbb{E}\left\|\mathbf{X}_{\perp}^{(\tau)}\right\|_F^2 \tag{20}$$

$$\leq \left(\frac{C_0}{T} + \left(\frac{1}{(1-\rho)^2\kappa L} + \frac{(1+1/L)}{\sqrt{\beta}L}\right)\beta^2 \sum_{t=0}^{T-1}\frac{\Upsilon_{t+1}}{T}\right)$$

$$\cdot \left[\frac{20}{\eta_{\mathbf{x}}} + 6\eta_{\mathbf{x}}\left(2L^2(3+5\kappa^2) + \frac{5}{\eta_{\mathbf{x}}^2}\right) + \frac{1600\kappa^2 L}{\sqrt{\beta}} + \frac{80L}{\sqrt{\beta}} + \frac{5(1-\rho)^2}{\eta_{\mathbf{x}}}\right]$$

*and*

$$\mathbb{E}\left\|\nabla_{\mathbf{\Lambda}} P(\bar{\mathbf{x}}^{(\tau)}, \mathbf{\Lambda}^{(\tau)})\right\|_F^2 \tag{21}$$

$$\leq \left(\frac{C_0}{T} + \left(\frac{1}{(1-\rho)^2\kappa L} + \frac{(1+1/L)}{\sqrt{\beta}L}\right)\beta^2 \sum_{t=0}^{T-1}\frac{\Upsilon_{t+1}}{T}\right) \cdot \left[\frac{4}{\eta_{\mathbf{\Lambda}}} + 24\kappa^2 L^2 \eta_{\mathbf{x}} + \frac{640\kappa^2 L}{\sqrt{\beta}}\right],$$

*where $\Upsilon_t := \frac{\sigma^2}{\mathcal{S}_t}$ for any $t \geq 0$, and $C_0$ depends on the initial points, defined in (77).*

Below, we give the sample and communication complexity result of Algorithm 1 with VR-tag == STORM. The proof is given in Appendix C.5.

**Corollary 4.2** *Let $\varepsilon \in \left(0, \sigma(1-\rho)^2\right]$ be given and assume $L \geq 1$. Under the same conditions as in Theorem 4.2, assume $\left\|\mathbf{V}_{\perp,\mathbf{x}}^{(0)}\right\|_F^2 \leq 40m(1-\rho)\kappa L$, $\left\|\widetilde{\mathbf{Y}}^{(0)} - \mathbf{Y}^{(0)}\right\|_F^2 = \mathcal{O}\left(\frac{m\kappa}{L}\right)$, and $\mathbb{E}\left\|\mathbf{R}^{(0)}\right\|_F^2 \leq \frac{m\sigma^2}{\mathcal{S}_0}$ with $\mathcal{S}_0 = \left\lceil\left(\frac{1}{\sqrt{\beta}L} + \frac{1}{4(1-\rho)^2\kappa L}\right)\sigma^2\right\rceil$ and $\mathcal{S}_t = \mathcal{O}(1)$ for all $t \geq 1$. Then, Algorithm 1 with VR-tag = STORM and*

$$\beta = \frac{\varepsilon^2}{1440\sigma^2(24\kappa^2 + 7\kappa + 4)}, \tag{22}$$

can find an $\varepsilon$-stationary point in expectation of (7) by $T_s = \Theta\left(\frac{\sigma\kappa^3 L}{\varepsilon^3} + \frac{\sigma^3\kappa}{\varepsilon L}\right)$ stochastic gradients and $T_c = \Theta\left(\frac{\sigma\kappa^3 L}{\varepsilon^3}\right)$ local neighbor communications.

A few remarks are in order regarding the above corollary.

**Remark 4.2** *First, similar to other recent works (Xin et al., 2021b; Mancino-Ball et al., 2023a), our complexity results are stated for the high-accuracy regime (i.e. small enough $\varepsilon$). Technically, this accuracy requirement removes the final dependence of* VRLM*-STORM on the spectrum of the graph. Second, the initialization requirements in Corollary 4.2 are akin to those in (Mancino-Ball et al., 2023a). If necessary, e.g., when $1 - \rho$ is much smaller than $\frac{1}{\kappa L}$, we can perform multiple communications at the initial step to have $\left\|\mathbf{V}_{\perp,\mathbf{x}}^{(0)}\right\|_F^2 \leq 40m(1 - \rho)\kappa L$. Third, the recent Acc-MDA (Huang et al., 2022) method can attain a convergence rate of $\tilde{\mathcal{O}}(\kappa^{4.5}\varepsilon^{-3})$, however,* this is only in the single-agent setting *where $m = 1$ and further, this result only holds when $g \equiv h \equiv 0$. When $m = 1$, we have $\mathbf{W} = 1$, which eliminates the variable $\mathbf{\Lambda}$, as well as the consensus constraint on $\{\mathbf{x}_i\}_{i\in\mathcal{V}}$. Hence, (7) is actually identical to (1) which indicates our analysis provides a better dependence on $\kappa$, as well proves convergence on a broader class of problems (i.e., $g \not\equiv 0$ and/or $h \not\equiv 0$).*

In addition, we make a remark regarding the dependence of our results on the number of agents, $m$.

**Remark 4.3** *Similar to existing works (Liu et al., 2023; Zhang et al., 2021), the complexity results presented in Corollaries 4.1 and 4.2 are sub-optimal in terms of the dependence on the number of agents, $m$. Specifically, for the finite-sum setting using the SPIDER-type variance reduction, we should expect $T_s = n + \sqrt{\frac{n}{m}}T_c$ as is found in (Chen et al., 2024) and for the general stochastic setting using the STORM-type variance reduction, we should expect to see a linear speed-up of $T_s$ in $m$ (Mancino-Ball et al., 2023a). This shortcoming is due to the coefficient on the stochastic gradient error term, $\left\|\mathbf{R}^{(t)}\right\|_F^2$, in Lemma C.5 being $m$ times too large. While we acknowledge this shortcoming, we stress that (Chen et al., 2024) utilizes the multi-communication trick in their analysis which pushes a method closer to its centralized counterpart. Without using multi-communication, it is still an open question whether or not the optimal sample complexity can be achieved for the decentralized minimax problem with or without a regularizer.*

## 5 Numerical experiments

In this section, we empirically validate our proposed methods on two benchmark problems which fit (3). We compare our methods to three methods: DPSOG (Liu et al., 2020), DM-HSGD (Xian et al., 2021), and GT-SRVR (Zhang et al., 2021); since these methods are only presented for the case of (3) with $g \equiv 0$, we simply wrap their $\mathbf{x}_i$ updates with $\mathbf{prox}_{\eta_{\mathbf{x}}g}(\cdot)$. For the experiments in this section, we use $m = 8$ agents, where each agent is an NVIDIA Tesla V100 GPU. The agents are connected via a ring structured graph, where self-weighting and neighbor weighting are $\frac{1}{3}$. This represents one of the worst-case connectivity patterns (in terms of $\rho$) which still fit Assumption 3 and hence serves as a good benchmark for comparing decentralized methods (Lu & De Sa, 2021; Singh et al., 2021). We show results in both epoch and iteration number. We define one epoch as one pass of the *global* dataset[2] and one iteration as one update step of a method[3]. In Appendix B, we provide additional experiments under extreme data heterogeneity, different communication topologies, and larger agent counts.

### 5.1 Sparse distributionally robust optimization

We test our proposed method on the decentralized distributionally robust optimization problem (Xu, 2024) using both the MNIST (Lecun et al., 1998) and Fashion-MNIST (Xiao et al., 2017) datasets. Each agent maintains a local dataset, $\{\mathbf{a}_{ij}, b_{ij}\}_{j=1}^n$, where $\mathbf{a}_{ij} \in \mathbb{R}^{28\times 28}$ is the $j$-th image on the $i$-th agent and $b_{ij} \in$

---

[2]In our experiments, the datasets are split evenly among the agents, hence one full local agent dataset pass will amount to one full global dataset pass due to the synchronicity of the compared methods.

[3]Since each method may communicate a different number of variables at each iteration, we compare across iterations to keep in line with the theory of each method.

$\{1, \ldots, C\}$ is the corresponding label among $C$ classes. The total number of data points is given by $N = mn$ and we let $\mathcal{J}_i \subseteq \{1, \ldots, N\}$ be an index set which contains the indices of data points on agent $i$. The agents' local objective functions are given by

$$f_i(\mathbf{x}_i, \mathbf{y}_i) = m \sum_{j \in \mathcal{J}_i} (\mathbf{y}_{i,j}) \ell \left( Z_{\mathbf{x}_i}(\mathbf{a}_{ij}), b_{ij} \right) - \frac{\mu}{2} \left\| \mathbf{y}_i - \frac{1}{N} \right\|_2^2, \quad h(\mathbf{y}_i) \equiv \mathbb{I}_{\Delta_N}(\mathbf{y}_i), \quad g(\mathbf{x}_i) \equiv \lambda \|\mathbf{x}_i\|_1. \tag{23}$$

Here, $\mathbf{y}_{i,j}$ denotes the $j$-th component of $\mathbf{y}_i$, $\ell$ is the cross-entropy loss function taken over each class, $Z_{\mathbf{x}_i}$ is a neural network governed by the parameter $\mathbf{x}_i$, and $\mu$ is a parameter controling the deviation from the uniform distribution. The set $\Delta_N := \{\mathbf{y} : \mathbf{y}^\top \mathbf{1} = 1, \mathbf{y} \geq \mathbf{0}\}$ is the standard probability simplex. The regularizer $g$ induces sparsity on the $\mathbf{x}$ variable. We choose $Z_{\mathbf{x}_i}$ to be a two-layer neural network with 200 hidden units and Tanh activation function. Further, we let $\mu = 10.0$ and $\lambda = 5 \times 10^{-4}$.

The dataset is split uniformly at random among the agents, hence each agent receives $n = 7,500$ local data points. We let the mini-batch size for all methods be 100 and tune $\eta_\mathbf{y} \in \{0.1, 0.01, 0.001, 0.0001\}$ and $\frac{\eta_\mathbf{x}}{\eta_\mathbf{y}} \in \{1, 0.1, 0.01, 0.001\}$. For DM-HSGD, we tune $\beta_\mathbf{x} = \beta_\mathbf{y} = 0.01$ following the paper guidelines and set the initial batch-size to 3000. For GT-SRVR, we tune $q \in \{100, 300\}$ and the large mini-batch size from $\{3000, 7500\}$. For VRLM-STORM we let $\beta = 0.01$, $L = \frac{2}{\sqrt{m}}$, and $\eta_\mathbf{\Lambda} = 0.001$. For VRLM-SPIDER, we tune $q \in \{100, 300\}$ and the large mini-batch size from $\{3000, 7500\}$ and let $L = \frac{2}{\sqrt{m}}$ and $\eta_\mathbf{\Lambda} = 0.001$. We run all methods to 50,000 iterations and compare the training loss value (as if performing centralized training; note this is not the objective value), testing accuracy, average number of non-zeros, and stationarity violation, where the stationarity violation is computed as

$$\left\| \bar{\mathbf{x}} - \mathbf{prox}_g \left( \bar{\mathbf{x}} - \frac{1}{m} \sum_{i=1}^m \nabla_\mathbf{x} f_i(\bar{\mathbf{x}}, \mathbf{y}^{(*)}) \right) \right\|_2^2 + \|\mathbf{X}_\perp\|_F^2 + \|\mathbf{Y}_\perp\|_F^2 \tag{24}$$

where $\mathbf{y}^{(*)} := \arg\max_\mathbf{y} \frac{1}{m} \sum_{i=1}^m f_i(\bar{\mathbf{x}}, \mathbf{y}) - h(\mathbf{y})$ for $\bar{\mathbf{x}} = \frac{1}{m} \sum_{i=1}^m \mathbf{x}_i$. We run each experiment with 5 different initial seeds and report the average results across both iterations and epochs.

From Figure 1, we can see that VRLM (both variants) outperform all competing methods in terms of training loss and testing accuracy, while VRLM-STORM can additionally find sparser solutions. For this problem, DM-HSGD is not competitive when $\lambda > 0$. All methods appear to struggle with reducing (24) for both datasets, but we remark that this does not appear to affect each method's ability to minimize the training loss and obtain reasonable testing accuracy.

## 5.2 Fair Classification

We also test our method on the Fair Classification problem (Mohri et al., 2019) using the CIFAR-10 (Krizhevsky, 2009) dataset. Each agent maintains a local dataset, $\{\mathbf{a}_{ij}, b_{ij}\}_{j=1}^n$, where $\mathbf{a}_{ij} \in \mathbb{R}^{32 \times 32}$ is the $j$-th image on the $i$-th agent and $b_{ij} \in \{1, \ldots, C\}$ is the corresponding label. The agents' local objective functions are given by

$$f_i(\mathbf{x}_i, \mathbf{y}_i) = \sum_{c=1}^C (\mathbf{y}_{i,c}) \ell \left( Z_{\mathbf{x}_i}(\{\mathbf{a}_{ij}\}_{b_{ij}=c}), \{b_{ij}\}_{b_{ij}=c} \right) - \frac{\mu}{2} \|\mathbf{y}_i\|_2^2, \quad h(\mathbf{y}_i) \equiv \mathbb{I}_{\Delta_C}(\mathbf{y}_i), \quad g(\mathbf{x}_i) \equiv 0, \tag{25}$$

where $\mathbf{y}_{i,c}$ denotes the $c$-th component of $\mathbf{y}_i$, $\ell$ is the cross-entropy loss function which computes the average loss over each class, $Z_{\mathbf{x}_i}$ is a neural network governed by the parameter $\mathbf{x}_i$, $\mu$ is a parameter to tune, and $\Delta_C$ is the standard probability simplex. We choose $Z_{\mathbf{x}_i}$ to be the All-CNN network (Springenberg et al., 2015) and let $\mu = 0.1$.

The dataset is split uniformly at random among the agents, hence each agent receives $n = 6,250$ local data points. We let the mini-batch size for all methods be 100 and tune $\eta_\mathbf{y} \in \{0.1, 0.2, 0.3, 0.4, 0.5\}$ and $\frac{\eta_\mathbf{x}}{\eta_\mathbf{y}} \in \{1, 0.1, 0.01, 0.001\}$. For DM-HSGD, we tune $\beta_\mathbf{x} = \beta_\mathbf{y} \in \{0.01, 0.1, 0.5, 0.8, 0.9, 0.99\}$ and set the initial batch-size to 3000. For GT-SRVR, we tune $q \in \{100, 300\}$ and the large mini-batch size from $\{2000, 3000\}$. For VRLM-STORM, we tune $\beta \in \{0.01, 0.1, 0.5, 0.8, 0.9, 0.99\}$ and let $L = \frac{2}{\sqrt{m}}$ and $\eta_\mathbf{\Lambda} = 0.1$. We found VRLM-SPIDER noncompetitive on this instance, and the results are omitted. We run all methods to 20,000 iterations and compare the training loss value (as if performing centralized training; note this is not the objective value), testing accuracy, and stationarity violation, where the stationarity violation is computed as

$$\left\| \frac{1}{m} \sum_{i=1}^m \nabla_\mathbf{x} f_i(\bar{\mathbf{x}}, \mathbf{y}_i^{(*)}) \right\|_2^2 + \|\mathbf{X}_\perp\|_F^2 + \|\mathbf{Y}_\perp\|_F^2 \tag{26}$$

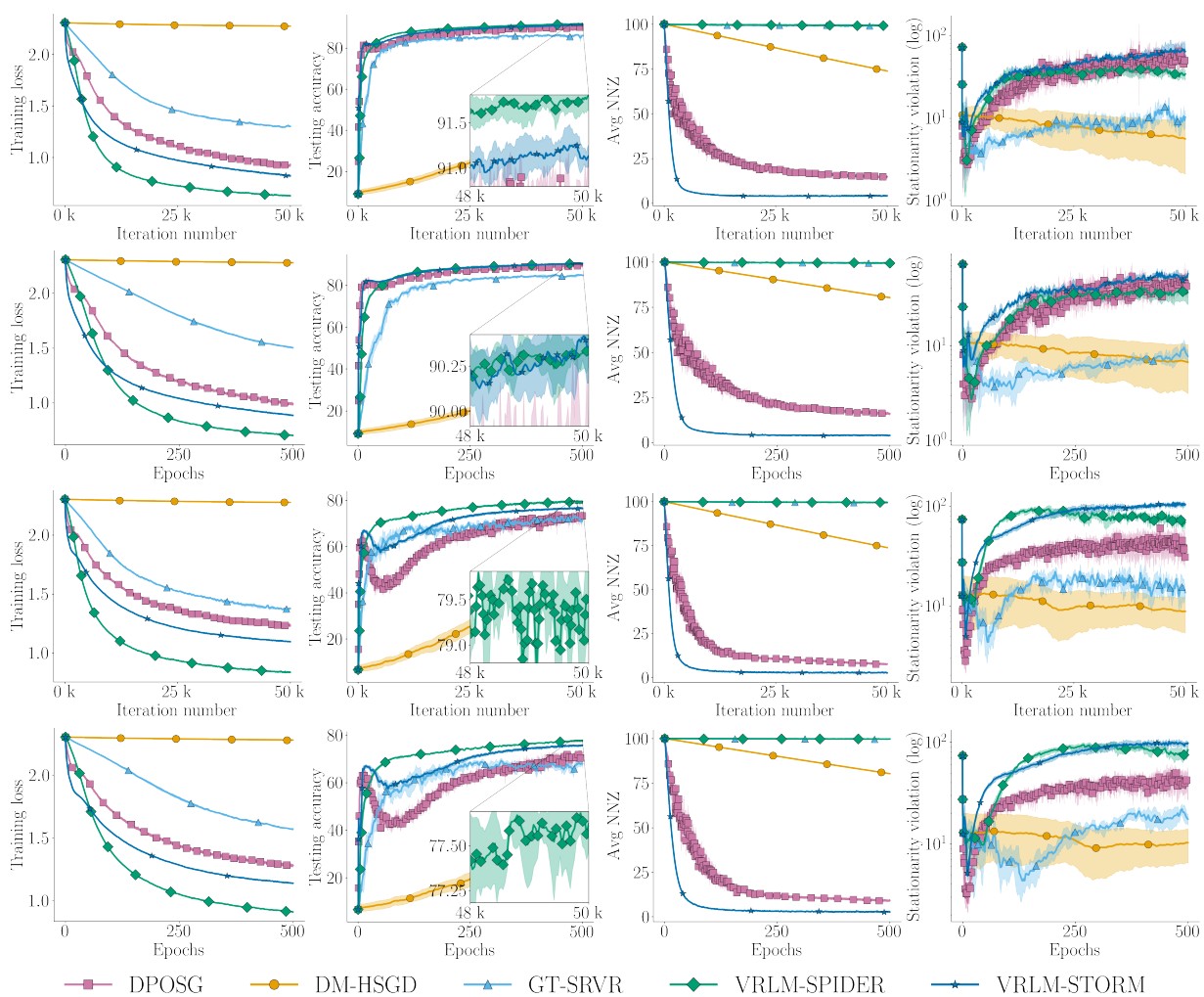

Figure 1: Experimental results for the sparse distributionally robust optimization problem (23) using the MNIST and Fashion-MNIST datasets. The top two row depict results for the MNIST dataset in terms of iteration number and epochs, respectively. The bottom two rows depict the same results for the Fashion-MNIST dataset. Shaded regions represent 95% confidence intervals computed over 5 trials.

where $\mathbf{y}_i^{(*)} := \arg\max_{\mathbf{y}_i} f_i(\bar{\mathbf{x}}, \mathbf{y}_i) - h(\mathbf{y}_i)$ for $\bar{\mathbf{x}} = \frac{1}{m} \sum_{i=1}^m \mathbf{x}_i$. We use (26) instead of (24) due to the memory constraint of our computing environment which prohibits us from putting all the data on one machine. Again, we run each experiment with 5 different initial seeds and report the average results across both iterations and epochs.

From Figure 2, we can see that DM-HSGD and our proposed method perform similarly in terms of stationarity violation. However, our method can achieve higher testing accuracy and lower training loss. DPOSG yields higher testing accuracy, but its convergence for the double-regularized problem (7) has not been studied. The GT-SRVR method is not competitive on this problem, which could be due to the lack of theoretical guarantees for the double-regularized problem (7), or due to the periodic large-batch stochastic gradient computation. Overall, we find our proposed method to be competitive against other state-of-the-art methods, while providing improved theoretical guarantees.

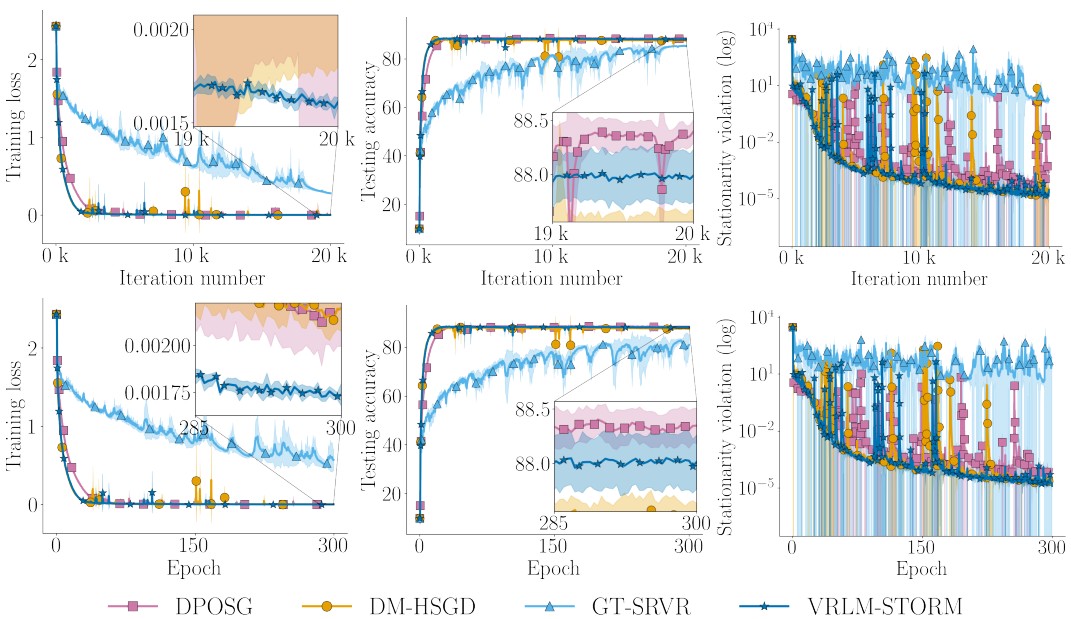

Figure 2: Experimental results for the Fair Classification problem (25) using the All-CNN network with the CIFAR-10 dataset. The top row depicts results in terms of iteration number, while the bottom is in terms of epochs. Shaded regions represent 95% confidence intervals computed over 5 trials.

## 6 Conclusions

In this work, we have presented the `Variance Reduced Lagrangian Multiplier` (`VRLM`) based method for solving the decentralized, double-regularized, stochastic nonconvex strongly-concave minimax problem. We analyzed `VRLM` with both large-batch and small-batch variance-reduction techniques. Under mild assumptions, both versions are able to achieve the best-known complexity results that are achieved by existing methods for solving special cases of the problem we consider. Finally, we demonstrated the effectiveness of our proposed methods in a real decentralized computing environment on two benchmark machine learning problems.

## Acknowledgements

The authors would like to thank three anonymous reviewers for valuable comments to improve the quality of the paper. This work is partly supported by DMS-2208394, the ONR award N00014-22-1-2573, and also by IBM through the IBM-Rensselaer Future of Computing Research Collaboration..

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

## A  Additional related work

All the methods we mentioned in Section 1.2 assume strong concavity about the dual variable $\mathbf{y}$ and also convexity of the regularization terms or constraint sets, if there are any. A few existing decentralized methods for minimax problems make weaker or different assumptions. For example, DPOSG (Liu et al., 2020) is designed to solve smooth stochastic nonconvex nonconcave decentralized minimax problems. Instead of concavity structure, the Minty VI condition is assumed by DPOSG in order to have guaranteed convergence. However, its complexity has a high-order dependence on a given accuracy, reaching $\mathcal{O}(\varepsilon^{-12})$ to produce an $\varepsilon$-stationary solution. In addition, it cannot handle problems with nonsmooth regularization terms or a hard constraint. This is another fundamental difference from our method. We summarize the settings of a few representative methods and their complexity results in Table 1. Notice that these methods adopt different notions of $\varepsilon$-stationarity so their complexity results are not directly comparable. However, their broadness on applicability is revealed from the settings on $g$ and $h$.

In an orthogonal axis, we note that VRLM leverages gradient tracking to overcome divergence in heterogeneous data environments (Tang et al., 2018a). An alternative to gradient tracking is to use exact diffusion-type updates (Shi et al., 2015; Cai et al., 2025). Recent work has shown that gradient tracking and exact diffusion are closely related; exact diffusion can be viewed as an implementation of *implicit* gradient tracking (Li et al., 2022; Li & Su, 2022). It is an open question whether or not changing our scheme to use exact diffusion would improve our analytical results or numerical performance.

Though our focus is on decentralized computation, our method is also applicable in the single-agent (or non-distributed) setting, i.e., the problem (1) with $m = 1$. In this case, many methods have been proposed under various settings of $f$. The GDA (Lin et al., 2020a) method can be applied to dual-constrained smooth deterministic NCSC minimax problems, i.e., $g \equiv 0$ and $h = \mathbb{I}_{\mathcal{Y}}$ in (1). To achieve an $\varepsilon$-accurate solution, it requires $\mathcal{O}(\kappa^2 \varepsilon^{-2})$ gradient evaluations, which was later reduced to $\tilde{\mathcal{O}}(\sqrt{\kappa}\varepsilon^{-2})$ by the Minimax-PPA (Lin et al., 2020b) method. When $g \not\equiv 0$ or $h \not\equiv 0$, proximal-AltGDAm (Chen et al., 2021) can achieve a complexity result of $\mathcal{O}(\kappa^{\frac{11}{6}}\varepsilon^{-2})$. In the stochastic setting, GDA utilizes large-batch sampling and can

Table 1: Representative decentralized minimax optimization methods: GT-DA (Tsaknakis et al., 2020), GT-SRVR (Zhang et al., 2021), PRECISION (Liu et al., 2023), and DREAM (Chen et al., 2022). *Notice that these methods adopt different notions of $\varepsilon$-stationarity.* The Stoch./F.S. column indicates whether the method handles a stochastic setting or a finite-sum setting in (2); S.C. stands for "single communication" and hence an "No" in this column indicates a method requires multiple communications to ensure convergence. The final two columns, Samp. Comp. and Comm. Comp., indicate the complexity results to produce an $\varepsilon$-stationary solution by each method. Here, the $\mathcal{O}(\cdot)$ notation hides dependence on non-key values; some works do not make the dependence on the spectral gap or condition number clear - we use constants $a, b, c, d, e, f > 0$ with subscripts $s$ and $c$ to denote whether or not these unknowns are related to the *sample* or *communication* complexities, respectively. ¶PRECISION is guaranteed to converge in the finite-sum setting, but the dependence upon the number of local component functions $n$ is unclear. †For cleanness, we assume here that $\varepsilon \leq (1-\rho)^2$. ‡We assume here that $1 \leq \kappa(1-\rho)^2$; in these two regimes, the dominant terms in our results are graph-topology independent. For results of other regimes, please see Corollaries 4.1 and 4.2.

| Method | Stoch./F.S. | $(g, h)$ | S.C. | Samp. Comp. | Comm. Comp. |
|---|---|---|---|---|---|
| GT-DA | F.S. | $(0, \mathbb{I}_\mathcal{Y})$ | Yes | $\tilde{\mathcal{O}}\left(\frac{n\kappa^{a_s}}{(1-\rho)^{b_s}\varepsilon^2}\right)$ | $\tilde{\mathcal{O}}\left(\frac{\kappa^{a_c}}{(1-\rho)^{b_c}\varepsilon^2}\right)$ |
| GT-SRVR | F.S. | $(0, 0)$ | Yes | $\mathcal{O}\left(n + \frac{\sqrt{n}\kappa^{c_s}}{(1-\rho)^{d_s}\varepsilon^2}\right)$ | $\mathcal{O}\left(\frac{\kappa^{c_c}}{(1-\rho)^{d_c}\varepsilon^2}\right)$ |
| PRECISION¶ | F.S. | $(\text{convex}, \mathbb{I}_\mathcal{X})$ | Yes | $\mathcal{O}\left(\frac{\kappa^{e_s}}{(1-\rho)^{f_s}\varepsilon^2}\right)$ | $\mathcal{O}\left(\frac{\kappa^{e_c}}{(1-\rho)^{f_c}\varepsilon^2}\right)$ |
| DREAM | F.S. | $(0, \mathbb{I}_\mathcal{Y})$ | No | $\mathcal{O}\left(n + \frac{\sqrt{n}\kappa^2}{\varepsilon^2}\right)$ | $\mathcal{O}\left(\frac{\kappa^2}{\sqrt{1-\rho}\varepsilon^2}\right)$ |
| | Stoch. | | | $\mathcal{O}\left(\frac{\kappa^3}{\varepsilon^3}\right)$ | $\mathcal{O}\left(\frac{\kappa^2}{\sqrt{1-\rho}\varepsilon^2}\right)$ |
| VRLM-STORM† | Stoch. | (convex, convex) | Yes | $\mathcal{O}\left(\frac{\kappa^3}{\varepsilon^3}\right)$ | $\mathcal{O}\left(\frac{\kappa^3}{\varepsilon^3}\right)$ |
| VRLM-SPIDER‡ | F.S. | (convex, convex) | Yes | $\mathcal{O}\left(n + \frac{\sqrt{n}\kappa^2}{\varepsilon^2}\right)$ | $\mathcal{O}\left(\frac{\kappa^2}{\varepsilon^2}\right)$ |
| | Stoch. | | | $\mathcal{O}\left(\frac{\kappa^3}{\varepsilon^3}\right)$ | $\mathcal{O}\left(\frac{\kappa^2}{\varepsilon^2}\right)$ |

achieve a sample complexity of $\mathcal{O}(\kappa^3\varepsilon^{-4})$. SREDA (Luo et al., 2020) considers both stochastic and finite-sum structured minimax problems. Similar to one choice of our method in the single-agent setting, it adopts the SPIDER-type VR. However, different from our method, SREDA needs an inner loop to approximately solve the dual maximization problem, and thus it is a double-loop method. It achieves a sample complexity of $\tilde{\mathcal{O}}(n + \sqrt{n}\kappa^2\varepsilon^{-2})$ for the finite-sum case and $\mathcal{O}(\kappa^3\varepsilon^{-3})$ for the stochastic case. All aforementioned single-agent methods require a large batch-size: either all samples in a deterministic/finite-sum setting or as many as $\Theta(\varepsilon^{-2})$ in a stochastic setting where SPIDER-type VR is used. In contrast, Acc-MDA (Huang et al., 2022) can achieve a complexity of $\tilde{\mathcal{O}}(\kappa^{\frac{9}{2}}\varepsilon^{-3})$ by $\mathcal{O}(1)$ samples per update through the STORM-type variance reduction. SAPD+ (Zhang et al., 2022) can also have convergence guarantees by small-batch sampling and achieves a complexity of $\mathcal{O}(\kappa\varepsilon^{-4})$. When large-batch sampling is performed, SAPD+ can have a complexity of $\mathcal{O}(\kappa^2\varepsilon^{-3})$ by variance reduction. However, different from Acc-MDA and our method in a single-agent setting, SAPD+ is a double-loop method. A comprehensive literature review for single-agent methods designed for solving (1) is out of the scope of this work. The interested readers can refer to the references therein of previously mentioned works for a more thorough treatment of this subject, including whether or not each method can handle $g \not\equiv 0$ or $h \not\equiv 0$. For readers interested in works that handle the nonconvex concave or nonconvex nonconcave settings, see e.g., (Ostrovskii et al., 2021; Thekumparampil et al., 2019; Zhang et al., 2022; Jin et al., 2020; Lin et al., 2020a; Yang et al., 2022; Xu et al., 2023; Li et al., 2025; Khan & Xu, 2026) and the references therein.

## B   Additional numerical experiments

We conduct two additional experiments on the sparse distributionally robust optimization problem (23): heterogeneous data and increased agent count. These experiments are conducted on a 2024 M4-Macmini.

We vary the communication topology between a ring structure (each agent is connected to two other agents) and a grid structure (each agent has between two and four neighbors; two for corners, three for non-corner edges, and four for internal agents). Both experiments utilize the MNIST and Fashion-MNIST dataset.

## B.1 Experiments with Heterogeneous Data

To verify the performance of VRLM in a heterogeneous data setting, we increase the agent count to 10 and partition the data based on the labels such that each of the 10 classes in the MNIST and Fashion-MNIST dataset is distributed to one agent. We utilize a ring structured communication topology and a 2-by-5 grid communication topology. We adopt the same tuned hyperparameters as in Section 5.1 for each method and perform three independent trials.

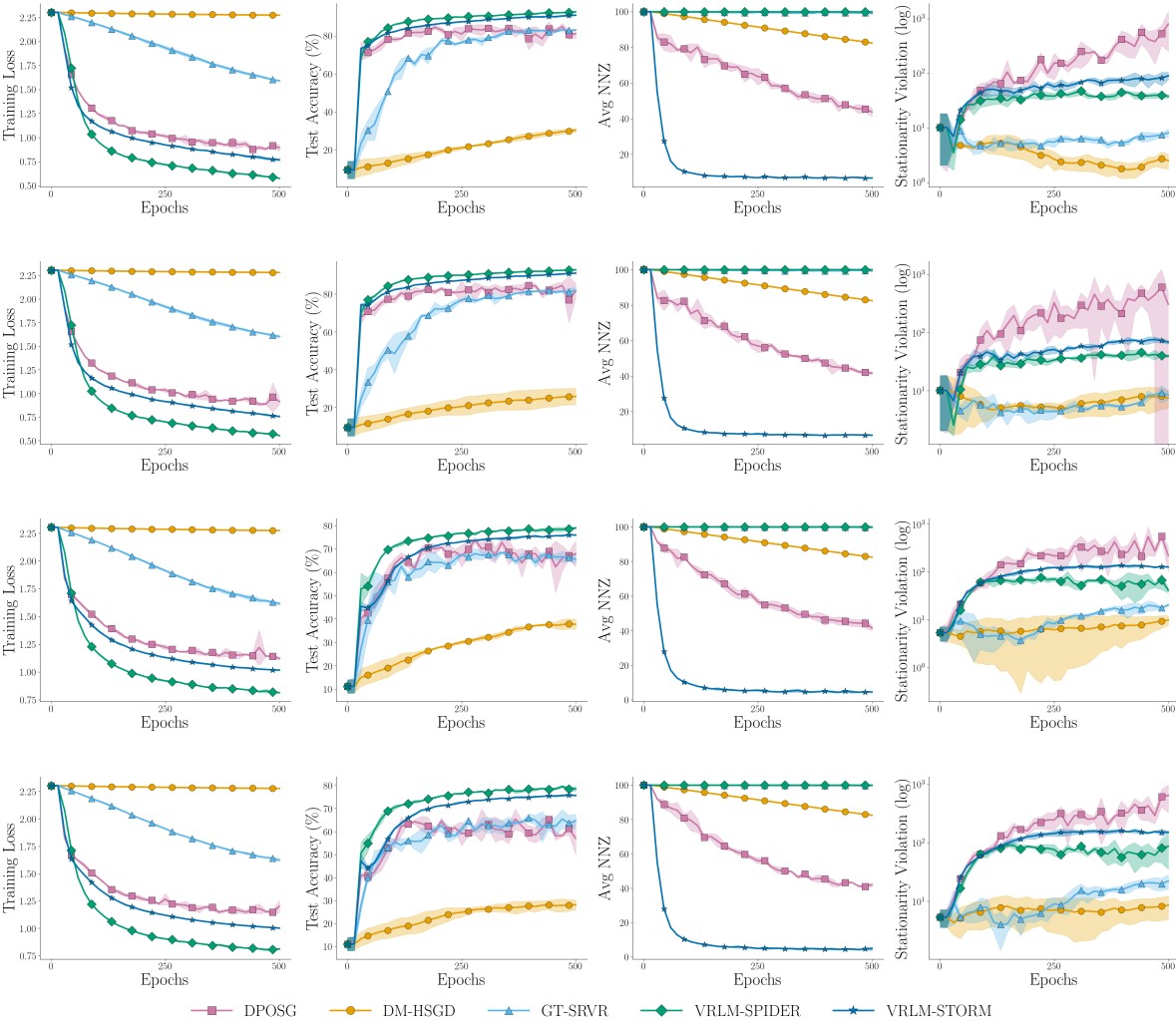

Figure 3: Experimental results for the sparse distributionally robust optimization problem (23) using the MNIST and Fashion-MNIST datasets in the heterogeneous setting. The first two rows depict results for the MNIST dataset using ring-structured and ladder-structured graphs of 10 agents, respectively. The bottom two rows depict the same results for the Fashion-MNIST dataset. Shaded regions represent 95% confidence intervals computed over three trials.

From Figure 3, both VRLM-SPIDER and VRLM-STORM consistently outperform other competing methods in terms of training loss and testing accuracy. VRLM-STORM yields the most sparse solutions. As before, all methods struggle to reduce the stationarity violation across both datasets.

## B.2 Experiments with Homogeneous Data and more agents

In this subsection, we adopt the homogeneous dataset split (i.e. uniform random based on label) but increase the agent count to 20. We utilize both a ring structued communication topology and a 4-by-5 grid communication topology. We perform slight tuning on the primal and dual learning rates for VRLM while leaving other hyperparameters fixed. For VRLM-SPIDER, we decreased the learning rates from $10^{-4}$ to $4 \times 10^{-5}$ on both datasets and for VRLM-STORM we decreased the learning rates from $10^{-2}$ to $4 \times 10^{-3}$ on the Fashion-MNIST dataset. Without decreasing the learning rates, they will diverge. Three independent trials are conducted.

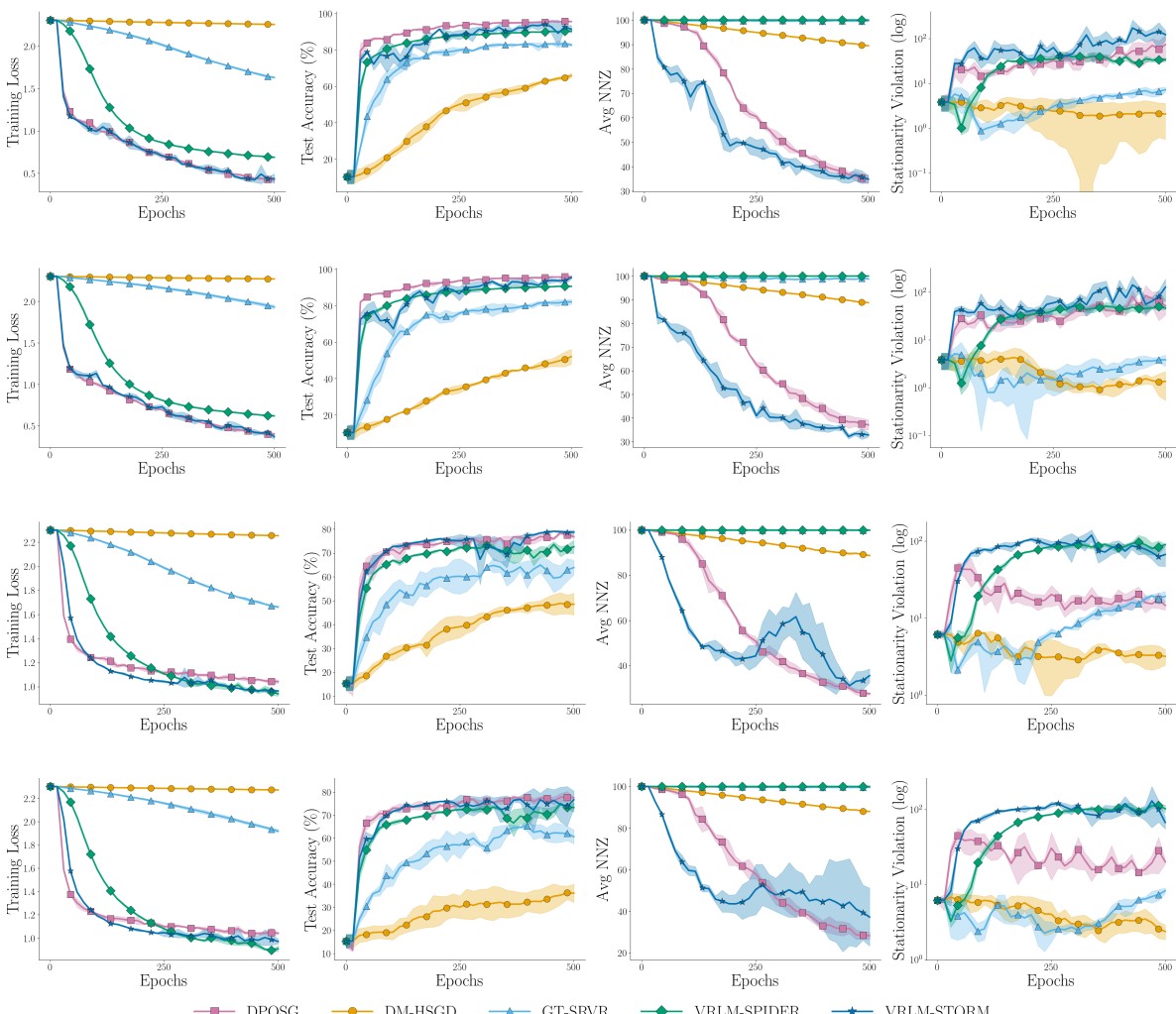

Figure 4: Experimental results for the sparse distributionally robust optimization problem (23) using the MNIST and Fashion-MNIST datasets in the homogeneous setting. The first two rows depict results for the MNIST dataset using ring-structured and grid-structured graphs of 20 agents, respectively. The bottom two rows depict the same results for the Fashion-MNIST dataset. Shaded regions represent 95% confidence intervals over 3 trials.

From Figure 4, we observe that DPOSG consistently gives the highest test accuracy. VRLM-STORM and DPOSG perform the best on MNIST in terms of training loss and on Fashion-MNIST in terms of test accuracy while VRLM-SPIDER achieves the lowest training loss on Fashion-MNIST. Again, all methods struggle to decrease the stationarity violation.

## C  Convergence analysis

We adhere to the following logical exposition of our theoretical analysis; some longer proofs are deferred to additional Appendix sections.

1. In Section C.1, we present necessary preparatory results which build key inequalities around the functions defined in (8) and (9).

2. In Section C.2, we build a one-iteration progress of the `VRLM` method without specifying the VR gradient estimator.

3. In Section C.3, we present bounds on consensus errors and local gradient estimator errors.

4. In Section C.4, we analyze Algorithm 1 with `VR`-tag $==$ SPIDER, demonstrating the convergence of the `VRLM` method in both the finite-sum and expectation structured cases.

5. In Section C.5, we analyze Algorithm 1 with `VR`-tag $==$ STORM, demonstrating the convergence of the `VRLM` method in the expectation structured case.

The following definitions are used throughout our analysis with $\Phi$ given in (6).

$$\widehat{\mathbf{Y}}^{(t)} := \arg\max_{\mathbf{Y}} \Phi(\mathbf{1}\bar{\mathbf{x}}^{(t)}, \boldsymbol{\Lambda}^{(t)}, \mathbf{Y}), \quad \widetilde{\mathbf{Y}}^{(t)} := \arg\max_{\mathbf{Y}} \Phi(\mathbf{X}^{(t)}, \boldsymbol{\Lambda}^{(t)}, \mathbf{Y}), \tag{27}$$

$$\mathbf{R}_{\mathbf{x}}^{(t)} := \mathbf{D}_{\mathbf{x}}^{(t)} - \nabla_{\mathbf{x}} F(\mathbf{Z}^{(t)}), \quad \mathbf{R}_{\mathbf{y}}^{(t)} := \mathbf{D}_{\mathbf{y}}^{(t)} - \nabla_{\mathbf{y}} F(\mathbf{Z}^{(t)}), \quad \mathbf{R}^{(t)} := \mathbf{D}^{(t)} - \nabla F(\mathbf{Z}^{(t)}), \tag{28}$$

$$\Gamma_t(\mathbf{Y}) := \tfrac{1}{m} \sum_{i=1}^{m} f_i(\mathbf{x}_i^{(t)}, \mathbf{y}_i) - \tfrac{L}{2\sqrt{m}} \big\langle (\mathbf{W} - \mathbf{I})\,\mathbf{Y}, \boldsymbol{\Lambda}^{(t)} \big\rangle. \tag{29}$$

By Danskin's theorem (Danskin, 1966), with $\mathbf{Y} = S_\Phi(\mathbf{1}\mathbf{x}^\top, \boldsymbol{\Lambda})$, we have

$$\nabla P(\mathbf{x}, \boldsymbol{\Lambda}) = \Big( \tfrac{1}{m} \sum_{i=1}^{m} \nabla_{\mathbf{x}} f_i(\mathbf{x}, \mathbf{y}_i), \ -\tfrac{L}{2\sqrt{m}} (\mathbf{W} - \mathbf{I})\mathbf{Y} \Big). \tag{30}$$

### C.1  Preparatory results

Under Assumption 3 and by the notation in (17), since $\mathbf{W} - \mathbf{I} = (\mathbf{W} - \mathbf{I})(\mathbf{I} - \tfrac{1}{m}\mathbf{1}\mathbf{1}^\top)$, it holds

$$\begin{aligned}
\big\| \mathbf{X}^{(t+1)} - \mathbf{X}^{(t)} \big\|_F^2 &\le 2 \big\| \mathbf{X}^{(t+1)} - \widetilde{\mathbf{X}}^{(t)} \big\|_F^2 + 2 \big\| (\mathbf{W} - \mathbf{I})\,\mathbf{X}_\perp^{(t)} \big\|_F^2 \\
&\le 2 \big\| \mathbf{X}^{(t+1)} - \widetilde{\mathbf{X}}^{(t)} \big\|_F^2 + 8 \big\| \mathbf{X}_\perp^{(t)} \big\|_F^2 .
\end{aligned} \tag{31}$$

We now present some preparatory propositions. The first one is directly from Xu (2024).

**Proposition C.1** *Let $P$ and $S_\Phi$ be defined in* (8). *Then with $L_P = L\sqrt{4\kappa^2 + 1}$, it holds*

$$\|\nabla P(\mathbf{x}, \boldsymbol{\Lambda}) - \nabla P(\tilde{\mathbf{x}}, \tilde{\boldsymbol{\Lambda}})\|_F^2 \le L_P^2 \big( \|\mathbf{x} - \tilde{\mathbf{x}}\|^2 + \|\boldsymbol{\Lambda} - \tilde{\boldsymbol{\Lambda}}\|_F^2 \big), \forall \mathbf{x}, \tilde{\mathbf{x}} \in \mathrm{dom}(g); \forall \boldsymbol{\Lambda}, \tilde{\boldsymbol{\Lambda}}, \tag{32a}$$

$$\|S_\Phi(\mathbf{X}, \boldsymbol{\Lambda}) - S_\Phi(\widetilde{\mathbf{X}}, \boldsymbol{\Lambda})\|_F^2 \le \kappa^2 \|\mathbf{X} - \widetilde{\mathbf{X}}\|_F^2, \forall \mathbf{x}_i, \tilde{\mathbf{x}}_i \in \mathrm{dom}(g), \forall i; \ \forall \boldsymbol{\Lambda}, \tag{32b}$$

$$\|S_\Phi(\mathbf{X}, \boldsymbol{\Lambda}) - S_\Phi(\widetilde{\mathbf{X}}, \tilde{\boldsymbol{\Lambda}})\|_F^2 \le 2\kappa^2 \|\mathbf{X} - \widetilde{\mathbf{X}}\|_F^2 + 2m\kappa^2 \|\boldsymbol{\Lambda} - \tilde{\boldsymbol{\Lambda}}\|_F^2, \forall \mathbf{x}_i, \tilde{\mathbf{x}}_i \in \mathrm{dom}(g), \forall i; \ \forall \boldsymbol{\Lambda}, \tilde{\boldsymbol{\Lambda}}. \tag{32c}$$

Based on Proposition C.1 and (27), we have

$$\big\| \widehat{\mathbf{Y}}^{(t)} - \widetilde{\mathbf{Y}}^{(t)} \big\|_F^2 \le \kappa^2 \big\| \bar{\mathbf{X}}^{(t)} - \mathbf{X}^{(t)} \big\|_F^2 = \kappa^2 \big\| \mathbf{X}_\perp^{(t)} \big\|_F^2 . \tag{33}$$

**Proposition C.2** *Let $Q$ be defined in* (8). *Then with $L_P = L\sqrt{4\kappa^2 + 1}$, it holds*

$$\begin{aligned}
& m \big\| \nabla_{\mathbf{X}} Q(\mathbf{X}, \boldsymbol{\Lambda}) - \nabla_{\mathbf{X}} Q(\widetilde{\mathbf{X}}, \tilde{\boldsymbol{\Lambda}}) \big\|_F^2 + \big\| \nabla_{\boldsymbol{\Lambda}} Q(\mathbf{X}, \boldsymbol{\Lambda}) - \nabla_{\boldsymbol{\Lambda}} Q(\widetilde{\mathbf{X}}, \tilde{\boldsymbol{\Lambda}}) \big\|_F^2 \\
& \le L_P^2 \Big( \tfrac{1}{m} \|\mathbf{X} - \widetilde{\mathbf{X}}\|_F^2 + \|\tilde{\boldsymbol{\Lambda}} - \boldsymbol{\Lambda}\|_F^2 \Big), \forall \mathbf{x}_i, \tilde{\mathbf{x}}_i \in \mathrm{dom}(g), \forall i; \ \forall \boldsymbol{\Lambda}, \tilde{\boldsymbol{\Lambda}}.
\end{aligned} \tag{34}$$

*Proof.* By the notation in (4), let $\mathbf{Y} = S_\Phi(\mathbf{X}, \mathbf{\Lambda})$ and $\widetilde{\mathbf{Y}} = S_\Phi(\widetilde{\mathbf{X}}, \tilde{\mathbf{\Lambda}})$. We have

$$\nabla Q(\mathbf{X}, \mathbf{\Lambda}) = \left(\tfrac{1}{m}\nabla F(\mathbf{Z})^\top, \ -\tfrac{L}{2\sqrt{m}}(\mathbf{W} - \mathbf{I})\mathbf{Y}\right), \ \nabla Q(\widetilde{\mathbf{X}}, \tilde{\mathbf{\Lambda}}) = \left(\tfrac{1}{m}\nabla F(\tilde{\mathbf{Z}})^\top, \ -\tfrac{L}{2\sqrt{m}}(\mathbf{W} - \mathbf{I})\widetilde{\mathbf{Y}}\right).$$

Hence, by the $L$-smoothness of each $f_i$, it follows

$$m\left\|\nabla_{\mathbf{X}} Q(\mathbf{X}, \mathbf{\Lambda}) - \nabla_{\mathbf{X}} Q(\widetilde{\mathbf{X}}, \tilde{\mathbf{\Lambda}})\right\|_F^2 + \left\|\nabla_{\mathbf{\Lambda}} Q(\mathbf{X}, \mathbf{\Lambda}) - \nabla_{\mathbf{\Lambda}} Q(\widetilde{\mathbf{X}}, \tilde{\mathbf{\Lambda}})\right\|_F^2$$

$$= \tfrac{1}{m}\left\|\nabla F(\mathbf{Z}) - \nabla F(\tilde{\mathbf{Z}})\right\|_F^2 + \tfrac{L^2}{4m}\left\|(\mathbf{W} - \mathbf{I})(\mathbf{Y} - \widetilde{\mathbf{Y}})\right\|_F^2$$

$$\leq \tfrac{L^2}{m}\left(\|\mathbf{X} - \widetilde{\mathbf{X}}\|_F^2 + \|\mathbf{Y} - \widetilde{\mathbf{Y}}\|_F^2\right) + \tfrac{L^2}{m}\|\mathbf{Y} - \widetilde{\mathbf{Y}}\|_F^2.$$

Now use (32c) in the inequality above to obtain the desired result. $\square$

The inequality in (34) indicates the smoothness of $Q$ under a weighted norm. By (Nesterov, 2013, Eqn. 2.12), we have that for any $\mathbf{X}, \widetilde{\mathbf{X}}$ with $\mathbf{x}_i, \tilde{\mathbf{x}}_i \in \mathrm{dom}(g), \forall i \in [m]$ and any $\mathbf{\Lambda}, \tilde{\mathbf{\Lambda}}$,

$$\left|Q(\widetilde{\mathbf{X}}, \tilde{\mathbf{\Lambda}}) - Q(\mathbf{X}, \mathbf{\Lambda}) - \langle \nabla Q(\mathbf{X}, \mathbf{\Lambda}), (\widetilde{\mathbf{X}}, \tilde{\mathbf{\Lambda}}) - (\mathbf{X}, \mathbf{\Lambda})\rangle\right| \leq \tfrac{L_P}{2}\left(\tfrac{1}{m}\|\mathbf{X} - \widetilde{\mathbf{X}}\|_F^2 + \|\tilde{\mathbf{\Lambda}} - \mathbf{\Lambda}\|_F^2\right). \quad (35)$$

The lemma below relates the stationarity violation of (7) to terms that appear in our analysis. This lemma will be utilized to provide our final convergence rate results. Its proof is given in Appendix D.

**Lemma C.1** *Let $\{(\mathbf{X}^{(t)}, \mathbf{\Lambda}^{(t)}, \mathbf{Y}^{(t)})\}$ be generated from Algorithm 1. For any $t \geq 0$, it holds that*

$$\mathbb{E}\left\|\tfrac{1}{\eta_{\mathbf{x}}}\left(\bar{\mathbf{x}}^{(t)} - \mathbf{prox}_{\eta_{\mathbf{x}} g}\left(\bar{\mathbf{x}}^{(t)} - \eta_{\mathbf{x}} \nabla_{\mathbf{x}} P(\bar{\mathbf{x}}^{(t)}, \mathbf{\Lambda}^{(t)})\right)\right)\right\|_2^2 + \tfrac{L^2}{m}\mathbb{E}\left\|\mathbf{X}_\perp^{(t)}\right\|_F^2$$

$$\leq \tfrac{5}{m\eta_{\mathbf{x}}^2}\mathbb{E}\left\|\mathbf{X}^{(t+1)} - \bar{\mathbf{X}}^{(t)}\right\|_F^2 + \left(\tfrac{2L^2(3+5\kappa^2)}{m} + \tfrac{5}{m\eta_{\mathbf{x}}^2}\right)\mathbb{E}\left\|\mathbf{X}_\perp^{(t)}\right\|_F^2 + \tfrac{10L^2}{m}\mathbb{E}\left\|\widetilde{\mathbf{Y}}^{(t)} - \mathbf{Y}^{(t)}\right\|_F^2 \quad (36)$$

$$+ \tfrac{5}{m}\left\|\mathbf{R}^{(t)}\right\|_F^2 + \tfrac{5}{m}\mathbb{E}\left\|\mathbf{V}_{\perp,\mathbf{x}}^{(t)}\right\|_F^2,$$

$$\mathbb{E}\left\|\nabla_{\mathbf{\Lambda}} P(\bar{\mathbf{x}}^{(t)}, \mathbf{\Lambda}^{(t)})\right\|_F^2 \leq \tfrac{2}{\eta_{\mathbf{\Lambda}}^2}\mathbb{E}\left\|\mathbf{\Lambda}^{(t+1)} - \mathbf{\Lambda}^{(t)}\right\|_F^2 + \tfrac{4L^2\kappa^2}{m}\mathbb{E}\left\|\mathbf{X}_\perp^{(t)}\right\|_F^2 + \tfrac{4L^2}{m}\mathbb{E}\left\|\widetilde{\mathbf{Y}}^{(t)} - \mathbf{Y}^{(t)}\right\|_F^2, \quad (37)$$

*where $P$ and $\mathbf{R}^{(t)}$ are defined in (9) and (28).*

### C.2 One-iteration progress inequality

Our analysis relies on establishing a one-iteration progress inequality about $\phi$ based on the updates in lines 8-10 in Algorithm 1. Its proof is deferred to Appendix E.

**Lemma C.2** *Let $\{(\mathbf{X}^{(t)}, \mathbf{\Lambda}^{(t)}, \mathbf{Y}^{(t)})\}$ be generated from Algorithm 1. For all $t \geq 0$, it holds that*

$$\phi(\bar{\mathbf{x}}^{(t+1)}, \mathbf{\Lambda}^{(t+1)}) - \phi(\bar{\mathbf{x}}^{(t)}, \mathbf{\Lambda}^{(t)})$$

$$\leq -\tfrac{1}{2m}\left(\tfrac{1}{\eta_{\mathbf{x}}} - L(\kappa + 1 + c_1) - L_P(1 + c_3)\right)\left\|\mathbf{X}^{(t+1)} - \bar{\mathbf{X}}^{(t)}\right\|_F^2$$

$$- \left(\tfrac{1}{\eta_{\mathbf{\Lambda}}} - \tfrac{L_P}{2} - Lc_2\right)\left\|\mathbf{\Lambda}^{(t+1)} - \mathbf{\Lambda}^{(t)}\right\|_F^2 - \tfrac{1}{2m\eta_{\mathbf{x}}}\left\|\mathbf{X}^{(t+1)} - \widetilde{\mathbf{X}}^{(t)}\right\|_F^2 \quad (38)$$

$$+ \tfrac{1}{2m}\left(L(\kappa + 1) + \tfrac{\kappa^2}{c_2} + \tfrac{\rho^2}{\eta_{\mathbf{x}}}\right)\left\|\mathbf{X}_\perp^{(t)}\right\|_F^2 + \tfrac{1}{2m}\left(\tfrac{L}{c_1} + \tfrac{1}{c_2}\right)\left\|\widetilde{\mathbf{Y}}^{(t)} - \mathbf{Y}^{(t)}\right\|_F^2$$

$$+ \tfrac{1}{2mL_P c_3}\sum_{i=1}^m\left\|\tfrac{1}{m}\sum_{j=1}^m \nabla_{\mathbf{x}} f_j(\mathbf{x}_j^{(t)}, \mathbf{y}_j^{(t)}) - \mathbf{v}_{\mathbf{x},i}^{(t)}\right\|_2^2,$$

*where $c_1, c_2, c_3 > 0$ are arbitrary constants, and $\phi, \widetilde{\mathbf{X}}^{(t)}$ and $\widetilde{\mathbf{Y}}^{(t)}$ are defined in (9), (17), and (27).*

From Lemma C.2, we see that the change in $\phi(\bar{\mathbf{x}}, \mathbf{\Lambda})$ is increasing in

$$\left\|\mathbf{X}_\perp^{(t)}\right\|_F^2, \quad \left\|\widetilde{\mathbf{Y}}^{(t)} - \mathbf{Y}^{(t)}\right\|_F^2, \quad \sum_{i=1}^m\left\|\tfrac{1}{m}\sum_{j=1}^m \nabla_{\mathbf{x}} f_j(\mathbf{x}_j^{(t)}, \mathbf{y}_j^{(t)}) - \mathbf{v}_{\mathbf{x},i}^{(t)}\right\|_2^2. \quad (39)$$

Hence we need to ensure these terms can be well controlled to establish convergence. The next subsection is devoted to providing upper bounds on each term in (39).

### C.3 Consensus and dual error bounds.

The proof of the following lemma can be found in Lemma C.7 of (Mancino-Ball et al., 2023a).

**Lemma C.3** *Let $\{(\mathbf{X}^{(t)}, \mathbf{V}^{(t)})\}$ be generated from Algorithm 1. For all $t \geq 0$, it holds that*

$$\left\| \mathbf{X}_{\perp}^{(t+1)} \right\|_F^2 \leq \rho \left\| \mathbf{X}_{\perp}^{(t)} \right\|_F^2 + \frac{\eta_{\mathbf{x}}^2}{1-\rho} \left\| \mathbf{V}_{\perp,\mathbf{x}}^{(t)} \right\|_F^2, \tag{40}$$

*where $\rho$ is defined in Assumption 3.*

Next, we provide an upper bound on the last term in (39). Its proof is given in Appendix F.

**Lemma C.4** *Let $\{(\mathbf{X}^{(t)}, \mathbf{Y}^{(t)}, \mathbf{V}^{(t)})\}$ be generated from Algorithm 1 and $\mathbf{R}^{(t)}$ defined in (28). Then*

$$\sum_{i=1}^m \mathbb{E} \left\| \frac{1}{m} \sum_{j=1}^m \nabla_{\mathbf{x}} f_j(\mathbf{x}_j^{(t)}, \mathbf{y}_j^{(t)}) - \mathbf{v}_{\mathbf{x},i}^{(t)} \right\|_2^2 \leq 2\mathbb{E} \left\| \mathbf{R}^{(t)} \right\|_F^2 + 2\mathbb{E} \left\| \mathbf{V}_{\perp,\mathbf{x}}^{(t)} \right\|_F^2, \forall t \geq 0. \tag{41}$$

Finally, we provide upper bounds to the dual errors. The proofs are given in Appendix F.

**Lemma C.5** *Let $\{(\mathbf{X}^{(t)}, \mathbf{\Lambda}^{(t)}, \mathbf{Y}^{(t)})\}$ be generated from Algorithm 1. Then provided $\eta_{\mathbf{y}} \leq \frac{1}{4L}$, it holds that for any $c_4, c_5 > 0$,*

$$\left\| \mathbf{Y}^{(t+1)} - \mathbf{Y}^{(t)} \right\|_F^2 \leq 4m\eta_{\mathbf{y}} \left( \hat{\delta}_t - \hat{\delta}_{t+1} \right) + 4\eta_{\mathbf{y}}^2 \left\| \mathbf{R}^{(t)} \right\|_F^2 + 4\eta_{\mathbf{y}} \left( \frac{L^2}{2c_4} + \frac{L\sqrt{m}}{c_5} \right) \left\| \widetilde{\mathbf{Y}}^{(t+1)} - \mathbf{Y}^{(t+1)} \right\|_F^2$$
$$+ 4\eta_{\mathbf{y}} \left( \frac{L_P + L}{2} + \frac{c_4}{2} \right) \left\| \mathbf{X}^{(t+1)} - \mathbf{X}^{(t)} \right\|_F^2 + 4\eta_{\mathbf{y}} \left( \frac{mL_P}{2} + \frac{c_5 L\sqrt{m}}{4} \right) \left\| \mathbf{\Lambda}^{(t+1)} - \mathbf{\Lambda}^{(t)} \right\|_F^2, \tag{42}$$

*where $\widetilde{\mathbf{Y}}^{(t)}$ is defined in (27), $\mathbf{R}^{(t)}$ is defined in (28), $L_P = L\sqrt{4\kappa^2 + 1}$, and*

$$\hat{\delta}_t := Q(\mathbf{X}^{(t)}, \mathbf{\Lambda}^{(t)}) - \left( \Gamma_t(\mathbf{Y}^{(t)}) - \frac{1}{m} \sum_{i=1}^m h(\mathbf{y}_i^{(t)}) \right), \tag{43}$$

*with $Q(\mathbf{X}, \mathbf{\Lambda})$ and $\Gamma_t(\cdot)$ defined in (8) and (29).*

**Lemma C.6** *Let $\{(\mathbf{X}^{(t)}, \mathbf{\Lambda}^{(t)}, \mathbf{Y}^{(t)})\}$ be generated from Algorithm 1 and $\mathbf{R}^{(t)}$ defined in (28). Suppose $\eta_{\mathbf{y}} \leq \frac{1}{4L}$. Then it holds that*

$$\left\| \widetilde{\mathbf{Y}}^{(t+1)} - \mathbf{Y}^{(t+1)} \right\|_F^2$$
$$\leq (1 - \eta_{\mathbf{y}}\mu) \left\| \widetilde{\mathbf{Y}}^{(t)} - \mathbf{Y}^{(t)} \right\|_F^2 + \frac{4\eta_{\mathbf{y}}}{\mu} \left\| \mathbf{R}^{(t)} \right\|_F^2 + \frac{4\kappa^2}{\eta_{\mathbf{y}}\mu} \left\| \mathbf{X}^{(t+1)} - \mathbf{X}^{(t)} \right\|_F^2 + \frac{4\kappa^2 m}{\eta_{\mathbf{y}}\mu} \left\| \mathbf{\Lambda}^{(t+1)} - \mathbf{\Lambda}^{(t)} \right\|_F^2. \tag{44}$$

### C.4 Convergence results by SPIDER-type variance reduction

In this subsection, we set VR-tag = SPIDER in Algorithm 1, and we consider both the general stochastic case and the special finite-sum setting. The proofs of all the lemmas are given in Appendix G. We first bound the consensus error of the tracked gradient and the error of the gradient estimator.

**Lemma C.7** *Let $\{(\mathbf{X}^{(t)}, \mathbf{Y}^{(t)}, \mathbf{V}^{(t)})\}$ be generated from Algorithm 1 and $\mathbf{R}^{(t)}$ defined in (28). When (2) holds, we set $\mathcal{S}_0 = \mathcal{S}_1 = n$ and take all data samples. Define $n_t \in \mathbb{Z}_+$ as the unique integer such that $n_t q \leq t < (n_t + 1)q$ for all $t \geq 0$. Then it holds that*

$$\mathbb{E} \left\| \mathbf{V}_{\perp,\mathbf{x}}^{(t+1)} \right\|_F^2 \leq \rho \mathbb{E} \left\| \mathbf{V}_{\perp,\mathbf{x}}^{(t)} \right\|_F^2 + \frac{6m\Upsilon}{1-\rho} \tag{45}$$
$$+ \frac{3L^2}{1-\rho} \mathbb{E} \left( \left\| \mathbf{Z}^{(t+1)} - \mathbf{Z}^{(t)} \right\|_F^2 + \frac{2}{\mathcal{S}_2} \sum_{r=n_t q}^t \left\| \mathbf{Z}^{(r+1)} - \mathbf{Z}^{(r)} \right\|_F^2 \right),$$

$$\mathbb{E} \left\| \mathbf{R}^{(t)} \right\|_F^2 \leq \frac{L^2}{\mathcal{S}_2} \sum_{r=n_t q}^{t-1} \mathbb{E} \left\| \mathbf{Z}^{(r+1)} - \mathbf{Z}^{(r)} \right\|_F^2 + m\Upsilon, \tag{46}$$

where $\mathbf{Z}^{(t)} = (\mathbf{X}^{(t)}, \mathbf{Y}^{(t)})$ by the notation in (4a), and

$$\Upsilon := \tfrac{\sigma^2}{\mathcal{S}_1}, \text{ for general distributions } \{\mathcal{D}_i\}; \ \Upsilon = 0, \text{ for the special finite-sum setting in (2).} \tag{47}$$

**Remark C.1** *Notice that for any $t \geq 0$, we have $n_t q \leq t \leq (n_t + 1)q - 1$. Therefore*

$$\sum_{t=0}^{T-1} \sum_{r=n_t q}^{t} \mathbb{E} \left\| \mathbf{Z}^{(r+1)} - \mathbf{Z}^{(r)} \right\|_F^2 \leq q \sum_{t=0}^{T-1} \mathbb{E} \left\| \mathbf{Z}^{(t+1)} - \mathbf{Z}^{(t)} \right\|_F^2. \tag{48}$$

*The relation in (48) is standard in the analysis of SPIDER-type methods; e.g., see (Xin et al., 2021a, Eqn. (85)).*

In the rest of this subsection, we set

$$q = \mathcal{S}_2, \ c_1 = c_2 = 32\kappa^2, \ c_3 = 60, \ c_4 = 16\kappa^2 L, \ c_5 = 32\sqrt{m}\kappa^2, \ \eta_{\mathbf{y}} = \tfrac{1}{4L}. \tag{49}$$

**Lemma C.8** *Let $\{(\mathbf{X}^{(t)}, \mathbf{\Lambda}^{(t)}, \mathbf{Y}^{(t)})\}$ be generated from Algorithm 1, $\widetilde{\mathbf{Y}}^{(t)}$ defined in (27), and $\hat{\delta}_t$ be defined in (43). Then for any integer $T \geq 1$, it holds that*

$$\sum_{t=0}^{T-1} \mathbb{E} \left\| \widetilde{\mathbf{Y}}^{(t+1)} - \mathbf{Y}^{(t+1)} \right\|_F^2 \leq 16\kappa^2(20\kappa^2 + \kappa + 2) \sum_{t=0}^{T-1} \mathbb{E} \left( \left\| \mathbf{X}^{(t+1)} - \widetilde{\mathbf{X}}^{(t)} \right\|_F^2 + 4 \left\| \mathbf{X}_\perp^{(t)} \right\|_F^2 \right) \tag{50}$$

$$+ (6\kappa - \tfrac{3}{2}) \left\| \widetilde{\mathbf{Y}}^{(0)} - \mathbf{Y}^{(0)} \right\|_F^2 + \tfrac{8m\kappa^2}{L}\hat{\delta}_0 + \tfrac{8mT\Upsilon}{\mu^2} + 8m\kappa^2(20\kappa^2 + \kappa + 1) \sum_{t=0}^{T-1} \mathbb{E} \left\| \mathbf{\Lambda}^{(t+1)} - \mathbf{\Lambda}^{(t)} \right\|_F^2$$

*and*

$$\sum_{t=0}^{T-1} \mathbb{E} \left\| \mathbf{Y}^{(t+1)} - \mathbf{Y}^{(t)} \right\|_F^2 \leq 2(24\kappa^2 + 2\kappa + 3) \sum_{t=0}^{T-1} \mathbb{E} \left( \left\| \mathbf{X}^{(t+1)} - \widetilde{\mathbf{X}}^{(t)} \right\|_F^2 + 4 \left\| \mathbf{X}_\perp^{(t)} \right\|_F^2 \right) \tag{51}$$

$$+ \tfrac{1}{2\kappa} \left\| \widetilde{\mathbf{Y}}^{(0)} - \mathbf{Y}^{(0)} \right\|_F^2 + \tfrac{2m}{L}\hat{\delta}_0 + \tfrac{mT\Upsilon}{L^2} + 2m(12\kappa^2 + \kappa + 1) \sum_{t=0}^{T-1} \mathbb{E} \left\| \mathbf{\Lambda}^{(t+1)} - \mathbf{\Lambda}^{(t)} \right\|_F^2.$$

Below we show the square summability of the generated sequence, which is crucial to obtain our convergence and complexity results.

**Theorem C.1** *Under Assumptions 1 and 2, let $\{(\mathbf{X}^{(t)}, \widetilde{\mathbf{X}}^{(t)}, \mathbf{\Lambda}^{(t)}, \mathbf{Y}^{(t)}, \mathbf{V}^{(t)})\}_{t \geq 0}$ be generated from Algorithm 1 with $\mathtt{VR\text{-}tag} == SPIDER$, $q = \mathcal{S}_2$, $\eta_{\mathbf{y}} = \tfrac{1}{4L}$, and $\eta_{\mathbf{x}}$ and $\eta_{\mathbf{\Lambda}}$ set to*

$$\eta_{\mathbf{x}} = \min \left\{ \frac{(1-\rho)^2}{180L_P}, \frac{1}{20(L+1)(12\kappa^2 + 2\kappa + 5)} \right\}, \tag{52a}$$

$$\eta_{\mathbf{\Lambda}} = \min \left\{ \frac{5L_P(1-\rho)^2}{24L^2(12\kappa^2 + \kappa + 1)}, \frac{1}{2L_P + 128L\kappa^2 + \frac{(L+1)(20\kappa^2 + \kappa + 1)}{2} + \frac{4L^2(12\kappa^2 + \kappa + 1)}{30L_P}} \right\}, \tag{52b}$$

*where $L_P = L\sqrt{4\kappa^2 + 1}$ is given in Proposition C.1. Then it holds for any $T \geq 1$,*

$$\tfrac{1}{4m\eta_{\mathbf{x}}} \sum_{t=0}^{T-1} \mathbb{E} \left\| \mathbf{X}^{(t+1)} - \bar{\mathbf{X}}^{(t)} \right\|_F^2 + \tfrac{1}{2\eta_{\mathbf{\Lambda}}} \sum_{t=0}^{T-1} \mathbb{E} \left\| \mathbf{\Lambda}^{(t+1)} - \mathbf{\Lambda}^{(t)} \right\|_F^2$$

$$+ \tfrac{1}{4m\eta_{\mathbf{x}}} \sum_{t=0}^{T-1} \mathbb{E} \left\| \mathbf{X}^{(t+1)} - \widetilde{\mathbf{X}}^{(t)} \right\|_F^2 + \tfrac{3}{4m\eta_{\mathbf{x}}} \sum_{t=0}^{T-1} \mathbb{E} \left\| \mathbf{X}_\perp^{(t)} \right\|_F^2 + \tfrac{1}{60mL_P} \sum_{t=0}^{T-1} \mathbb{E} \left\| \mathbf{V}_{\perp,\mathbf{x}}^{(t)} \right\|_F^2$$

$$\leq C_0 + T \left( \tfrac{1}{30L_P} + \tfrac{1}{L_P(1-\rho)^2} + \tfrac{L+1}{8L^2} \right) \Upsilon, \tag{53}$$

*where $\Upsilon$ is defined in (47), and*

$$C_0 := \phi(\bar{\mathbf{x}}^{(0)}, \mathbf{\Lambda}^{(0)}) - \phi^* + \left( \tfrac{1}{20mL_P} + \tfrac{\rho}{15mL_P(1-\rho)} \right) \mathbb{E} \left\| \mathbf{V}_{\perp,\mathbf{x}}^{(0)} \right\|^2$$

$$+ \left( \tfrac{6(L+1)}{64m\kappa} + \tfrac{L^2}{120mL_P\kappa} + \tfrac{3L^2}{10mL_P\kappa(1-\rho)^2} \right) \left\| \widetilde{\mathbf{Y}}^{(0)} - \mathbf{Y}^{(0)} \right\|_F^2 + \left( \tfrac{1+1/L}{8} + \tfrac{L}{30L_P} + \tfrac{6L}{5L_P(1-\rho)^2} \right) \hat{\delta}_0.$$

*Proof.* We first take the expectation of (38), apply (41), and plug in the values of $c_1, c_2$ and $c_3$ to have

$$
\begin{aligned}
\mathbb{E}\left[\phi(\bar{\mathbf{x}}^{(t+1)}, \mathbf{\Lambda}^{(t+1)}) - \phi(\bar{\mathbf{x}}^{(t)}, \mathbf{\Lambda}^{(t)})\right] \leq{}& -\tfrac{1}{4m\eta_{\mathbf{x}}}\mathbb{E}\left\|\mathbf{X}^{(t+1)} - \bar{\mathbf{X}}^{(t)}\right\|_F^2 - \tfrac{1}{2m\eta_{\mathbf{x}}}\mathbb{E}\left\|\mathbf{X}^{(t+1)} - \widetilde{\mathbf{X}}^{(t)}\right\|_F^2 \\
& -\left(\tfrac{1}{\eta_{\mathbf{\Lambda}}} - \tfrac{L_P}{2} - 32L\kappa^2\right)\mathbb{E}\left\|\mathbf{\Lambda}^{(t+1)} - \mathbf{\Lambda}^{(t)}\right\|_F^2 + \tfrac{1}{2m}\left(L(\kappa+1) + \tfrac{1}{32} + \tfrac{\rho^2}{\eta_{\mathbf{x}}}\right)\mathbb{E}\left\|\mathbf{X}_\perp^{(t)}\right\|_F^2 \\
& + \tfrac{L+1}{64m\kappa^2}\mathbb{E}\left\|\widetilde{\mathbf{Y}}^{(t)} - \mathbf{Y}^{(t)}\right\|_F^2 + \tfrac{1}{60mL_P}\mathbb{E}\left(\left\|\mathbf{R}^{(t)}\right\|_F^2 + \left\|\mathbf{V}_{\perp,\mathbf{x}}^{(t)}\right\|_F^2\right),
\end{aligned}
\tag{54}
$$

where the coefficient of $\mathbb{E}\left\|\mathbf{X}^{(t+1)} - \bar{\mathbf{X}}^{(t)}\right\|_F^2$ is obtained by the arguments

$$
\begin{aligned}
& L(\kappa + 1 + c_1) + L_P(1 + c_3) \\
&= L(\kappa + 1 + 32\kappa^2) + 61L\sqrt{4\kappa^2 + 1} \leq \tfrac{L(\kappa+1)+1+20(L+1)(5+\kappa+12\kappa^2)}{2} \leq \tfrac{1}{2\eta_{\mathbf{x}}}.
\end{aligned}
$$

Next, for any $\gamma_1 > 0$ and $\gamma_2 > 0$, we add $\gamma_1\mathbb{E}\left\|\mathbf{X}_\perp^{(t+1)}\right\|_F^2, \gamma_2\mathbb{E}\left\|\mathbf{V}_{\perp,\mathbf{x}}^{(t+1)}\right\|_F^2$ to both sides of (54) and use the results of Lemmas C.3 and C.7 to obtain

$$
\mathbb{E}\left[\phi(\bar{\mathbf{x}}^{(t+1)}, \mathbf{\Lambda}^{(t+1)}) - \phi(\bar{\mathbf{x}}^{(t)}, \mathbf{\Lambda}^{(t)})\right] + \gamma_1\mathbb{E}\left\|\mathbf{X}_\perp^{(t+1)}\right\|_F^2 + \gamma_2\mathbb{E}\left\|\mathbf{V}_{\perp,\mathbf{x}}^{(t+1)}\right\|_F^2
\tag{55}
$$

$$
\begin{aligned}
\leq{}& -\tfrac{1}{4m\eta_{\mathbf{x}}}\mathbb{E}\left\|\mathbf{X}^{(t+1)} - \bar{\mathbf{X}}^{(t)}\right\|_F^2 - \tfrac{1}{2m\eta_{\mathbf{x}}}\mathbb{E}\left\|\mathbf{X}^{(t+1)} - \widetilde{\mathbf{X}}^{(t)}\right\|_F^2 \\
& -\left(\tfrac{1}{\eta_{\mathbf{\Lambda}}} - \tfrac{L_P}{2} - 32L\kappa^2\right)\mathbb{E}\left\|\mathbf{\Lambda}^{(t+1)} - \mathbf{\Lambda}^{(t)}\right\|_F^2 \\
& + \tfrac{1}{2m}\left(L(\kappa+1) + \tfrac{1}{32} + \tfrac{\rho^2}{\eta_{\mathbf{x}}}\right)\mathbb{E}\left\|\mathbf{X}_\perp^{(t)}\right\|_F^2 + \tfrac{L+1}{64m\kappa^2}\mathbb{E}\left\|\widetilde{\mathbf{Y}}^{(t)} - \mathbf{Y}^{(t)}\right\|_F^2 + \gamma_1\rho\left\|\mathbf{X}_\perp^{(t)}\right\|_F^2 \\
& + \tfrac{1}{60mL_P}\mathbb{E}\left(\tfrac{L^2}{\mathcal{S}_2}\sum_{r=n_t q}^{t-1}\mathbb{E}\left\|\mathbf{Z}^{(r+1)} - \mathbf{Z}^{(r)}\right\|_F^2 + m\Upsilon + \left\|\mathbf{V}_{\perp,\mathbf{x}}^{(t)}\right\|_F^2\right) + \tfrac{\gamma_1\eta_{\mathbf{x}}^2}{1-\rho}\left\|\mathbf{V}_{\perp,\mathbf{x}}^{(t)}\right\|_F^2 \\
& + \gamma_2\left(\rho\mathbb{E}\left\|\mathbf{V}_{\perp,\mathbf{x}}^{(t)}\right\|_F^2 + \tfrac{6m\Upsilon}{1-\rho} + \tfrac{3L^2}{1-\rho}\mathbb{E}\left[\left\|\mathbf{Z}^{(t+1)} - \mathbf{Z}^{(t)}\right\|_F^2 + \tfrac{2}{\mathcal{S}_2}\sum_{r=n_t q}^t\left\|\mathbf{Z}^{(r+1)} - \mathbf{Z}^{(r)}\right\|_F^2\right]\right).
\end{aligned}
$$

Sum up (55) over $t = 0$ to $T - 1$, utilize (48), and recall $\mathcal{S}_2 = q$ to have

$$
\mathbb{E}\left[\phi(\bar{\mathbf{x}}^{(T)}, \mathbf{\Lambda}^{(T)}) - \phi(\bar{\mathbf{x}}^{(0)}, \mathbf{\Lambda}^{(0)})\right] + \sum_{t=0}^{T-1}\mathbb{E}\left(\gamma_1\left\|\mathbf{X}_\perp^{(t+1)}\right\|_F^2 + \gamma_2\left\|\mathbf{V}_{\perp,\mathbf{x}}^{(t+1)}\right\|_F^2\right)
$$

$$
\begin{aligned}
\leq{}& \sum_{t=0}^{T-1}\mathbb{E}\left[-\tfrac{1}{4m\eta_{\mathbf{x}}}\mathbb{E}\left\|\mathbf{X}^{(t+1)} - \bar{\mathbf{X}}^{(t)}\right\|_F^2 - \tfrac{1}{2m\eta_{\mathbf{x}}}\mathbb{E}\left\|\mathbf{X}^{(t+1)} - \widetilde{\mathbf{X}}^{(t)}\right\|_F^2 \right. \\
& \left. -\left(\tfrac{1}{\eta_{\mathbf{\Lambda}}} - \tfrac{L_P}{2} - 32L\kappa^2\right)\mathbb{E}\left\|\mathbf{\Lambda}^{(t+1)} - \mathbf{\Lambda}^{(t)}\right\|_F^2 + \tfrac{L+1}{64m\kappa^2}\mathbb{E}\left\|\widetilde{\mathbf{Y}}^{(t)} - \mathbf{Y}^{(t)}\right\|_F^2\right] \\
& + \tfrac{1}{2m}\left(L(\kappa+1) + \tfrac{1}{32} + \tfrac{\rho^2}{\eta_{\mathbf{x}}} + 2m\gamma_1\rho\right)\sum_{t=0}^{T-1}\mathbb{E}\left\|\mathbf{X}_\perp^{(t)}\right\|_F^2 \\
& + \left(\tfrac{L^2}{60mL_P} + \tfrac{9\gamma_2 L^2}{1-\rho}\right)\sum_{t=0}^{T-1}\mathbb{E}\left\|\mathbf{Z}^{(t+1)} - \mathbf{Z}^{(t)}\right\|_F^2 \\
& + \left(\tfrac{1}{60mL_P} + \tfrac{\gamma_1\eta_{\mathbf{x}}^2}{1-\rho} + \gamma_2\rho\right)\sum_{t=0}^{T-1}\mathbb{E}\left\|\mathbf{V}_{\perp,\mathbf{x}}^{(t)}\right\|_F^2 + T\left(\tfrac{1}{60L_P} + \tfrac{6m\gamma_2}{1-\rho}\right)\Upsilon.
\end{aligned}
\tag{56}
$$

Now plug (50) and (51) into (56) and also bound $\|\mathbf{X}^{(t+1)} - \mathbf{X}^{(t)}\|_F^2$ by (31) to have

$$
\mathbb{E}\left[\phi(\bar{\mathbf{x}}^{(T)}, \boldsymbol{\Lambda}^{(T)}) - \phi(\bar{\mathbf{x}}^{(0)}, \boldsymbol{\Lambda}^{(0)})\right]
$$

$$
+ \sum_{t=0}^{T-1} \mathbb{E}\left(\gamma_1 \left\|\mathbf{X}_\perp^{(t+1)}\right\|_F^2 + \gamma_2 \left\|\mathbf{V}_{\perp,\mathbf{x}}^{(t+1)}\right\|_F^2 + \tfrac{1}{4m\eta_\mathbf{x}}\mathbb{E}\left\|\mathbf{X}^{(t+1)} - \bar{\mathbf{X}}^{(t)}\right\|_F^2\right) \tag{57}
$$

$$
\leq -\sum_{t=0}^{T-1}\mathbb{E}\left[\tfrac{1}{2m}\left(\tfrac{1}{\eta_\mathbf{x}} - \tfrac{(L+1)(20\kappa^2+\kappa+2)}{2} - 2(24\kappa^2+2\kappa+4)\left(\tfrac{L^2}{30L_P} + \tfrac{18m\gamma_2 L^2}{1-\rho}\right)\right)\mathbb{E}\left\|\mathbf{X}^{(t+1)} - \widetilde{\mathbf{X}}^{(t)}\right\|_F^2\right.
$$

$$
+ \left.\left(\tfrac{1}{\eta_{\boldsymbol{\Lambda}}} - \tfrac{L_P}{2} - 32L\kappa^2 - \tfrac{(L+1)(20\kappa^2+\kappa+1)}{8} - (12\kappa^2+\kappa+1)\left(\tfrac{L^2}{30L_P} + \tfrac{18m\gamma_2 L^2}{1-\rho}\right)\right)\mathbb{E}\left\|\boldsymbol{\Lambda}^{(t+1)} - \boldsymbol{\Lambda}^{(t)}\right\|_F^2\right]
$$

$$
+ \tfrac{1}{2m}\left(L(\kappa+1) + \tfrac{1}{32} + \tfrac{\rho^2}{\eta_\mathbf{x}} + 2m\gamma_1\rho + 2(L+1)(20\kappa^2+\kappa+2)\right.
$$

$$
\left. + 8(24\kappa^2+2\kappa+4)\left(\tfrac{L^2}{30L_P} + \tfrac{18m\gamma_2 L^2}{1-\rho}\right)\right)\sum_{t=0}^{T-1}\mathbb{E}\left\|\mathbf{X}_\perp^{(t)}\right\|_F^2
$$

$$
+ \left(\tfrac{1}{60mL_P} + \tfrac{\gamma_1\eta_\mathbf{x}^2}{1-\rho} + \gamma_2\rho\right)\sum_{t=0}^{T-1}\mathbb{E}\left\|\mathbf{V}_{\perp,\mathbf{x}}^{(t)}\right\|_F^2 + T\left(\tfrac{1}{30L_P} + \tfrac{15m\gamma_2}{1-\rho} + \tfrac{L+1}{8L^2}\right)\Upsilon.
$$

$$
+ \left(\tfrac{6(L+1)}{64m\kappa} + \tfrac{L^2}{120mL_P\kappa} + \tfrac{9\gamma_2 L^2}{2\kappa(1-\rho)}\right)\left\|\widetilde{\mathbf{Y}}^{(0)} - \mathbf{Y}^{(0)}\right\|_F^2 + \left(\tfrac{L+1}{8m} + \tfrac{L^2}{30mL_P} + \tfrac{18\gamma_2 L^2}{1-\rho}\right)\tfrac{m}{L}\hat{\delta}_0.
$$

Set $\gamma_1$ and $\gamma_2$ to

$$
\gamma_1 = \tfrac{3}{2m(1-\rho)}\left(L(\kappa+1) + \tfrac{1}{32} + \tfrac{1}{\eta_\mathbf{x}} + 2(L+1)(20\kappa^2+\kappa+2)\right.
$$

$$
\left. + 8(24\kappa^2+2\kappa+4)\left(\tfrac{L^2}{30L_P} + \tfrac{3L^2}{5L_P(1-\rho)^2}\right)\right), \tag{58}
$$

$$
\gamma_2 = \tfrac{2}{1-\rho}\left(\tfrac{1}{60mL_P} + \tfrac{\gamma_1\eta_\mathbf{x}^2}{1-\rho}\right). \tag{59}
$$

By $\eta_\mathbf{x} \leq \tfrac{(1-\rho)^2}{180L_P}$ and $L_P = L\sqrt{4\kappa^2+1}$, it is straightforward to have $\tfrac{144\eta_\mathbf{x}^2 L^2(24\kappa^2+2\kappa+4)}{(1-\rho)^3} \leq \tfrac{1-\rho}{6}$. Then we have from (58) and (59) that

$$
\tfrac{1}{2m}\left(L(\kappa+1) + \tfrac{1}{32} + \tfrac{\rho^2}{\eta_\mathbf{x}} + 2(L+1)(20\kappa^2+\kappa+2) + 8(24\kappa^2+2\kappa+4)\left(\tfrac{L^2}{30L_P} + \tfrac{18m\gamma_2 L^2}{1-\rho}\right)\right)
$$

$$
\leq \tfrac{1}{2m}\left(L(\kappa+1) + \tfrac{1}{32} + \tfrac{1}{\eta_\mathbf{x}} + 2(L+1)(20\kappa^2+\kappa+2) + 8(24\kappa^2+2\kappa+4)\left(\tfrac{L^2}{30L_P} + \tfrac{3L^2}{10L_P(1-\rho)^2}\right)\right.
$$

$$
\left. + 8(24\kappa^2+2\kappa+4)\tfrac{36m\gamma_1\eta_x^2 L^2}{(1-\rho)^3}\right) \leq \tfrac{(1-\rho)\gamma_1}{3} + \tfrac{(1-\rho)\gamma_1}{6} = \tfrac{(1-\rho)\gamma_1}{2}. \tag{60}
$$

In addition, by the choice of $\gamma_2$ in (59), it follows

$$
\gamma_2 - \left(\tfrac{1}{60mL_P} + \tfrac{\gamma_1\eta_\mathbf{x}^2}{1-\rho} + \gamma_2\rho\right) \geq \tfrac{(1-\rho)\gamma_2}{2}. \tag{61}
$$

By the choice of $\gamma_1$ and $\eta_\mathbf{x} \leq \tfrac{(1-\rho)^2}{180L_P}$, it follows that

$$
\tfrac{m\gamma_1\eta_\mathbf{x}^2}{1-\rho} \leq \tfrac{\eta_\mathbf{x}}{120L_P}\left(L(\kappa+1) + \tfrac{1}{32} + \tfrac{1}{\eta_\mathbf{x}} + 2(L+1)(20\kappa^2+\kappa+2)\right.
$$

$$
\left. + 8(24\kappa^2+2\kappa+4)\left(\tfrac{L^2}{30L_P} + \tfrac{3L^2}{5L_P(1-\rho)^2}\right)\right). \tag{62}
$$

Also, $\tfrac{\eta_\mathbf{x}\cdot 8(24\kappa^2+2\kappa+4)\cdot 3L^2}{5L_P(1-\rho)^2} \leq \tfrac{8(24\kappa^2+2\kappa+4)\cdot 3L^2}{5\cdot 180L_P^2} = \tfrac{8(24\kappa^2+2\kappa+4)\cdot 3L^2}{5\cdot 180L^2(4\kappa^2+1)} \leq \tfrac{1}{5}$, and by $L_P \geq 2L\kappa$, it holds

$$
L(\kappa+1) + \tfrac{1}{32} + 2(L+1)(20\kappa^2+\kappa+2) + \tfrac{8(24\kappa^2+2\kappa+4)L^2}{30L_P}
$$

$$
\leq L(\kappa+1) + 1 + 2(L+1)(20\kappa^2+\kappa+2) + \tfrac{8(24\kappa^2+2\kappa+4)L^2}{60L\kappa} \leq 16(L+1)(12\kappa^2+2\kappa+5).
$$

Hence from (62), we have $\frac{\gamma_1 \eta_{\mathbf{x}}^2}{1-\rho} \leq \frac{1}{60mL_P}$. Thus $m\gamma_2 \leq \frac{1}{15L_P(1-\rho)}$ follows from (59), and we have

$$\frac{1}{\eta_{\mathbf{x}}} - \frac{(L+1)(20\kappa^2+\kappa+2)}{2} - 2(24\kappa^2+2\kappa+4)\left(\frac{L^2}{30L_P} + \frac{18m\gamma_2 L^2}{1-\rho}\right)$$

$$\geq \frac{1}{\eta_{\mathbf{x}}} - \frac{(L+1)(20\kappa^2+\kappa+2)}{2} - \frac{2L^2(24\kappa^2+2\kappa+4)}{30L_P}\left(1 + \frac{36}{(1-\rho)^2}\right) \geq \frac{1}{2\eta_{\mathbf{x}}}, \tag{63}$$

where the last inequality can be verified by plugging the value of $\eta_{\mathbf{x}}$ given in (52a). Similarly, by the choice of $\eta_{\mathbf{\Lambda}}$ in (52b), it is straightforward to have

$$\frac{1}{\eta_{\mathbf{\Lambda}}} - \frac{L_P}{2} - 32L\kappa^2 - \frac{(L+1)(20\kappa^2+\kappa+1)}{8} - (12\kappa^2+\kappa+1)\left(\frac{L^2}{30L_P} + \frac{18m\gamma_2 L^2}{1-\rho}\right) \geq \frac{1}{2\eta_{\mathbf{\Lambda}}}. \tag{64}$$

Moreover, by $m\gamma_2 \leq \frac{1}{15L_P(1-\rho)}$, it holds

$$\left(\frac{L+1}{8m} + \frac{L^2}{30mL_P} + \frac{18\gamma_2 L^2}{1-\rho}\right)\frac{m}{L}\hat{\delta}_0 \leq \left(\frac{1+1/L}{8} + \frac{L}{30L_P} + \frac{6L}{5L_P(1-\rho)^2}\right)\hat{\delta}_0. \tag{65}$$

Therefore, we obtain (53) from (57) by using (60)-(65), the lower bounds $\frac{(1-\rho)\gamma_1}{2} \geq \frac{3}{4m\eta_{\mathbf{x}}}$ and $\frac{(1-\rho)\gamma_2}{2} \geq \frac{1}{60mL_P}$, the upper bounds $\frac{\gamma_1 \eta_{\mathbf{x}}^2}{1-\rho} \leq \frac{1}{60mL_P}$ and $\gamma_2 \leq \frac{1}{15mL_P(1-\rho)}$, and $\phi(\bar{\mathbf{x}}^{(T)}, \mathbf{\Lambda}^{(T)}) \geq \phi^*$. $\qquad \square$

By Theorem C.1, we first show a last-iterate convergence in probability for the finite-sum case.

**Theorem C.2 (Convergence in probability for finite-sum case)** *Under Assumptions 1,2, and 3, let $\{(\mathbf{X}^{(t)}, \mathbf{\Lambda}^{(t)}, \mathbf{Y}^{(t)})\}_{t\geq 0}$ be generated from Algorithm 1 with VR-tag = SPIDER and $\eta_{\mathbf{x}}, \eta_{\mathbf{\Lambda}}, \eta_{\mathbf{y}}$ chosen as in (52) and (49). If (2) holds and $\mathcal{S}_0 = \mathcal{S}_1 = n$ in Algorithm 1, then for any $\varepsilon > 0$,*

$$\lim_{t\to\infty} \mathrm{Prob}\left\{\left\|\frac{1}{\eta_{\mathbf{x}}}\left(\bar{\mathbf{x}}^{(t)} - \mathbf{prox}_{\eta_{\mathbf{x}}g}(\bar{\mathbf{x}}^{(t)} - \eta_{\mathbf{x}}\nabla_{\mathbf{x}}P(\bar{\mathbf{x}}^{(t)}, \mathbf{\Lambda}^{(t)}))\right)\right\|_2^2 + \frac{L^2}{m}\left\|\mathbf{X}_\perp^{(t)}\right\|_F^2 \geq \varepsilon\right\} = 0, \tag{66a}$$

$$\lim_{t\to\infty} \mathrm{Prob}\left\{\left\|\nabla_{\mathbf{\Lambda}}P(\bar{\mathbf{x}}^{(t)}, \mathbf{\Lambda}^{(t)})\right\|_F^2 \geq \varepsilon\right\} = 0. \tag{66b}$$

*Proof.* Recall that $\Upsilon = 0$ when (2) holds and $\mathcal{S}_0 = \mathcal{S}_1 = n$. Hence, Theorem C.1 indicates

$$\sum_{t=0}^{\infty} \mathbb{E}\left[\left\|\mathbf{X}^{(t+1)} - \bar{\mathbf{X}}^{(t)}\right\|_F^2 + \left\|\mathbf{\Lambda}^{(t+1)} - \mathbf{\Lambda}^{(t)}\right\|_F^2 + \left\|\mathbf{X}^{(t+1)} - \widetilde{\mathbf{X}}^{(t)}\right\|_F^2 + \left\|\mathbf{X}_\perp^{(t)}\right\|_F^2 + \left\|\mathbf{V}_{\perp,\mathbf{x}}^{(t)}\right\|_F^2\right] < \infty,$$

which together with (50) further implies $\sum_{t=0}^{\infty} \mathbb{E}\left\|\widetilde{\mathbf{Y}}^{(t+1)} - \mathbf{Y}^{(t+1)}\right\|_F^2 < \infty$. Therefore, each of the terms $\left\|\mathbf{X}^{(t+1)} - \bar{\mathbf{X}}^{(t)}\right\|_F^2$, $\left\|\mathbf{\Lambda}^{(t+1)} - \mathbf{\Lambda}^{(t)}\right\|_F^2$, $\left\|\mathbf{X}^{(t+1)} - \widetilde{\mathbf{X}}^{(t)}\right\|_F^2$, $\left\|\mathbf{X}_\perp^{(t)}\right\|_F^2$, $\left\|\mathbf{V}_{\perp,\mathbf{x}}^{(t)}\right\|_F^2$, and $\left\|\widetilde{\mathbf{Y}}^{(t+1)} - \mathbf{Y}^{(t+1)}\right\|_F^2$ approaches 0 in expectation as $t \to \infty$. Now the desired results follow immediately from Lemma C.1 and the Markov inequality. $\qquad \square$

**Remark C.2** *In order to show the last-iterate convergence in probability for the general stochastic case, we need to increase $\mathcal{S}_1$ periodically such that the cumulated variance is finite. This requires us to make $\mathcal{S}_1$ dependent on the number of periods, which will cause confusion on the notation. We do not extend the analysis here. In addition, Theorem C.2 together with Remark 2.1 implies that $\{\mathbf{X}^{(t)}\}_{t\geq 0}$ satisfies the convergence in probability for (3) when (2) holds and $\mathcal{S}_0 = \mathcal{S}_1 = n$ in Algorithm 1, i.e.,*

$$\lim_{t\to\infty} \mathrm{Prob}\left\{\left\|\frac{1}{\eta_{\mathbf{x}}}\left(\bar{\mathbf{x}}^{(t)} - \mathbf{prox}_{\eta_{\mathbf{x}}g}(\bar{\mathbf{x}}^{(t)} - \eta_{\mathbf{x}}\nabla p(\bar{\mathbf{x}}^{(t)}))\right)\right\|_2^2 + \frac{L^2}{m}\left\|\mathbf{X}_\perp^{(t)}\right\|_F^2 \geq \varepsilon\right\} = 0,$$

*where $p(\cdot)$ is defined in (9).*

Moreover, by Theorem C.1, we are ready to prove the expected convergence rate result, stated in Theorem 4.1, and furthermore the complexity result, stated in Corollary 4.1.

**Proof of Theorem 4.1** By the selection of $\tau$, we have $\mathbb{E}\left\|\widetilde{\mathbf{Y}}^{(\tau)} - \mathbf{Y}^{(\tau)}\right\|_F^2 = \frac{1}{T}\sum_{t=0}^{T-1}\mathbb{E}\left\|\widetilde{\mathbf{Y}}^{(t)} - \mathbf{Y}^{(t)}\right\|_F^2$.
Hence, it follows from (50) and (53) that

$$
\begin{aligned}
\mathbb{E}\left\|\widetilde{\mathbf{Y}}^{(\tau)} - \mathbf{Y}^{(\tau)}\right\|_F^2 &\leq \frac{6\kappa}{T}\left\|\widetilde{\mathbf{Y}}^{(0)} - \mathbf{Y}^{(0)}\right\|_F^2 + \frac{8m\Upsilon}{\mu^2} + \frac{8m\kappa^2\hat{\delta}_0}{TL} \\
&+ \left(\frac{C_0}{T} + \left(\frac{1}{30L_P} + \frac{1}{L_P(1-\rho)^2} + \frac{L+1}{8L^2}\right)\Upsilon\right)\cdot\left(150m\eta_{\mathbf{x}}\kappa^2(20\kappa^2 + \kappa + 2) + 16m\eta_{\mathbf{\Lambda}}\kappa^2(20\kappa^2 + \kappa + 1)\right).
\end{aligned}
\tag{67}
$$

Similarly, we have from (51) and (53) that

$$
\begin{aligned}
\mathbb{E}\left\|\mathbf{Y}^{(\tau+1)} - \mathbf{Y}^{(\tau)}\right\|_F^2 &= \frac{1}{T}\sum_{t=0}^{T-1}\mathbb{E}\left\|\mathbf{Y}^{(t+1)} - \mathbf{Y}^{(t)}\right\|_F^2 \leq \frac{2m}{TL}\hat{\delta}_0 + \frac{1}{2\kappa T}\left\|\widetilde{\mathbf{Y}}^{(0)} - \mathbf{Y}^{(0)}\right\|_F^2 + \frac{m\Upsilon}{L^2} \\
&+ \left(\frac{C_0}{T} + \left(\frac{1}{30L_P} + \frac{1}{L_P(1-\rho)^2} + \frac{L+1}{8L^2}\right)\Upsilon\right)\cdot\left(20m\eta_{\mathbf{x}}(24\kappa^2 + 2\kappa + 3) + 4m\eta_{\mathbf{\Lambda}}(12\kappa^2 + \kappa + 1)\right).
\end{aligned}
\tag{68}
$$

In addition, summing up (46) over $t = 0, \ldots, T-1$ and using (48) gives

$$
\sum_{t=0}^{T-1}\mathbb{E}\left\|\mathbf{R}^{(t)}\right\|_F^2 \leq L^2\sum_{t=0}^{T-1}\left(\mathbb{E}\left\|\mathbf{X}^{(t+1)} - \mathbf{X}^{(t)}\right\|_F^2 + \mathbb{E}\left\|\mathbf{Y}^{(t+1)} - \mathbf{Y}^{(t)}\right\|_F^2\right) + Tm\Upsilon.
\tag{69}
$$

Hence, by the choice of $\tau$ and (31), we obtain

$$
\begin{aligned}
\mathbb{E}\left\|\mathbf{R}^{(\tau)}\right\|_F^2 &\leq \frac{L^2}{T}\sum_{t=0}^{T-1}\mathbb{E}\left(2\left\|\mathbf{X}^{(t+1)} - \widetilde{\mathbf{X}}^{(t)}\right\|_F^2 + 8\left\|\mathbf{X}_\perp^{(t)}\right\|_F^2 + \left\|\mathbf{Y}^{(t+1)} - \mathbf{Y}^{(t)}\right\|_F^2\right) + m\Upsilon \\
&\stackrel{(53),(68)}{\leq} \frac{2mL}{T}\hat{\delta}_0 + \frac{L^2}{2\kappa T}\left\|\widetilde{\mathbf{Y}}^{(0)} - \mathbf{Y}^{(0)}\right\|_F^2 + 2m\Upsilon \\
&+ L^2\left(\frac{C_0}{T} + \left(\frac{1}{30L_P} + \frac{1}{L_P(1-\rho)^2} + \frac{L+1}{8L^2}\right)\Upsilon\right)\cdot\left(20m\eta_{\mathbf{x}}(24\kappa^2 + 2\kappa + 4) + 4m\eta_{\mathbf{\Lambda}}(12\kappa^2 + \kappa + 1)\right).
\end{aligned}
\tag{70}
$$

Therefore, plugging (67) and (70) into (36) and (37) with $t = \tau$, we obtain the desired results from (53) and by combining like terms. $\qquad\square$

**Proof of Corollary 4.1** By the choice of initialization and the definition of $C_0$ from Theorem C.1, we have

$$
\frac{L^2(6\kappa+3)}{mT}\left\|\widetilde{\mathbf{Y}}^{(0)} - \mathbf{Y}^{(0)}\right\|_F^2 = \mathcal{O}\left(\frac{L\kappa^2}{T}\right), \quad C_0 = \mathcal{O}\left(\phi(\bar{\mathbf{x}}^{(0)}, \mathbf{\Lambda}^{(0)}) - \phi^* + \frac{L\hat{\delta}_0}{L_P(1-\rho)^2} + 1\right).
$$

Additionally by (52) and $L_P = L\sqrt{4\kappa^2 + 1}$, the stepsizes in (52) are $\eta_{\mathbf{x}} = \Theta\left(\frac{\min\{1,\kappa(1-\rho)^2\}}{L\kappa^2}\right)$ and $\eta_{\mathbf{\Lambda}} = \Theta\left(\frac{\min\{1,\kappa(1-\rho)^2\}}{\kappa^2 L}\right)$. Thus we have from (18) that

$$
\begin{aligned}
&\mathbb{E}\left\|\frac{1}{\eta_{\mathbf{x}}}\left(\bar{\mathbf{x}}^{(\tau)} - \mathbf{prox}_{\eta_{\mathbf{x}}g}\left(\bar{\mathbf{x}}^{(\tau)} - \eta_{\mathbf{x}}\nabla_{\mathbf{x}}P(\bar{\mathbf{x}}^{(\tau)}, \mathbf{\Lambda}^{(\tau)})\right)\right)\right\|_2^2 + \frac{L^2}{m}\mathbb{E}\left\|\mathbf{X}_\perp^{(\tau)}\right\|_F^2 \\
&= \mathcal{O}\left(\kappa^2\Upsilon + \frac{L\kappa^2\hat{\delta}_0}{T} + \left(\frac{1}{T}\left(\phi(\bar{\mathbf{x}}^{(0)}, \mathbf{\Lambda}^{(0)}) - \phi^* + \frac{L\hat{\delta}_0}{L_P(1-\rho)^2} + 1\right) + \frac{\Upsilon}{L\cdot\min\{1,\kappa(1-\rho)^2\}}\right)\cdot\frac{L\kappa^2}{\min\{1,\kappa(1-\rho)^2\}}\right) \\
&= \mathcal{O}\left(\left(\frac{1}{T}\left(\phi(\bar{\mathbf{x}}^{(0)}, \mathbf{\Lambda}^{(0)}) - \phi^* + \frac{\hat{\delta}_0}{\min\{1,\kappa(1-\rho)^2\}} + 1\right) + \frac{\Upsilon}{L\cdot\min\{1,\kappa(1-\rho)^2\}}\right)\cdot\frac{L\kappa^2}{\min\{1,\kappa(1-\rho)^2\}}\right)
\end{aligned}
$$

and similarly

$$
\begin{aligned}
&\mathbb{E}\left\|\nabla_{\mathbf{\Lambda}}P(\bar{\mathbf{x}}^{(\tau)}, \mathbf{\Lambda}^{(\tau)})\right\|_F^2 \\
&= \mathcal{O}\left(\left(\frac{1}{T}\left(\phi(\bar{\mathbf{x}}^{(0)}, \mathbf{\Lambda}^{(0)}) - \phi^* + \frac{\hat{\delta}_0}{\min\{1,\kappa(1-\rho)^2\}} + 1\right) + \frac{\Upsilon}{L\cdot\min\{1,\kappa(1-\rho)^2\}}\right)\cdot\frac{L\kappa^2}{\min\{1,\kappa(1-\rho)^2\}}\right).
\end{aligned}
$$

Thus for the finite-sum setting, since $\Upsilon = 0$, we can produce an $\varepsilon$-stationary point as defined in Definition 2.1 by $T$ iterations, with

$$
T = \Theta\left(\frac{L\kappa^2}{\varepsilon^2\cdot\min\{1,\kappa(1-\rho)^2\}}\left(\phi(\bar{\mathbf{x}}^{(0)}, \mathbf{\Lambda}^{(0)}) - \phi^* + \frac{\hat{\delta}_0}{\min\{1,\kappa(1-\rho)^2\}} + 1\right)\right).
\tag{71}
$$

For the general case, $\Upsilon = \frac{\sigma^2}{\mathcal{S}_1}$, and an $\varepsilon$-stationary point can be produced with $\mathcal{S}_1 = \Theta\left(\frac{\sigma^2}{\varepsilon^2} \cdot \max\{\kappa^2, (1-\rho)^{-4}\}\right)$ and $T$ in the same order as that in (71). For both general case and finite-sum case, we set $\mathcal{S}_2 = q = \lceil\sqrt{\mathcal{S}_1}\rceil$. Noticing the total number of communication is $T_c = T$ and the total number of sample gradients $T_s = (T - \lfloor\frac{T}{q}\rfloor)\mathcal{S}_2 + \lceil\frac{T}{q}\rceil\mathcal{S}_1$, we complete the proof. $\qquad\square$

### C.5 Convergence results by STORM-type variance reduction

In this subsection, we analyze Algorithm 1 with VR-tag = STORM . The general proof structure mimics that of Section C.4. The proofs of all lemmas are given in Appendix H.

**Lemma C.9** *Let $\{(\mathbf{X}^{(t)}, \mathbf{Y}^{(t)}, \mathbf{V}^{(t)})\}$ be generated from Algorithm 1 and $\mathbf{R}^{(t)}$ defined in (28). Then*

$$\mathbb{E}\left\|\mathbf{V}_{\perp,\mathbf{x}}^{(t+1)}\right\|_F^2 \leq \rho\mathbb{E}\left\|\mathbf{V}_{\perp,\mathbf{x}}^{(t)}\right\|_F^2 + \frac{1}{1-\rho}\left(3L^2\mathbb{E}\left\|\mathbf{Z}^{(t+1)} - \mathbf{Z}^{(t)}\right\|_F^2 + 3\beta^2\mathbb{E}\left\|\mathbf{R}^{(t)}\right\|_F^2 + 3m\beta^2\Upsilon_{t+1}\right), \quad (72)$$

$$\mathbb{E}\left\|\mathbf{R}^{(t+1)}\right\|_F^2 \leq 2(1-\beta)^2 L^2\mathbb{E}\left\|\mathbf{Z}^{(t+1)} - \mathbf{Z}^{(t)}\right\|_F^2 + (1-\beta)^2\mathbb{E}\left\|\mathbf{R}^{(t)}\right\|_F^2 + 2m\beta^2\Upsilon_{t+1}, \quad (73)$$

*where $\mathbf{Z}^{(t)} = (\mathbf{X}^{(t)}, \mathbf{Y}^{(t)})$ by the notation in (4a) and $\Upsilon_t := \frac{\sigma^2}{\mathcal{S}_t}$ for any $t \geq 0$.*

In the rest of this subsection, we set

$$c_1 = c_2 = \frac{32\kappa^2}{\sqrt{\beta}}, \ c_3 = \frac{60}{\sqrt{\beta}}, \ c_4 = \frac{30\sqrt{2}\kappa^2 L}{\sqrt{\beta}}, \ c_5 = \frac{60\sqrt{2m}\kappa^2}{\sqrt{\beta}}, \ \eta_{\mathbf{y}} = \frac{\sqrt{\beta}}{4\sqrt{2}L}. \quad (74)$$

With Lemma C.9, we can use (38) to show a result similar to Theorem C.1.

**Lemma C.10** *Under Assumptions 1 and 2, let $\{(\mathbf{X}^{(t)}, \widetilde{\mathbf{X}}^{(t)}, \mathbf{\Lambda}^{(t)}, \mathbf{Y}^{(t)}, \mathbf{V}^{(t)})\}_{t\geq 0}$ be generated from Algorithm 1 with VR-tag = STORM, $\beta \in (0,1)$, $\eta_{\mathbf{y}} = \frac{\sqrt{\beta}}{4\sqrt{2}L}$, and $\eta_{\mathbf{x}}$ and $\eta_{\mathbf{\Lambda}}$ set to*

$$\eta_{\mathbf{x}} = \min\left\{\frac{\kappa(1-\rho)^2}{40L(24\kappa^2 + 8\kappa + 5)}, \ \frac{\sqrt{\beta}}{48(L+1)(24\kappa^2 + 7\kappa + 4)}\right\}, \quad (75a)$$

$$\eta_{\mathbf{\Lambda}} = \min\left\{\frac{(1-\rho)^2}{4L(20\kappa + 3)}, \ \frac{\sqrt{\beta}}{4(L+1)(52\kappa^2 + \kappa + 1)}\right\}. \quad (75b)$$

*Then it holds for any integer $T \geq 1$,*

$$\frac{1}{4m\eta_{\mathbf{x}}}\sum_{t=0}^{T-1}\mathbb{E}\left\|\mathbf{X}^{(t+1)} - \bar{\mathbf{X}}^{(t)}\right\|_F^2 + \frac{1}{6m\eta_{\mathbf{x}}}\sum_{t=0}^{T-1}\mathbb{E}\left\|\mathbf{X}_\perp^{(t)}\right\|_F^2 + \frac{1}{2\eta_{\mathbf{\Lambda}}}\sum_{t=0}^{T-1}\mathbb{E}\left\|\mathbf{\Lambda}^{(t+1)} - \mathbf{\Lambda}^{(t)}\right\|_F^2$$

$$+ \frac{\sqrt{\beta}(L+1)}{160m\kappa^2}\sum_{t=0}^{T-1}\mathbb{E}\left\|\widetilde{\mathbf{Y}}^{(t)} - \mathbf{Y}^{(t)}\right\|_F^2 + \frac{\eta_{\mathbf{x}}}{m(1-\rho)^2}\sum_{t=0}^{T-1}\mathbb{E}\left\|\mathbf{V}_{\perp,\mathbf{x}}^{(t)}\right\|_F^2 + \frac{\sqrt{\beta}}{16mL}\sum_{t=0}^{T-1}\mathbb{E}\left\|\mathbf{R}^{(t)}\right\|_F^2 \quad (76)$$

$$\leq C_0 + \left(\frac{1}{(1-\rho)^2\kappa L} + \frac{(1+1/L)}{\sqrt{\beta}L}\right)\beta^2\sum_{t=0}^{T-1}\Upsilon_{t+1},$$

*where $\mathbf{R}^{(t)}$ is defined in (28), $\Upsilon_t$ is defined in Lemma C.9, and*

$$C_0 := \phi(\bar{\mathbf{x}}^{(0)}, \mathbf{\Lambda}^{(0)}) - \phi^* + \frac{1}{40m(1-\rho)\kappa L}\left\|\mathbf{V}_{\perp,\mathbf{x}}^{(0)}\right\|_F^2 + \left(\frac{(1+1/L)}{2\sqrt{\beta}mL} + \frac{1}{4m\kappa L(1-\rho)^2}\right)\left\|\mathbf{R}^{(0)}\right\|_F^2$$

$$+ \left(\frac{L+1}{5m\kappa} + \frac{\sqrt{\beta}L}{10m\kappa^2(1-\rho)^2}\right)\left\|\widetilde{\mathbf{Y}}^{(0)} - \mathbf{Y}^{(0)}\right\|_F^2 + \frac{\sqrt{\beta}}{\sqrt{2}}\left(\frac{1}{(1-\rho)^2\kappa} + \frac{(1+1/L)}{\sqrt{\beta}}\right)\hat{\delta}_0. \quad (77)$$

By Lemma C.10, we are ready to prove the convergence rate and complexity results in Theorem 4.2 and Corollary 4.2.

**Proof of Theorem 4.2** The results in Theorem 4.2 follow directly from taking $\frac{1}{T}$ times of (76) and utilizing (36) and (37) with $t = \tau$. $\qquad\square$

**Proof of Corollary 4.2** By the choice of $\beta$ in (22) and the upper bound of $\varepsilon \leq \sigma(1 - \rho)^2$, it holds that $\sqrt{\beta} \leq (1 - \rho)^2$. Thus a straight-forward comparison of the two fractions in (75a) and (75b) yields

$$\eta_{\mathbf{x}} = \Theta\left(\frac{\sqrt{\beta}}{\kappa^2 L}\right) = \Theta\left(\frac{\varepsilon}{\sigma \kappa^3 L}\right), \quad \eta_{\mathbf{\Lambda}} = \Theta\left(\frac{\varepsilon}{\sigma \kappa^3 L}\right).$$

Next we upper bound the right-hand side of (20). First, by the initialization assumption and $\sqrt{\beta} \leq (1 - \rho)^2$, we have

$$\frac{1}{40m(1-\rho)\kappa L}\mathbb{E}\left\|\mathbf{V}_{\perp,\mathbf{x}}^{(0)}\right\|_F^2 \leq 1, \quad \left(\frac{L+1}{5m\kappa} + \frac{\sqrt{\beta}L}{10m\kappa^2(1-\rho)^2}\right)\left\|\widetilde{\mathbf{Y}}^{(0)} - \mathbf{Y}^{(0)}\right\|_F^2 = \mathcal{O}(1)$$

and

$$\left(\frac{1}{\sqrt{\beta}mL} + \frac{1}{4m(1-\rho)^2\kappa L}\right)\mathbb{E}\left\|\mathbf{R}^{(0)}\right\|_F^2 \leq \left(\frac{1}{\sqrt{\beta}mL} + \frac{1}{4m(1-\rho)^2\kappa L}\right)\frac{m\sigma^2}{\mathcal{S}_0} \leq 1,$$

where we have used the definition of $\mathcal{S}_0$. Second, by $\sqrt{\beta} \leq \frac{(1-\rho)^2}{8\sqrt{6}}$ and $L \geq 1$, it holds that $C_0 = \Theta\left(\phi(\bar{\mathbf{x}}^{(0)}, \mathbf{\Lambda}^{(0)}) - \phi^* - \hat{\delta}_0 + 1\right)$. Also by the choice of $\mathcal{S}_t = \mathcal{O}(1)$ for all $t \geq 1$, it holds that $\Upsilon_t = \mathcal{O}(\sigma^2)$ for all $t \geq 1$. Hence by (20), we have

$$\mathbb{E}\left\|\frac{1}{\eta_{\mathbf{x}}}\left(\bar{\mathbf{x}}^{(\tau)} - \mathbf{prox}_{\eta_{\mathbf{x}}g}\left(\bar{\mathbf{x}}^{(\tau)} - \eta_{\mathbf{x}}\nabla P(\bar{\mathbf{x}}^{(\tau)}, \mathbf{\Lambda}^{(\tau)})\right)\right)\right\|_2^2 + \frac{L^2}{m}\mathbb{E}\left\|\mathbf{X}_\perp^{(\tau)}\right\|_F^2$$
$$= \mathcal{O}\left(\left[\frac{\phi(\bar{\mathbf{x}}^{(0)}, \mathbf{\Lambda}^{(0)}) - \phi^* - \hat{\delta}_0 + 1}{T} + \frac{1}{\sqrt{\beta}L}\beta^2\sigma^2\right] \cdot \frac{\sigma \kappa^3 L}{\varepsilon}\right), \tag{78}$$

and similarly

$$\mathbb{E}\left\|\nabla_{\mathbf{\Lambda}}P(\bar{\mathbf{x}}^{(\tau)}, \mathbf{\Lambda}^{(\tau)})\right\|_F^2 = \mathcal{O}\left(\left[\frac{\phi(\bar{\mathbf{x}}^{(0)}, \mathbf{\Lambda}^{(0)}) - \phi^* + \hat{\delta}_0 + 1}{T} + \frac{1}{\sqrt{\beta}L}\beta^2\sigma^2\right] \cdot \frac{\sigma \kappa^3 L}{\varepsilon}\right). \tag{79}$$

Now by $\beta = \Theta\left(\frac{\varepsilon^2}{\sigma^2\kappa^2}\right)$, we have $\frac{\beta^2\sigma^2}{\sqrt{\beta}L} \cdot \frac{\sigma\kappa^3 L}{\varepsilon} = \Theta(\varepsilon^2)$, and the right-hand sides in (78) and (79) become $\mathcal{O}(T^{-1}\varepsilon^{-1}\sigma\kappa^3 L + \varepsilon^2)$. Hence $(\mathbf{X}^{(\tau)}, \mathbf{\Lambda}^{(\tau)})$ is an $\varepsilon$-stationary point when $T = \Theta\left(\frac{\sigma\kappa^3 L}{\varepsilon^3}\right)$. This completes the proof by noticing $T_c = T$ and $T_s = \Theta(T + \mathcal{S}_0)$. $\qquad\square$

## D  Proof of Lemma C.1

By the definition of $\widehat{\mathbf{Y}}^{(t)}$ in (27) and (30), we have

$$\mathbb{E}\left\|\bar{\mathbf{x}}^{(t)} - \mathbf{prox}_{\eta_{\mathbf{x}}g}\left(\bar{\mathbf{x}}^{(t)} - \eta_{\mathbf{x}}\nabla_{\mathbf{x}}P(\bar{\mathbf{x}}^{(t)}, \mathbf{\Lambda}^{(t)})\right)\right\|_2$$
$$= \mathbb{E}\left\|\bar{\mathbf{x}}^{(t)} - \mathbf{prox}_{\eta_{\mathbf{x}}g}\left(\bar{\mathbf{x}}^{(t)} - \frac{\eta_{\mathbf{x}}}{m}\sum_{j=1}^m \nabla_{\mathbf{x}}f_j(\bar{\mathbf{x}}^{(t)}, \hat{\mathbf{y}}_j^{(t)})\right)\right\|_2$$
$$\leq \mathbb{E}\left\|\bar{\mathbf{x}}^{(t)} - \mathbf{prox}_{\eta_{\mathbf{x}}g}\left(\tilde{\mathbf{x}}_i^{(t)} - \eta_{\mathbf{x}}\mathbf{v}_{\mathbf{x},i}^{(t)}\right)\right\|_2 + \mathbb{E}\left\|\bar{\mathbf{x}}^{(t)} - \frac{\eta_{\mathbf{x}}}{m}\sum_{j=1}^m \nabla_{\mathbf{x}}f_j(\bar{\mathbf{x}}^{(t)}, \hat{\mathbf{y}}_j^{(t)}) - \left(\tilde{\mathbf{x}}_i^{(t)} - \eta_{\mathbf{x}}\mathbf{v}_{\mathbf{x},i}^{(t)}\right)\right\|_2$$
$$\leq \mathbb{E}\left\|\bar{\mathbf{x}}^{(t)} - \mathbf{x}_i^{(t+1)}\right\|_2 + \mathbb{E}\left\|\bar{\mathbf{x}}^{(t)} - \tilde{\mathbf{x}}_i^{(t)}\right\|_2$$
$$\quad + \mathbb{E}\left\|\frac{\eta_{\mathbf{x}}}{m}\sum_{j=1}^m \nabla_{\mathbf{x}}f_j(\mathbf{x}_j^{(t)}, \mathbf{y}_j^{(t)}) - \frac{\eta_{\mathbf{x}}}{m}\sum_{j=1}^m \nabla_{\mathbf{x}}f_j(\bar{\mathbf{x}}^{(t)}, \hat{\mathbf{y}}_j^{(t)})\right\|_2$$
$$\quad + \left\|\frac{\eta_{\mathbf{x}}}{m}\sum_{j=1}^m \nabla_{\mathbf{x}}f_j(\mathbf{x}_j^{(t)}, \mathbf{y}_j^{(t)}) - \eta_{\mathbf{x}}\mathbf{v}_{\mathbf{x},i}^{(t)}\right\|_2$$
$$\leq \mathbb{E}\left\|\bar{\mathbf{x}}^{(t)} - \mathbf{x}_i^{(t+1)}\right\|_2 + \mathbb{E}\left\|\bar{\mathbf{x}}^{(t)} - \tilde{\mathbf{x}}_i^{(t)}\right\|_2 \tag{80}$$
$$\quad + \mathbb{E}\left\|\frac{\eta_{\mathbf{x}}}{m}\sum_{j=1}^m \nabla_{\mathbf{x}}f_j(\mathbf{x}_j^{(t)}, \mathbf{y}_j^{(t)}) - \frac{\eta_{\mathbf{x}}}{m}\sum_{j=1}^m \nabla_{\mathbf{x}}f_j(\bar{\mathbf{x}}^{(t)}, \hat{\mathbf{y}}_j^{(t)})\right\|_2$$
$$\quad + \left\|\frac{\eta_{\mathbf{x}}}{m}\sum_{j=1}^m \nabla_{\mathbf{x}}f_j(\mathbf{x}_j^{(t)}, \mathbf{y}_j^{(t)}) - \eta_{\mathbf{x}}\bar{\mathbf{d}}_{\mathbf{x}}^{(t)}\right\|_2 + \left\|\eta_{\mathbf{x}}\bar{\mathbf{v}}_{\mathbf{x}}^{(t)} - \eta_{\mathbf{x}}\mathbf{v}_{\mathbf{x},i}^{(t)}\right\|_2,$$

where the first inequality follows from the triangle inequality and the nonexpansiveness of $\mathbf{prox}_{\eta_{\mathbf{x}}g}$, the second inequality uses the update of $\mathbf{x}_i^{(t+1)}$ and the triangle inequality, and the last inequality holds by $\bar{\mathbf{d}}_{\mathbf{x}}^{(t)} = \bar{\mathbf{v}}_{\mathbf{x}}^{(t)}$ for all $t \geq 0$. Squaring both sides of (80), using Young's inequality, and summing over $i = 1, \ldots, m$ with the definition of $\mathbf{R}_{\mathbf{x}}$ in (28) yields

$$
m\mathbb{E}\left\|\bar{\mathbf{x}}^{(t)} - \mathbf{prox}_{\eta_{\mathbf{x}}g}\left(\bar{\mathbf{x}}^{(t)} - \eta_{\mathbf{x}}\nabla_{\mathbf{x}}P(\bar{\mathbf{x}}^{(t)}, \mathbf{\Lambda}^{(t)})\right)\right\|_2^2
$$
$$
\leq 5\mathbb{E}\left\|\mathbf{X}^{(t+1)} - \bar{\mathbf{X}}^{(t)}\right\|_F^2 + 5\mathbb{E}\left\|\bar{\mathbf{X}}^{(t)} - \widetilde{\mathbf{X}}^{(t)}\right\|_F^2 + 5\eta_{\mathbf{x}}^2 L^2 \left(\mathbb{E}\left\|\mathbf{X}_{\perp}^{(t)}\right\|_F^2 + \mathbb{E}\left\|\widehat{\mathbf{Y}}^{(t)} - \mathbf{Y}^{(t)}\right\|_F^2\right)
$$
$$
+ 5\eta_{\mathbf{x}}^2 \left\|\mathbf{R}_{\mathbf{x}}^{(t)}\right\|_F^2 + 5\eta_{\mathbf{x}}^2 \mathbb{E}\left\|\mathbf{V}_{\perp,\mathbf{x}}^{(t)}\right\|_F^2 \tag{81}
$$
$$
\overset{(33)}{\leq} 5\mathbb{E}\left\|\mathbf{X}^{(t+1)} - \bar{\mathbf{X}}^{(t)}\right\|_F^2 + \left(5\eta_{\mathbf{x}}^2 L^2 + 5 + 10\eta_{\mathbf{x}}^2 L^2 \kappa^2\right)\mathbb{E}\left\|\mathbf{X}_{\perp}^{(t)}\right\|_F^2 + 10\eta_{\mathbf{x}}^2 L^2 \mathbb{E}\left\|\widetilde{\mathbf{Y}}^{(t)} - \mathbf{Y}^{(t)}\right\|_F^2
$$
$$
+ 5\eta_{\mathbf{x}}^2 \left\|\mathbf{R}_{\mathbf{x}}^{(t)}\right\|_F^2 + 5\eta_{\mathbf{x}}^2 \mathbb{E}\left\|\mathbf{V}_{\perp,\mathbf{x}}^{(t)}\right\|_F^2,
$$

where we have used $\mathbf{W} - \frac{1}{m}\mathbf{1}\mathbf{1}^\top = (\mathbf{W} - \frac{1}{m}\mathbf{1}\mathbf{1}^\top)(\mathbf{I} - \frac{1}{m}\mathbf{1}\mathbf{1}^\top)$ to have $\left\|\bar{\mathbf{X}}^{(t)} - \widetilde{\mathbf{X}}^{(t)}\right\|_F^2 \leq \left\|\mathbf{X}_{\perp}^{(t)}\right\|_F^2$. By multiplying $\frac{1}{m\eta_{\mathbf{x}}^2}$ to (81), adding $\frac{L^2}{m}\mathbb{E}\left\|\mathbf{X}_{\perp}^{(t)}\right\|_F^2$, and using $\left\|\mathbf{R}_{\mathbf{x}}^{(t)}\right\|_F^2 \leq \left\|\mathbf{R}^{(t)}\right\|_F^2$, we obtain

$$
\mathbb{E}\left\|\frac{1}{\eta_{\mathbf{x}}}\left(\bar{\mathbf{x}}^{(t)} - \mathbf{prox}_{\eta_{\mathbf{x}}g}\left(\bar{\mathbf{x}}^{(t)} - \eta_{\mathbf{x}}\nabla_{\mathbf{x}}P(\bar{\mathbf{x}}^{(t)}, \mathbf{\Lambda}^{(t)})\right)\right)\right\|_2^2 + \frac{L^2}{m}\mathbb{E}\left\|\mathbf{X}_{\perp}^{(t)}\right\|_F^2
$$
$$
\leq \frac{5}{m\eta_{\mathbf{x}}^2}\mathbb{E}\left\|\mathbf{X}^{(t+1)} - \bar{\mathbf{X}}^{(t)}\right\|_F^2 + \left(\frac{2L^2(3+5\kappa^2)}{m} + \frac{5}{m\eta_{\mathbf{x}}^2}\right)\mathbb{E}\left\|\mathbf{X}_{\perp}^{(t)}\right\|_F^2 + \frac{10L^2}{m}\mathbb{E}\left\|\widetilde{\mathbf{Y}}^{(t)} - \mathbf{Y}^{(t)}\right\|_F^2 \tag{82}
$$
$$
+ \frac{5}{m}\left\|\mathbf{R}^{(t)}\right\|_F^2 + \frac{5}{m}\mathbb{E}\left\|\mathbf{V}_{\perp,\mathbf{x}}^{(t)}\right\|_F^2.
$$

This completes the proof of (36). Again, by the definition of $\widehat{\mathbf{Y}}^{(t)}$ in (27), in conjunction with (30) and Young's inequality, we have

$$
\mathbb{E}\left\|\nabla_{\mathbf{\Lambda}}P(\bar{\mathbf{x}}^{(t)}, \mathbf{\Lambda}^{(t)})\right\|_F^2 \leq 2\mathbb{E}\left\|\frac{L}{2\sqrt{m}}(\mathbf{I} - \mathbf{W})\mathbf{Y}^{(t)}\right\|_F^2 + \frac{L^2}{2m}\mathbb{E}\left\|(\mathbf{I} - \mathbf{W})(\widehat{\mathbf{Y}}^{(t)} - \mathbf{Y}^{(t)})\right\|_F^2
$$
$$
\leq \frac{2}{\eta_{\mathbf{\Lambda}}^2}\mathbb{E}\left\|\mathbf{\Lambda}^{(t+1)} - \mathbf{\Lambda}^{(t)}\right\|_F^2 + \frac{2L^2}{m}\mathbb{E}\left\|\widehat{\mathbf{Y}}^{(t)} - \mathbf{Y}^{(t)}\right\|_F^2 \tag{83}
$$
$$
\leq \frac{2}{\eta_{\mathbf{\Lambda}}^2}\mathbb{E}\left\|\mathbf{\Lambda}^{(t+1)} - \mathbf{\Lambda}^{(t)}\right\|_F^2 + \frac{4L^2\kappa^2}{m}\mathbb{E}\left\|\mathbf{X}_{\perp}^{(t)}\right\|_F^2 + \frac{4L^2}{m}\mathbb{E}\left\|\widetilde{\mathbf{Y}}^{(t)} - \mathbf{Y}^{(t)}\right\|_F^2,
$$

where we have used (16), $\|\mathbf{I} - \mathbf{W}\|_2^2 \leq 4$ by Assumption 3, and (33). This completes the proof.

## E  Proof of Lemma C.2

To prove Lemma C.2, we first state a few supporting lemmas. The following lemma can be proved in the same way as the proof of (Mancino-Ball et al., 2023a, Lemma C.3).

**Lemma E.1** *For all $t \geq 0$ and for all $i = 1, \ldots, m$,*

$$
g(\mathbf{x}_i^{(t+1)}) - g(\bar{\mathbf{x}}^{(t)})
$$
$$
\leq -\frac{1}{2\eta_{\mathbf{x}}}\left(\left\|\mathbf{x}_i^{(t+1)} - \bar{\mathbf{x}}^{(t)}\right\|_2^2 + \left\|\mathbf{x}_i^{(t+1)} - \tilde{\mathbf{x}}^{(t)}\right\|_2^2 - \left\|\bar{\mathbf{x}}^{(t)} - \tilde{\mathbf{x}}^{(t)}\right\|_2^2\right) - \left\langle\mathbf{x}_i^{(t+1)} - \bar{\mathbf{x}}^{(t)}, \mathbf{v}_{\mathbf{x},i}^{(t)}\right\rangle. \tag{84}
$$

**Lemma E.2** *For all $t \geq 0$ and arbitrary constants $c_1, c_2 > 0$, the following inequality holds*

$$
\begin{aligned}
&\phi(\bar{\mathbf{x}}^{(t+1)}, \mathbf{\Lambda}^{(t+1)}) - \phi(\bar{\mathbf{x}}^{(t)}, \mathbf{\Lambda}^{(t)}) \\
\leq & \frac{1}{m} \sum_{i=1}^{m} \left\langle \nabla_{\mathbf{x}} f_i(\mathbf{x}_i^{(t)}, \mathbf{y}_i^{(t)}), \bar{\mathbf{x}}^{(t+1)} - \bar{\mathbf{x}}^{(t)} \right\rangle - \frac{1}{m} \sum_{i=1}^{m} \left\langle \mathbf{x}_i^{(t+1)} - \bar{\mathbf{x}}^{(t)}, \mathbf{v}_{\mathbf{x},i}^{(t)} \right\rangle \\
& - \frac{1}{2m\eta_{\mathbf{x}}} \left( \left\| \mathbf{X}^{(t+1)} - \bar{\mathbf{X}}^{(t)} \right\|_F^2 + \left\| \mathbf{X}^{(t+1)} - \widetilde{\mathbf{X}}^{(t)} \right\|_F^2 - \left\| \widetilde{\mathbf{X}}^{(t)} - \bar{\mathbf{X}}^{(t)} \right\|_F^2 \right) \\
& + \left( \frac{L(\kappa+1)}{2m} + \frac{\kappa^2}{2mc_2} \right) \left\| \mathbf{X}_{\perp}^{(t)} \right\|_F^2 + \frac{L(\kappa+1+c_1) + L_P}{2} \left\| \bar{\mathbf{x}}^{(t+1)} - \bar{\mathbf{x}}^{(t)} \right\|_2^2 \\
& + \left( \frac{L}{2mc_1} + \frac{1}{2mc_2} \right) \left\| \widetilde{\mathbf{Y}}^{(t)} - \mathbf{Y}^{(t)} \right\|_F^2 \\
& + \frac{Lc_2}{4} \left\| (\mathbf{W} - \mathbf{I})^{\top} \left( \mathbf{\Lambda}^{(t+1)} - \mathbf{\Lambda}^{(t)} \right) \right\|_F^2 - \left( \frac{1}{\eta_{\mathbf{\Lambda}}} - \frac{L_P}{2} \right) \left\| \mathbf{\Lambda}^{(t+1)} - \mathbf{\Lambda}^{(t)} \right\|_F^2 .
\end{aligned}
\tag{85}
$$

*Proof.* By the $L_P$-smoothness of $P$ defined in (8), it holds that

$$
\begin{aligned}
&\phi(\bar{\mathbf{x}}^{(t+1)}, \mathbf{\Lambda}^{(t+1)}) - \phi(\bar{\mathbf{x}}^{(t)}, \mathbf{\Lambda}^{(t)}) \\
\leq & \frac{L_P}{2} \left( \left\| \bar{\mathbf{x}}^{(t+1)} - \bar{\mathbf{x}}^{(t)} \right\|_2^2 + \left\| \mathbf{\Lambda}^{(t+1)} - \mathbf{\Lambda}^{(t)} \right\|_F^2 \right) + \left\langle \nabla P(\bar{\mathbf{x}}^{(t)}, \mathbf{\Lambda}^{(t)}), \left( \bar{\mathbf{x}}^{(t+1)} - \bar{\mathbf{x}}^{(t)}, \mathbf{\Lambda}^{(t+1)} - \mathbf{\Lambda}^{(t)} \right) \right\rangle \\
& + g(\bar{\mathbf{x}}^{(t+1)}) - g(\bar{\mathbf{x}}^{(t)}) \\
\leq & \frac{L_P}{2} \left( \left\| \bar{\mathbf{x}}^{(t+1)} - \bar{\mathbf{x}}^{(t)} \right\|_2^2 + \left\| \mathbf{\Lambda}^{(t+1)} - \mathbf{\Lambda}^{(t)} \right\|_F^2 \right) \\
& + \left\langle \nabla P(\bar{\mathbf{x}}^{(t)}, \mathbf{\Lambda}^{(t)}), \left( \bar{\mathbf{x}}^{(t+1)} - \bar{\mathbf{x}}^{(t)}, \mathbf{\Lambda}^{(t+1)} - \mathbf{\Lambda}^{(t)} \right) \right\rangle - \frac{1}{m} \sum_{i=1}^{m} \left\langle \mathbf{x}_i^{(t+1)} - \bar{\mathbf{x}}^{(t)}, \mathbf{v}_{\mathbf{x},i}^{(t)} \right\rangle \\
& - \frac{1}{2m\eta_{\mathbf{x}}} \left( \left\| \mathbf{X}^{(t+1)} - \bar{\mathbf{X}}^{(t)} \right\|_F^2 + \left\| \mathbf{X}^{(t+1)} - \widetilde{\mathbf{X}}^{(t)} \right\|_F^2 - \left\| \widetilde{\mathbf{X}}^{(t)} - \bar{\mathbf{X}}^{(t)} \right\|_F^2 \right),
\end{aligned}
\tag{86}
$$

where the second inequality uses the convexity of $g$ to have $g(\bar{\mathbf{x}}^{(t+1)}) \leq \frac{1}{m} \sum_{i=1}^{m} g(\mathbf{x}_i^{(t+1)})$ and (84). By the definition of $\nabla P(\bar{\mathbf{x}}^{(t)}, \mathbf{\Lambda}^{(t)})$ in (30) and the definition of $\widehat{\mathbf{Y}}^{(t)}$ in (27), we have

$$
\begin{aligned}
&\left\langle \nabla P(\bar{\mathbf{x}}^{(t)}, \mathbf{\Lambda}^{(t)}), (\bar{\mathbf{x}}^{(t+1)} - \bar{\mathbf{x}}^{(t)}, \mathbf{\Lambda}^{(t+1)} - \mathbf{\Lambda}^{(t)}) \right\rangle \\
= & \frac{1}{m} \sum_{i=1}^{m} \left\langle \nabla_{\mathbf{x}} f_i(\bar{\mathbf{x}}^{(t)}, \hat{\mathbf{y}}_i^{(t)}), \bar{\mathbf{x}}^{(t+1)} - \bar{\mathbf{x}}^{(t)} \right\rangle - \frac{L}{2\sqrt{m}} \left\langle (\mathbf{W} - \mathbf{I}) \widehat{\mathbf{Y}}^{(t)}, \mathbf{\Lambda}^{(t+1)} - \mathbf{\Lambda}^{(t)} \right\rangle .
\end{aligned}
\tag{87}
$$

By the $L$-smoothness of $\{f_i\}$ and the Peter-Paul inequality, we further have

$$
\frac{1}{m}\sum_{i=1}^{m}\left\langle \nabla_{\mathbf{x}}f_i(\bar{\mathbf{x}}^{(t)},\hat{\mathbf{y}}_i^{(t)}),\bar{\mathbf{x}}^{(t+1)}-\bar{\mathbf{x}}^{(t)}\right\rangle
$$

$$
=\frac{1}{m}\sum_{i=1}^{m}\left(\left\langle \nabla_{\mathbf{x}}f_i(\mathbf{x}_i^{(t)},\mathbf{y}_i^{(t)}),\bar{\mathbf{x}}^{(t+1)}-\bar{\mathbf{x}}^{(t)}\right\rangle+\left\langle \nabla_{\mathbf{x}}f_i(\bar{\mathbf{x}}^{(t)},\hat{\mathbf{y}}_i^{(t)})-\nabla_{\mathbf{x}}f_i(\mathbf{x}_i^{(t)},\hat{\mathbf{y}}_i^{(t)}),\bar{\mathbf{x}}^{(t+1)}-\bar{\mathbf{x}}^{(t)}\right\rangle\right)
$$

$$
+\frac{1}{m}\sum_{i=1}^{m}\bigg(\left\langle \nabla_{\mathbf{x}}f_i(\mathbf{x}_i^{(t)},\hat{\mathbf{y}}_i^{(t)})-\nabla_{\mathbf{x}}f_i(\mathbf{x}_i^{(t)},\tilde{\mathbf{y}}_i^{(t)}),\bar{\mathbf{x}}^{(t+1)}-\bar{\mathbf{x}}^{(t)}\right\rangle
$$

$$
+\left\langle \nabla_{\mathbf{x}}f_i(\mathbf{x}_i^{(t)},\tilde{\mathbf{y}}_i^{(t)})-\nabla_{\mathbf{x}}f_i(\mathbf{x}_i^{(t)},\mathbf{y}_i^{(t)}),\bar{\mathbf{x}}^{(t+1)}-\bar{\mathbf{x}}^{(t)}\right\rangle\bigg)
$$

$$
\leq\frac{1}{m}\sum_{i=1}^{m}\left\langle \nabla_{\mathbf{x}}f_i(\mathbf{x}_i^{(t)},\mathbf{y}_i^{(t)}),\bar{\mathbf{x}}^{(t+1)}-\bar{\mathbf{x}}^{(t)}\right\rangle+\frac{L}{2m}\sum_{i=1}^{m}\left(\left\|\bar{\mathbf{x}}^{(t)}-\mathbf{x}_i^{(t)}\right\|_2^2+\left\|\bar{\mathbf{x}}^{(t+1)}-\bar{\mathbf{x}}^{(t)}\right\|_2^2\right)
$$

$$
+\frac{L}{2m}\sum_{i=1}^{m}\left(\frac{1}{\kappa}\left\|\hat{\mathbf{y}}^{(t)}-\tilde{\mathbf{y}}^{(t)}\right\|_2^2+\kappa\left\|\bar{\mathbf{x}}^{(t+1)}-\bar{\mathbf{x}}^{(t)}\right\|_2^2\right)
$$

$$
+\frac{L}{2m}\sum_{i=1}^{m}\left(\frac{1}{c_1}\left\|\tilde{\mathbf{y}}^{(t)}-\mathbf{y}_i^{(t)}\right\|_2^2+c_1\left\|\bar{\mathbf{x}}^{(t+1)}-\bar{\mathbf{x}}^{(t)}\right\|_2^2\right)
$$

$$
\leq\frac{1}{m}\sum_{i=1}^{m}\left\langle \nabla_{\mathbf{x}}f_i(\mathbf{x}_i^{(t)},\mathbf{y}_i^{(t)}),\bar{\mathbf{x}}^{(t+1)}-\bar{\mathbf{x}}^{(t)}\right\rangle+\frac{L(\kappa+1)}{2m}\left\|\mathbf{X}_\perp^{(t)}\right\|_F^2 \tag{88}
$$

$$
+\frac{L(\kappa+1+c_1)}{2m}\left\|\bar{\mathbf{X}}^{(t+1)}-\bar{\mathbf{X}}^{(t)}\right\|_F^2+\frac{L}{2mc_1}\left\|\widetilde{\mathbf{Y}}^{(t)}-\mathbf{Y}^{(t)}\right\|_F^2,
$$

where the last inequality uses (32b). Additionally, by the Peter-Paul inequality,

$$
-\frac{L}{2\sqrt{m}}\left\langle(\mathbf{W}-\mathbf{I})\widehat{\mathbf{Y}}^{(t)},\mathbf{\Lambda}^{(t+1)}-\mathbf{\Lambda}^{(t)}\right\rangle
$$

$$
=-\frac{L}{2\sqrt{m}}\left\langle(\mathbf{W}-\mathbf{I})(\widehat{\mathbf{Y}}^{(t)}-\mathbf{Y}^{(t)})+(\mathbf{W}-\mathbf{I})\mathbf{Y}^{(t)},\mathbf{\Lambda}^{(t+1)}-\mathbf{\Lambda}^{(t)}\right\rangle
$$

$$
\overset{(16)}{=}-\frac{L}{2\sqrt{m}}\left\langle\widehat{\mathbf{Y}}^{(t)}-\mathbf{Y}^{(t)},(\mathbf{W}-\mathbf{I})^\top(\mathbf{\Lambda}^{(t+1)}-\mathbf{\Lambda}^{(t)})\right\rangle-\frac{1}{\eta_{\mathbf{\Lambda}}}\left\|\mathbf{\Lambda}^{(t+1)}-\mathbf{\Lambda}^{(t)}\right\|_F^2
$$

$$
\leq\frac{L}{4mc_2}\left\|\widehat{\mathbf{Y}}^{(t)}-\mathbf{Y}^{(t)}\right\|_F^2+\frac{Lc_2}{4}\left\|(\mathbf{W}-\mathbf{I})^\top\left(\mathbf{\Lambda}^{(t+1)}-\mathbf{\Lambda}^{(t)}\right)\right\|_F^2-\frac{1}{\eta_{\mathbf{\Lambda}}}\left\|\mathbf{\Lambda}^{(t+1)}-\mathbf{\Lambda}^{(t)}\right\|_F^2
$$

$$
\leq\frac{1}{2mc_2}\left(\left\|\widehat{\mathbf{Y}}^{(t)}-\widetilde{\mathbf{Y}}^{(t)}\right\|_F^2+\left\|\widetilde{\mathbf{Y}}^{(t)}-\mathbf{Y}^{(t)}\right\|_F^2\right)
$$

$$
+\frac{Lc_2}{4}\left\|(\mathbf{W}-\mathbf{I})^\top\left(\mathbf{\Lambda}^{(t+1)}-\mathbf{\Lambda}^{(t)}\right)\right\|_F^2-\frac{1}{\eta_{\mathbf{\Lambda}}}\left\|\mathbf{\Lambda}^{(t+1)}-\mathbf{\Lambda}^{(t)}\right\|_F^2
$$

$$
\leq\frac{1}{2mc_2}\left(\kappa^2\left\|\mathbf{X}_\perp^{(t)}\right\|_F^2+\left\|\widetilde{\mathbf{Y}}^{(t)}-\mathbf{Y}^{(t)}\right\|_F^2\right)+\frac{Lc_2}{4}\left\|(\mathbf{W}-\mathbf{I})^\top\left(\mathbf{\Lambda}^{(t+1)}-\mathbf{\Lambda}^{(t)}\right)\right\|_F^2 \tag{89}
$$

$$
-\frac{1}{\eta_{\mathbf{\Lambda}}}\left\|\mathbf{\Lambda}^{(t+1)}-\mathbf{\Lambda}^{(t)}\right\|_F^2,
$$

where the last inequality uses (32b). Plugging (88) and (89) into (87) results in

$$
\left\langle\nabla P(\bar{\mathbf{x}}^{(t)},\mathbf{\Lambda}^{(t)}),\left(\bar{\mathbf{x}}^{(t+1)}-\bar{\mathbf{x}}^{(t)},\mathbf{\Lambda}^{(t+1)}-\mathbf{\Lambda}^{(t)}\right)\right\rangle
$$

$$
\leq\frac{1}{m}\sum_{i=1}^{m}\left\langle\nabla_{\mathbf{x}}f_i(\mathbf{x}_i^{(t)},\mathbf{y}_i^{(t)}),\bar{\mathbf{x}}^{(t+1)}-\bar{\mathbf{x}}^{(t)}\right\rangle+\left(\frac{L(\kappa+1)}{2m}+\frac{\kappa^2}{2mc_2}\right)\left\|\mathbf{X}_\perp^{(t)}\right\|_F^2
$$

$$
+\frac{L(\kappa+1+c_1)}{2m}\left\|\bar{\mathbf{X}}^{(t+1)}-\bar{\mathbf{X}}^{(t)}\right\|_F^2+\left(\frac{L}{2mc_1}+\frac{1}{2mc_2}\right)\left\|\widetilde{\mathbf{Y}}^{(t)}-\mathbf{Y}^{(t)}\right\|_F^2 \tag{90}
$$

$$
+\frac{Lc_2}{4}\left\|(\mathbf{W}-\mathbf{I})^\top\left(\mathbf{\Lambda}^{(t+1)}-\mathbf{\Lambda}^{(t)}\right)\right\|_F^2-\frac{1}{\eta_{\mathbf{\Lambda}}}\left\|\mathbf{\Lambda}^{(t+1)}-\mathbf{\Lambda}^{(t)}\right\|_F^2.
$$

Utilizing (90) in (86) and noting $\frac{1}{m}\left\|\bar{\mathbf{X}}^{(t+1)} - \bar{\mathbf{X}}^{(t)}\right\|_F^2 = \left\|\bar{\mathbf{x}}^{(t+1)} - \bar{\mathbf{x}}^{(t)}\right\|_2^2$ completes the proof. $\qquad\square$

**Lemma E.3** *For all $t \geq 0$, it holds that*

$$
\frac{1}{m}\sum_{i=1}^{m}\left\langle \nabla_{\mathbf{x}}f_i(\mathbf{x}_i^{(t)}, \mathbf{y}_i^{(t)}), \bar{\mathbf{x}}^{(t+1)} - \bar{\mathbf{x}}^{(t)}\right\rangle - \frac{1}{m}\sum_{i=1}^{m}\left\langle \mathbf{x}_i^{(t+1)} - \bar{\mathbf{x}}^{(t)}, \mathbf{v}_{\mathbf{x},i}^{(t)}\right\rangle
$$
$$
\leq \frac{1}{2m}\sum_{i=1}^{m}\left( L_P c_3 \left\|\mathbf{x}_i^{(t+1)} - \bar{\mathbf{x}}^{(t)}\right\|_2^2 + \frac{1}{L_P c_3}\left\|\frac{1}{m}\sum_{j=1}^{m}\nabla_{\mathbf{x}}f_j(\mathbf{x}_j^{(t)}, \mathbf{y}_j^{(t)}) - \mathbf{v}_{\mathbf{x},i}^{(t)}\right\|_2^2\right), \tag{91}
$$

*where $c_3 > 0$ is an arbitrary constant.*

*Proof.* Notice

$$
\left\langle \frac{1}{m}\sum_{i=1}^{m}\nabla_{\mathbf{x}}f_i(\mathbf{x}_i^{(t)}, \mathbf{y}_i^{(t)}), \bar{\mathbf{x}}^{(t+1)} - \bar{\mathbf{x}}^{(t)}\right\rangle - \frac{1}{m}\sum_{i=1}^{m}\left\langle \mathbf{x}_i^{(t+1)} - \bar{\mathbf{x}}^{(t)}, \mathbf{v}_{\mathbf{x},i}^{(t)}\right\rangle
$$
$$
= \frac{1}{m}\sum_{i=1}^{m}\left\langle \frac{1}{m}\sum_{j=1}^{m}\nabla_{\mathbf{x}}f_j(\mathbf{x}_j^{(t)}, \mathbf{y}_j^{(t)}), \mathbf{x}_i^{(t+1)} - \bar{\mathbf{x}}^{(t)}\right\rangle - \frac{1}{m}\sum_{i=1}^{m}\left\langle \mathbf{x}_i^{(t+1)} - \bar{\mathbf{x}}^{(t)}, \mathbf{v}_{\mathbf{x},i}^{(t)}\right\rangle
$$
$$
= \frac{1}{m}\sum_{i=1}^{m}\left\langle \frac{1}{m}\sum_{j=1}^{m}\nabla_{\mathbf{x}}f_j(\mathbf{x}_j^{(t)}, \mathbf{y}_j^{(t)}) - \mathbf{v}_{\mathbf{x},i}^{(t)}, \mathbf{x}_i^{(t+1)} - \bar{\mathbf{x}}^{(t)}\right\rangle.
$$

Then the desired result follows from the Peter-Paul inequality. $\qquad\square$

*Proof.* (Of Lemma C.2) The proof follows from applying (91) to (85) to have

$$
\phi(\bar{\mathbf{x}}^{(t+1)}, \mathbf{\Lambda}^{(t+1)}) - \phi(\bar{\mathbf{x}}^{(t)}, \mathbf{\Lambda}^{(t)})
$$
$$
\leq \frac{1}{2m}\sum_{i=1}^{m}\left( L_P c_3 \left\|\mathbf{x}_i^{(t+1)} - \bar{\mathbf{x}}^{(t)}\right\|_2^2 + \frac{1}{L_P c_3}\left\|\frac{1}{m}\sum_{j=1}^{m}\nabla_{\mathbf{x}}f_j(\mathbf{x}_j^{(t)}, \mathbf{y}_j^{(t)}) - \mathbf{v}_{\mathbf{x},i}^{(t)}\right\|_2^2\right)
$$
$$
- \frac{1}{2m\eta_{\mathbf{x}}}\left(\left\|\mathbf{X}^{(t+1)} - \bar{\mathbf{X}}^{(t)}\right\|_F^2 + \left\|\mathbf{X}^{(t+1)} - \widetilde{\mathbf{X}}^{(t)}\right\|_F^2 - \left\|\widetilde{\mathbf{X}}^{(t)} - \bar{\mathbf{X}}^{(t)}\right\|_F^2\right)
$$
$$
+ \left(\frac{L(\kappa+1)}{2m} + \frac{\kappa^2}{2mc_2}\right)\left\|\mathbf{X}_\perp^{(t)}\right\|_F^2
$$
$$
+ \frac{L(\kappa+1+c_1)+L_P}{2}\left\|\bar{\mathbf{x}}^{(t+1)} - \bar{\mathbf{x}}^{(t)}\right\|_2^2 + \left(\frac{L}{2mc_1} + \frac{1}{2mc_2}\right)\left\|\widetilde{\mathbf{Y}}^{(t)} - \mathbf{Y}^{(t)}\right\|_F^2
$$
$$
+ \frac{Lc_2}{4}\left\|(\mathbf{W}-\mathbf{I})^\top\left(\mathbf{\Lambda}^{(t+1)} - \mathbf{\Lambda}^{(t)}\right)\right\|_F^2 - \left(\frac{1}{\eta_{\mathbf{\Lambda}}} - \frac{L_P}{2}\right)\left\|\mathbf{\Lambda}^{(t+1)} - \mathbf{\Lambda}^{(t)}\right\|_F^2 \tag{92}
$$

and using the inequalities $\left\|\widetilde{\mathbf{X}}^{(t)} - \bar{\mathbf{X}}^{(t)}\right\|_F^2 = \left\|\left(\mathbf{W} - \frac{1}{m}\mathbf{1}\mathbf{1}^\top\right)\mathbf{X}_\perp^{(t)}\right\|_F^2 \leq \rho^2\left\|\mathbf{X}_\perp^{(t)}\right\|_F^2$, $\|\mathbf{W}-\mathbf{I}\|_2^2 \leq 4$, and $\left\|\bar{\mathbf{x}}^{(t+1)} - \bar{\mathbf{x}}^{(t)}\right\|_2^2 \leq \frac{1}{m}\left\|\mathbf{X}^{(t+1)} - \bar{\mathbf{X}}^{(t)}\right\|_F^2$, to further upper bound the right-hand side of (92). $\qquad\square$

## F   Proofs of Lemmas in Section C.3

**Proof of Lemma C.4** By Young's inequality, it holds that

$$
\sum_{i=1}^{m}\left\|\frac{1}{m}\sum_{j=1}^{m}\nabla_{\mathbf{x}}f_j(\mathbf{x}_j^{(t)}, \mathbf{y}_j^{(t)}) - \mathbf{v}_{\mathbf{x},i}^{(t)}\right\|_2^2
$$
$$
\leq 2\sum_{i=1}^{m}\left(\left\|\frac{1}{m}\sum_{j=1}^{m}\nabla_{\mathbf{x}}f_j(\mathbf{x}_j^{(t)}, \mathbf{y}_j^{(t)}) - \bar{\mathbf{d}}_{\mathbf{x}}^{(t)}\right\|_2^2 + \left\|\bar{\mathbf{d}}_{\mathbf{x}}^{(t)} - \mathbf{v}_{\mathbf{x},i}^{(t)}\right\|_2^2\right)
$$
$$
\leq 2\left\|\nabla_{\mathbf{x}}F(\mathbf{X}^{(t)}, \mathbf{Y}^{(t)}) - \mathbf{D}_{\mathbf{x}}^{(t)}\right\|_F^2 + 2\left\|\mathbf{V}_{\perp,\mathbf{x}}^{(t)}\right\|_F^2 \leq 2\left\|\nabla F(\mathbf{X}^{(t)}, \mathbf{Y}^{(t)}) - \mathbf{D}^{(t)}\right\|_F^2 + 2\left\|\mathbf{V}_{\perp,\mathbf{x}}^{(t)}\right\|_F^2,
$$

where the second inequality follows from Jensen's inequality and $\bar{\mathbf{d}}_{\mathbf{x}}^{(t)} = \bar{\mathbf{v}}_{\mathbf{x}}^{(t)}, \forall\, t \geq 0$. Taking the expectation yields (41). $\qquad\square$

**Proof of Lemma C.5** By the $\mathbf{Y}$ update defined in (17), it holds for all $i = 1, \ldots, m$ that

$$\mathbf{0} \in \partial h(\mathbf{y}_i^{(t+1)}) + \frac{1}{\eta_{\mathbf{y}}} \left( \mathbf{y}_i^{(t+1)} - \left( \mathbf{y}_i^{(t)} + \eta_{\mathbf{y}} \mathbf{v}_{\mathbf{y},i}^{(t)} \right) \right), \tag{93}$$

and hence for some $\tilde{\nabla} h(\mathbf{y}_i^{(t+1)}) \in \partial h(\mathbf{y}_i^{(t+1)})$ and any $\mathbf{y}_i \in \mathrm{dom}(h)$, it holds that

$$0 = \left\langle \mathbf{y}_i^{(t+1)} - \mathbf{y}_i, \tilde{\nabla} h(\mathbf{y}_i^{(t+1)}) + \frac{1}{\eta_{\mathbf{y}}} \left( \mathbf{y}_i^{(t+1)} - \left( \mathbf{y}_i^{(t)} + \eta_{\mathbf{y}} \mathbf{v}_{\mathbf{y},i}^{(t)} \right) \right) \right\rangle. \tag{94}$$

By the convexity of $h$, we further have

$$\begin{aligned}
h(\mathbf{y}_i) \geq & h(\mathbf{y}_i^{(t+1)}) + \left\langle \mathbf{y}_i - \mathbf{y}_i^{(t+1)}, \tilde{\nabla} h(\mathbf{y}_i^{(t+1)}) \right\rangle \\
& \overset{(94)}{=} h(\mathbf{y}_i^{(t+1)}) + \left\langle \mathbf{y}_i^{(t+1)} - \mathbf{y}_i, \frac{1}{\eta_{\mathbf{y}}} \left( \mathbf{y}_i^{(t+1)} - \left( \mathbf{y}_i^{(t)} + \eta_{\mathbf{y}} \mathbf{v}_{\mathbf{y},i}^{(t)} \right) \right) \right\rangle.
\end{aligned} \tag{95}$$

Summing (95) over $i = 1, \ldots, m$ and taking $\mathbf{y}_i = \mathbf{y}_i^{(t)}$ for all $i = 1, \ldots, m$ gives

$$\frac{1}{\eta_{\mathbf{y}}} \left\| \mathbf{Y}^{(t+1)} - \mathbf{Y}^{(t)} \right\|_F^2 - \left\langle \mathbf{Y}^{(t+1)} - \mathbf{Y}^{(t)}, \mathbf{V}_{\mathbf{y}}^{(t)} \right\rangle \leq \sum_{i=1}^m \left( h(\mathbf{y}_i^{(t)}) - h(\mathbf{y}_i^{(t+1)}) \right). \tag{96}$$

Now by the definition of $\Gamma_t(\mathbf{Y})$ from (29) and the update for $\mathbf{V}_{\mathbf{y}}$ from (15), it holds that

$$m \nabla \Gamma_t(\mathbf{Y}^{(t)}) - \mathbf{V}_{\mathbf{y}}^{(t)} = \nabla_{\mathbf{y}} F(\mathbf{X}^{(t)}, \mathbf{Y}^{(t)}) - \mathbf{D}_{\mathbf{y}}^{(t)}. \tag{97}$$

By Assumption 1, $-\Gamma_t(\cdot)$ is $\frac{L}{m}$-smooth for all $t \geq 0$ and hence

$$\begin{aligned}
& - \left\langle \mathbf{Y}^{(t+1)} - \mathbf{Y}^{(t)}, \mathbf{V}_{\mathbf{y}}^{(t)} \right\rangle \\
= & - m \left\langle \mathbf{Y}^{(t+1)} - \mathbf{Y}^{(t)}, \nabla \Gamma_t(\mathbf{Y}^{(t)}) \right\rangle - \left\langle \mathbf{Y}^{(t+1)} - \mathbf{Y}^{(t)}, \mathbf{D}_{\mathbf{y}}^{(t)} - \nabla_{\mathbf{y}} F(\mathbf{X}^{(t)}, \mathbf{Y}^{(t)}) \right\rangle \\
\geq & m \left( \Gamma_t(\mathbf{Y}^{(t)}) - \Gamma_t(\mathbf{Y}^{(t+1)}) \right) - \frac{L}{2} \left\| \mathbf{Y}^{(t+1)} - \mathbf{Y}^{(t)} \right\|_F^2 - \left\langle \mathbf{Y}^{(t+1)} - \mathbf{Y}^{(t)}, \mathbf{R}_{\mathbf{y}}^{(t)} \right\rangle,
\end{aligned} \tag{98}$$

where we have used the definition of $\mathbf{R}_{\mathbf{y}}$ from (28). Next, we compute

$$\begin{aligned}
& - \Gamma_t(\mathbf{Y}^{(t+1)}) + \Gamma_{t+1}(\mathbf{Y}^{(t+1)}) \\
= & - \frac{1}{m} \sum_{i=1}^m f_i(\mathbf{x}_i^{(t)}, \mathbf{y}_i^{(t+1)}) + \frac{L}{2\sqrt{m}} \left\langle (\mathbf{W} - \mathbf{I})\mathbf{Y}^{(t+1)}, \mathbf{\Lambda}^{(t)} \right\rangle \\
& + \frac{1}{m} \sum_{i=1}^m f_i(\mathbf{x}_i^{(t+1)}, \mathbf{y}_i^{(t+1)}) - \frac{L}{2\sqrt{m}} \left\langle (\mathbf{W} - \mathbf{I})\mathbf{Y}^{(t+1)}, \mathbf{\Lambda}^{(t+1)} \right\rangle \\
\geq & \frac{1}{m} \sum_{i=1}^m \left( \left\langle \nabla_{\mathbf{x}} f_i(\mathbf{x}_i^{(t+1)}, \mathbf{y}_i^{(t+1)}), \mathbf{x}_i^{(t+1)} - \mathbf{x}_i^{(t)} \right\rangle - \frac{L}{2} \left\| \mathbf{x}_i^{(t+1)} - \mathbf{x}_i^{(t)} \right\|_2^2 \right) \\
& + \frac{L}{2\sqrt{m}} \left\langle (\mathbf{W} - \mathbf{I})\mathbf{Y}^{(t+1)}, \mathbf{\Lambda}^{(t)} - \mathbf{\Lambda}^{(t+1)} \right\rangle \\
= & \frac{1}{m} \sum_{i=1}^m \left( \left\langle \nabla_{\mathbf{x}} f_i(\mathbf{x}_i^{(t+1)}, \tilde{\mathbf{y}}_i^{(t+1)}), \mathbf{x}_i^{(t+1)} - \mathbf{x}_i^{(t)} \right\rangle - \frac{L}{2} \left\| \mathbf{x}_i^{(t+1)} - \mathbf{x}_i^{(t)} \right\|_2^2 \right) \\
& + \frac{1}{m} \left\langle \nabla_{\mathbf{x}} F(\mathbf{X}^{(t+1)}, \mathbf{Y}^{(t+1)}) - \nabla_{\mathbf{x}} F(\mathbf{X}^{(t+1)}, \widetilde{\mathbf{Y}}^{(t+1)}), \mathbf{X}^{(t+1)} - \mathbf{X}^{(t)} \right\rangle \\
& + \frac{L}{2\sqrt{m}} \left\langle (\mathbf{W} - \mathbf{I})\widetilde{\mathbf{Y}}^{(t+1)}, \mathbf{\Lambda}^{(t)} - \mathbf{\Lambda}^{(t+1)} \right\rangle + \frac{L}{2\sqrt{m}} \left\langle (\mathbf{W} - \mathbf{I})(\mathbf{Y}^{(t+1)} - \widetilde{\mathbf{Y}}^{(t+1)}), \mathbf{\Lambda}^{(t)} - \mathbf{\Lambda}^{(t+1)} \right\rangle,
\end{aligned} \tag{99}$$

where the inequality uses Assumption 1 and $\widetilde{\mathbf{Y}}^{(t)}$ is defined in (27) for all $t \geq 0$. By

$$\nabla Q(\mathbf{X}^{(t+1)}, \boldsymbol{\Lambda}^{(t+1)}) = \left( \tfrac{1}{m} \nabla F(\mathbf{X}^{(t+1)}, \widetilde{\mathbf{Y}}^{(t+1)})^\top, \ -\tfrac{L}{2\sqrt{m}}(\mathbf{W} - \mathbf{I})\widetilde{\mathbf{Y}}^{(t+1)} \right),$$

we obtain from (35) that

$$
\begin{aligned}
&\frac{1}{m} \sum_{i=1}^m \left\langle \nabla_\mathbf{x} f_i(\mathbf{x}_i^{(t+1)}, \tilde{\mathbf{y}}_i^{(t+1)}), \mathbf{x}_i^{(t+1)} - \mathbf{x}_i^{(t)} \right\rangle + \frac{L}{2\sqrt{m}} \left\langle (\mathbf{W} - \mathbf{I})\widetilde{\mathbf{Y}}^{(t+1)}, \boldsymbol{\Lambda}^{(t)} - \boldsymbol{\Lambda}^{(t+1)} \right\rangle \\
&\geq -Q(\mathbf{X}^{(t)}, \boldsymbol{\Lambda}^{(t)}) + Q(\mathbf{X}^{(t+1)}, \boldsymbol{\Lambda}^{(t+1)}) - \frac{L_P}{2m} \left\| \mathbf{X}^{(t+1)} - \mathbf{X}^{(t)} \right\|_F^2 - \frac{L_P}{2} \left\| \boldsymbol{\Lambda}^{(t+1)} - \boldsymbol{\Lambda}^{(t)} \right\|_F^2.
\end{aligned}
\tag{100}
$$

Applying (100) to (99) and rearranging results in

$$
\begin{aligned}
&- \Gamma_t(\mathbf{Y}^{(t+1)}) + Q(\mathbf{X}^{(t)}, \boldsymbol{\Lambda}^{(t)}) + \Gamma_{t+1}(\mathbf{Y}^{(t+1)}) - Q(\mathbf{X}^{(t+1)}, \boldsymbol{\Lambda}^{(t+1)}) \\
&\geq \frac{1}{m} \left\langle \nabla_\mathbf{x} F(\mathbf{X}^{(t+1)}, \mathbf{Y}^{(t+1)}) - \nabla_\mathbf{x} F(\mathbf{X}^{(t+1)}, \widetilde{\mathbf{Y}}^{(t+1)}), \mathbf{X}^{(t+1)} - \mathbf{X}^{(t)} \right\rangle \\
&\quad + \frac{L}{2\sqrt{m}} \left\langle (\mathbf{W} - \mathbf{I})(\mathbf{Y}^{(t+1)} - \widetilde{\mathbf{Y}}^{(t+1)}), \boldsymbol{\Lambda}^{(t)} - \boldsymbol{\Lambda}^{(t+1)} \right\rangle \\
&\quad - \frac{L_P + L}{2m} \left\| \mathbf{X}^{(t+1)} - \mathbf{X}^{(t)} \right\|_F^2 - \frac{L_P}{2} \left\| \boldsymbol{\Lambda}^{(t+1)} - \boldsymbol{\Lambda}^{(t)} \right\|_F^2.
\end{aligned}
\tag{101}
$$

Adding $m$ times of (101) to (98) results in

$$
\begin{aligned}
&- \left\langle \mathbf{Y}^{(t+1)} - \mathbf{Y}^{(t)}, \mathbf{V}_\mathbf{y}^{(t)} \right\rangle + m \left( \Gamma_{t+1}(\mathbf{Y}^{(t+1)}) - Q(\mathbf{X}^{(t+1)}, \boldsymbol{\Lambda}^{(t+1)}) - \Gamma_t(\mathbf{Y}^{(t)}) + Q(\mathbf{X}^{(t)}, \boldsymbol{\Lambda}^{(t)}) \right) \\
&\geq -\frac{L}{2} \left\| \mathbf{Y}^{(t+1)} - \mathbf{Y}^{(t)} \right\|_F^2 - \left\langle \mathbf{Y}^{(t+1)} - \mathbf{Y}^{(t)}, \mathbf{R}_\mathbf{y}^{(t)} \right\rangle \\
&\quad + \left\langle \nabla_\mathbf{x} F(\mathbf{X}^{(t+1)}, \mathbf{Y}^{(t+1)}) - \nabla_\mathbf{x} F(\mathbf{X}^{(t+1)}, \widetilde{\mathbf{Y}}^{(t+1)}), \mathbf{X}^{(t+1)} - \mathbf{X}^{(t)} \right\rangle \\
&\quad + \frac{L\sqrt{m}}{2} \left\langle (\mathbf{W} - \mathbf{I})(\mathbf{Y}^{(t+1)} - \widetilde{\mathbf{Y}}^{(t+1)}), \boldsymbol{\Lambda}^{(t)} - \boldsymbol{\Lambda}^{(t+1)} \right\rangle - \frac{L_P + L}{2} \left\| \mathbf{X}^{(t+1)} - \mathbf{X}^{(t)} \right\|_F^2 \\
&\quad - \frac{m L_P}{2} \left\| \boldsymbol{\Lambda}^{(t+1)} - \boldsymbol{\Lambda}^{(t)} \right\|_F^2.
\end{aligned}
\tag{102}
$$

Applying (102) to (96), using the definition of $\hat{\delta}_t$ from (43), and rearranging terms results in

$$
\begin{aligned}
&m\left( \hat{\delta}_t - \hat{\delta}_{t+1} \right) \\
&\geq \left( \tfrac{1}{\eta_\mathbf{y}} - \tfrac{L}{2} \right) \left\| \mathbf{Y}^{(t+1)} - \mathbf{Y}^{(t)} \right\|_F^2 - \left\langle \mathbf{Y}^{(t+1)} - \mathbf{Y}^{(t)}, \mathbf{R}_\mathbf{y}^{(t)} \right\rangle \\
&\quad + \left\langle \nabla_\mathbf{x} F(\mathbf{X}^{(t+1)}, \mathbf{Y}^{(t+1)}) - \nabla_\mathbf{x} F(\mathbf{X}^{(t+1)}, \widetilde{\mathbf{Y}}^{(t+1)}), \mathbf{X}^{(t+1)} - \mathbf{X}^{(t)} \right\rangle \\
&\quad + \frac{L\sqrt{m}}{2} \left\langle (\mathbf{W} - \mathbf{I})(\mathbf{Y}^{(t+1)} - \widetilde{\mathbf{Y}}^{(t+1)}), \boldsymbol{\Lambda}^{(t)} - \boldsymbol{\Lambda}^{(t+1)} \right\rangle - \frac{L_P + L}{2} \left\| \mathbf{X}^{(t+1)} - \mathbf{X}^{(t)} \right\|_F^2 \\
&\quad - \frac{m L_P}{2} \left\| \boldsymbol{\Lambda}^{(t+1)} - \boldsymbol{\Lambda}^{(t)} \right\|_F^2.
\end{aligned}
\tag{103}
$$

By the Peter-Paul inequality, we have that for any $c_4, c_5 > 0$,

$$\left\langle \mathbf{Y}^{(t)} - \mathbf{Y}^{(t+1)}, \mathbf{R}_\mathbf{y}^{(t)} \right\rangle \leq \tfrac{1}{4\eta_\mathbf{y}} \left\| \mathbf{Y}^{(t+1)} - \mathbf{Y}^{(t)} \right\|_F^2 + \eta_\mathbf{y} \left\| \mathbf{R}_\mathbf{y}^{(t)} \right\|_F^2, \tag{104}$$

$$\left\langle \nabla_\mathbf{x} F(\mathbf{X}^{(t+1)}, \mathbf{Y}^{(t+1)}) - \nabla_\mathbf{x} F(\mathbf{X}^{(t+1)}, \widetilde{\mathbf{Y}}^{(t+1)}), \mathbf{X}^{(t)} - \mathbf{X}^{(t+1)} \right\rangle \tag{105}$$

$$\leq \tfrac{L^2}{2c_4} \left\| \widetilde{\mathbf{Y}}^{(t+1)} - \mathbf{Y}^{(t+1)} \right\|_F^2 + \tfrac{c_4}{2} \left\| \mathbf{X}^{(t+1)} - \mathbf{X}^{(t)} \right\|_F^2,$$

$$\tfrac{L\sqrt{m}}{2} \left\langle (\mathbf{W} - \mathbf{I})(\mathbf{Y}^{(t+1)} - \widetilde{\mathbf{Y}}^{(t+1)}), \boldsymbol{\Lambda}^{(t+1)} - \boldsymbol{\Lambda}^{(t)} \right\rangle \tag{106}$$

$$\leq \tfrac{L\sqrt{m}}{c_5} \left\| \widetilde{\mathbf{Y}}^{(t+1)} - \mathbf{Y}^{(t+1)} \right\|_F^2 + \tfrac{c_5 L\sqrt{m}}{4} \left\| \boldsymbol{\Lambda}^{(t+1)} - \boldsymbol{\Lambda}^{(t)} \right\|_F^2,$$

where we have used Assumption 1 in (105) and Assumption 3 in (106). Applying (104)-(106) to (103) results in

$$
\begin{aligned}
&\left(\tfrac{1}{\eta_{\mathbf{y}}} - \tfrac{L}{2} - \tfrac{1}{4\eta_{\mathbf{y}}}\right)\left\|\mathbf{Y}^{(t+1)} - \mathbf{Y}^{(t)}\right\|_F^2 \\
&\leq m\left(\hat{\delta}_t - \hat{\delta}_{t+1}\right) + \eta_{\mathbf{y}}\left\|\mathbf{R}_{\mathbf{y}}^{(t)}\right\|_F^2 + \left(\tfrac{L^2}{2c_4} + \tfrac{L\sqrt{m}}{c_5}\right)\left\|\widetilde{\mathbf{Y}}^{(t+1)} - \mathbf{Y}^{(t+1)}\right\|_F^2 \\
&\quad + \left(\tfrac{L_P + L}{2} + \tfrac{c_4}{2}\right)\left\|\mathbf{X}^{(t+1)} - \mathbf{X}^{(t)}\right\|_F^2 + \left(\tfrac{mL_P}{2} + \tfrac{c_5 L\sqrt{m}}{4}\right)\left\|\mathbf{\Lambda}^{(t+1)} - \mathbf{\Lambda}^{(t)}\right\|_F^2.
\end{aligned}
\tag{107}
$$

By $\eta_{\mathbf{y}} \leq \tfrac{1}{4L}$, it holds that $\tfrac{1}{4\eta_{\mathbf{y}}} \leq \tfrac{1}{\eta_{\mathbf{y}}} - \tfrac{L}{2} - \tfrac{1}{4\eta_{\mathbf{y}}}$ and hence

$$
\begin{aligned}
&\left\|\mathbf{Y}^{(t+1)} - \mathbf{Y}^{(t)}\right\|_F^2 \\
&\leq 4m\eta_{\mathbf{y}}\left(\hat{\delta}_t - \hat{\delta}_{t+1}\right) + 4\eta_{\mathbf{y}}^2\left\|\mathbf{R}_{\mathbf{y}}^{(t)}\right\|_F^2 + 4\eta_{\mathbf{y}}\left(\tfrac{L^2}{2c_4} + \tfrac{L\sqrt{m}}{c_5}\right)\left\|\widetilde{\mathbf{Y}}^{(t+1)} - \mathbf{Y}^{(t+1)}\right\|_F^2 \\
&\quad + 4\eta_{\mathbf{y}}\left(\tfrac{L_P + L}{2} + \tfrac{c_4}{2}\right)\left\|\mathbf{X}^{(t+1)} - \mathbf{X}^{(t)}\right\|_F^2 + 4\eta_{\mathbf{y}}\left(\tfrac{mL_P}{2} + \tfrac{c_5 L\sqrt{m}}{4}\right)\left\|\mathbf{\Lambda}^{(t+1)} - \mathbf{\Lambda}^{(t)}\right\|_F^2.
\end{aligned}
\tag{108}
$$

Applying $\left\|\mathbf{R}_{\mathbf{y}}^{(t)}\right\|_F^2 \leq \left\|\mathbf{R}^{(t)}\right\|_F^2$ completes the proof. $\qquad\square$

**Proof of Lemma C.6** By the definition of $\widetilde{\mathbf{Y}}^{(t)}$ in (27), it holds that

$$
\mathbf{0} \in \frac{1}{m}\left[\nabla_{\mathbf{y}} f_1(\mathbf{x}_1^{(t)}, \tilde{\mathbf{y}}_1^{(t)}) - \partial h(\tilde{\mathbf{y}}_1^{(t)}), \ldots, \nabla_{\mathbf{y}} f_m(\mathbf{x}_m^{(t)}, \tilde{\mathbf{y}}_m^{(t)}) - \partial h(\tilde{\mathbf{y}}_m^{(t)})\right]^\top - \frac{L}{2\sqrt{m}}(\mathbf{W} - \mathbf{I})^\top \mathbf{\Lambda}^{(t)}.
\tag{109}
$$

Defining $\tilde{\mathbf{\Lambda}}^{(t)} := (\mathbf{W} - \mathbf{I})^\top \mathbf{\Lambda}^{(t)}$, we have for all $i = 1, \ldots, m$,

$$
\tilde{\mathbf{y}}_i^{(t)} = \arg\max_{\mathbf{y}_i}\left\{\left\langle \mathbf{y}_i, \nabla_{\mathbf{y}} f_i(\mathbf{x}_i^{(t)}, \tilde{\mathbf{y}}_i^{(t)}) - \frac{L\sqrt{m}}{2}\tilde{\mathbf{\lambda}}_i^{(t)}\right\rangle - \frac{1}{2\eta_{\mathbf{y}}}\left\|\mathbf{y}_i - \tilde{\mathbf{y}}_i^{(t)}\right\|_2^2 - h(\mathbf{y}_i)\right\}
\tag{110}
$$

where $(\tilde{\mathbf{\lambda}}_i^{(t)})^\top$ denotes the $i$-th row of $\tilde{\mathbf{\Lambda}}^{(t)}$. Hence,

$$
\tilde{\mathbf{y}}_i^{(t)} = \mathbf{prox}_{\eta_{\mathbf{y}} h}\left(\tilde{\mathbf{y}}_i^{(t)} + \eta_{\mathbf{y}}\left(\nabla_{\mathbf{y}} f_i(\mathbf{x}_i^{(t)}, \tilde{\mathbf{y}}_i^{(t)}) - \frac{L\sqrt{m}}{2}\tilde{\mathbf{\lambda}}_i^{(t)}\right)\right).
\tag{111}
$$

By the non-expansiveness of the proximal operator,

$$
\begin{aligned}
&\left\|\tilde{\mathbf{y}}_i^{(t)} - \mathbf{y}_i^{(t+1)}\right\|_2^2 \\
&= \left\|\mathbf{prox}_{\eta_{\mathbf{y}} h}\left(\tilde{\mathbf{y}}_i^{(t)} + \eta_{\mathbf{y}}\left(\nabla_{\mathbf{y}} f_i(\mathbf{x}_i^{(t)}, \tilde{\mathbf{y}}_i^{(t)}) - \frac{L\sqrt{m}}{2}\tilde{\mathbf{\lambda}}_i^{(t)}\right)\right) - \mathbf{prox}_{\eta_{\mathbf{y}} h}\left(\mathbf{y}_i^{(t)} + \eta_{\mathbf{y}}\mathbf{v}_{\mathbf{y},i}^{(t)}\right)\right\|_2^2 \\
&\leq \left\|\tilde{\mathbf{y}}_i^{(t)} + \eta_{\mathbf{y}}\left(\nabla_{\mathbf{y}} f_i(\mathbf{x}_i^{(t)}, \tilde{\mathbf{y}}_i^{(t)}) - \frac{L\sqrt{m}}{2}\tilde{\mathbf{\lambda}}_i^{(t)}\right) - \mathbf{y}_i^{(t)} - \eta_{\mathbf{y}}\mathbf{v}_{\mathbf{y},i}^{(t)}\right\|_2^2.
\end{aligned}
\tag{112}
$$

Utilizing the $\mathbf{V}_{\mathbf{y}}$-update (15) in (112) and the Peter-Paul inequality we obtain

$$
\begin{aligned}
\left\|\tilde{\mathbf{y}}_i^{(t)} - \mathbf{y}_i^{(t+1)}\right\|_F^2 &\leq (1+b)\left\|\tilde{\mathbf{y}}_i^{(t)} - \mathbf{y}_i^{(t)} + \eta_{\mathbf{y}}\left(\nabla_{\mathbf{y}} f_i(\mathbf{x}_i^{(t)}, \tilde{\mathbf{y}}_i^{(t)}) - \nabla_{\mathbf{y}} f_i(\mathbf{x}_i^{(t)}, \mathbf{y}_i^{(t)})\right)\right\|_2^2 \\
&\quad + (1 + \tfrac{1}{b})\eta_{\mathbf{y}}^2\left\|\nabla_{\mathbf{y}} f_i(\mathbf{x}_i^{(t)}, \mathbf{y}_i^{(t)}) - \mathbf{d}_{\mathbf{y},i}^{(t)}\right\|_2^2,
\end{aligned}
\tag{113}
$$

where $b > 0$ is an arbitrary constant. By the $L$-smoothness and $\mu$-strong convexity of $-f_i(\mathbf{x}, \cdot)$, it holds for all $i \in [m]$ that

$$
\left\| \tilde{\mathbf{y}}_i^{(t)} - \mathbf{y}_i^{(t)} + \eta_{\mathbf{y}} \left( \nabla_{\mathbf{y}} f_i(\mathbf{x}_i^{(t)}, \tilde{\mathbf{y}}_i^{(t)}) - \nabla_{\mathbf{y}} f_i(\mathbf{x}_i^{(t)}, \mathbf{y}_i^{(t)}) \right) \right\|_2^2
$$

$$
= \left\| \tilde{\mathbf{y}}_i^{(t)} - \mathbf{y}_i^{(t)} \right\|_2^2 + 2\eta_{\mathbf{y}} \left\langle \tilde{\mathbf{y}}_i^{(t)} - \mathbf{y}_i^{(t)}, \nabla_{\mathbf{y}} f_i(\mathbf{x}_i^{(t)}, \tilde{\mathbf{y}}_i^{(t)}) - \nabla_{\mathbf{y}} f_i(\mathbf{x}_i^{(t)}, \mathbf{y}_i^{(t)}) \right\rangle
$$

$$
+ \eta_{\mathbf{y}}^2 \left\| \nabla_{\mathbf{y}} f_i(\mathbf{x}_i^{(t)}, \tilde{\mathbf{y}}_i^{(t)}) - \nabla_{\mathbf{y}} f_i(\mathbf{x}_i^{(t)}, \mathbf{y}_i^{(t)}) \right\|_2^2
$$

$$
\leq \left\| \tilde{\mathbf{y}}_i^{(t)} - \mathbf{y}_i^{(t)} \right\|_2^2 + \left( 2\eta_{\mathbf{y}} - \eta_{\mathbf{y}}^2 L \right) \left\langle \tilde{\mathbf{y}}_i^{(t)} - \mathbf{y}_i^{(t)}, \nabla_{\mathbf{y}} f_i(\mathbf{x}_i^{(t)}, \tilde{\mathbf{y}}_i^{(t)}) - \nabla_{\mathbf{y}} f_i(\mathbf{x}_i^{(t)}, \mathbf{y}_i^{(t)}) \right\rangle
$$

$$
\leq \left( 1 - 2\eta_{\mathbf{y}}\mu + \eta_{\mathbf{y}}^2 \mu L \right) \left\| \tilde{\mathbf{y}}_i^{(t)} - \mathbf{y}_i^{(t)} \right\|_2^2 \leq \left( 1 - \frac{7}{4}\eta_{\mathbf{y}}\mu \right) \left\| \tilde{\mathbf{y}}_i^{(t)} - \mathbf{y}_i^{(t)} \right\|_2^2, \tag{114}
$$

where the third to last inequality holds from $\left( 2\eta_{\mathbf{y}} - \eta_{\mathbf{y}}^2 L \right) > 0$ by $\eta_{\mathbf{y}} \leq \frac{1}{4L}$, the second to last inequality uses the strong concavity of $f_i(\mathbf{x}, \cdot)$ and the last uses $\eta_{\mathbf{y}} \leq \frac{1}{4L}$ to have $1 - 2\eta_{\mathbf{y}}\mu + \eta_{\mathbf{y}}^2 \mu L \leq 1 - \frac{7}{4}\eta_{\mathbf{y}}\mu$. Hence, by the Peter-Paul inequality and utilizing (114) within (113), we have

$$
\left\| \widetilde{\mathbf{Y}}^{(t+1)} - \mathbf{Y}^{(t+1)} \right\|_F^2
$$

$$
\leq (1+a) \left\| \widetilde{\mathbf{Y}}^{(t)} - \mathbf{Y}^{(t+1)} \right\|_F^2 + (1 + \tfrac{1}{a}) \left\| \widetilde{\mathbf{Y}}^{(t+1)} - \widetilde{\mathbf{Y}}^{(t)} \right\|_F^2
$$

$$
\leq (1+a)(1+b) \left( 1 - \tfrac{7}{4}\eta_{\mathbf{y}}\mu \right) \left\| \widetilde{\mathbf{Y}}^{(t)} - \mathbf{Y}^{(t)} \right\|_F^2 + (1+a)(1 + \tfrac{1}{b})\eta_{\mathbf{y}}^2 \left\| \mathbf{R}_{\mathbf{y}}^{(t)} \right\|_F^2
$$

$$
+ (1 + \frac{1}{a}) \left\| \widetilde{\mathbf{Y}}^{(t+1)} - \widetilde{\mathbf{Y}}^{(t)} \right\|_F^2
$$

$$
\overset{(32c)}{\leq} (1+a)(1+b) \left( 1 - \tfrac{7}{4}\eta_{\mathbf{y}}\mu \right) \left\| \widetilde{\mathbf{Y}}^{(t)} - \mathbf{Y}^{(t)} \right\|_F^2 + (1+a)(1 + \tfrac{1}{b})\eta_{\mathbf{y}}^2 \left\| \mathbf{R}_{\mathbf{y}}^{(t)} \right\|_F^2 \tag{115}
$$

$$
+ 2(1 + \tfrac{1}{a})\kappa^2 \left( \left\| \mathbf{X}^{(t+1)} - \mathbf{X}^{(t)} \right\|_F^2 + m \left\| \mathbf{\Lambda}^{(t+1)} - \mathbf{\Lambda}^{(t)} \right\|_F^2 \right),
$$

where $a > 0$ is an arbitrary constant. Setting $a = \frac{\eta_{\mathbf{y}}\mu}{2 - 3\eta_{\mathbf{y}}\mu} = \frac{1 - \eta_{\mathbf{y}}\mu}{1 - \frac{3}{2}\eta_{\mathbf{y}}\mu} - 1$ and $b = \frac{\frac{1}{4}\eta_{\mathbf{y}}\mu}{1 - \frac{7}{4}\eta_{\mathbf{y}}\mu} = \frac{1 - \frac{3}{2}\eta_{\mathbf{y}}\mu}{1 - \frac{7}{4}\eta_{\mathbf{y}}\mu} - 1$ in (115) and utilizing $\eta_{\mathbf{y}} \leq \frac{1}{4L}$, it holds that

$$
(1+a)(1+b) \left( 1 - \frac{7}{4}\eta_{\mathbf{y}}\mu \right) = 1 - \eta_{\mathbf{y}}\mu, \tag{116}
$$

$$
(1+a)(1 + \frac{1}{b})\eta_{\mathbf{y}}^2 = \left( \frac{1 - \eta_{\mathbf{y}}\mu}{1 - \frac{3}{2}\eta_{\mathbf{y}}\mu} \right) \left( \frac{1 - \frac{3}{2}\eta_{\mathbf{y}}\mu}{\frac{1}{4}\eta_{\mathbf{y}}\mu} \right) \eta_{\mathbf{y}}^2 = \frac{4\eta_{\mathbf{y}} - 4\eta_{\mathbf{y}}^2 \mu}{\mu} \leq \frac{4\eta_{\mathbf{y}}}{\mu}, \tag{117}
$$

$$
1 + \frac{1}{a} = \frac{2(1 - \eta_{\mathbf{y}}\mu)}{\eta_{\mathbf{y}}\mu} \leq \frac{2}{\eta_{\mathbf{y}}\mu}. \tag{118}
$$

Applying the bounds in (116)-(118) to (115) and utilizing $\left\| \mathbf{R}_{\mathbf{y}}^{(t)} \right\|_F^2 \leq \left\| \mathbf{R}^{(t)} \right\|_F^2$ completes the proof. $\qquad \square$

## G   Proofs of Lemmas in Section C.4

**Proof of Lemma C.7** We first prove (46), for which we break the proof into two cases. For the first case, we assume $t = n_t q$, i.e., $t$ is divisible by $q$. Then for all $i \in [m]$, we have by the definition of $\Upsilon$ that

$$
\mathbb{E} \left\| \mathbf{r}_i^{(t)} \right\|_2^2 = \mathbb{E} \left\| \nabla f_i(\mathbf{x}_i^{(t)}, \mathbf{y}_i^{(t)}) - \mathbf{d}_i^{(t)} \right\|_2^2 \overset{(12)}{=} \mathbb{E} \left[ \mathbb{E}_{\tilde{\mathcal{B}}_i^{(t)}} \left\| \nabla f_i(\mathbf{x}_i^{(t)}, \mathbf{y}_i^{(t)}) - G_i^{(t)}(\tilde{\mathcal{B}}_i^{(t)}) \right\|_2^2 \right] \leq \Upsilon, \tag{119}
$$

where the second equality uses the law of total expectation and the inequality uses Assumption 1(ii) and $|\tilde{\mathcal{B}}_i^{(t)}| = \mathcal{S}_1$. Next, we assume $n_t q < t < (n_t + 1)q$ and compute

$$
\begin{aligned}
\mathbb{E}\left\|\mathbf{r}_i^{(t)}\right\|_2^2 &= \mathbb{E}\left\|\nabla f_i(\mathbf{x}_i^{(t)}, \mathbf{y}_i^{(t)}) - \mathbf{d}_i^{(t)}\right\|_2^2 \overset{(12)}{=} \mathbb{E}\left\|\nabla f_i(\mathbf{x}_i^{(t)}, \mathbf{y}_i^{(t)}) - G_i^{(t)}(\mathcal{B}_i^{(t)}) + G_i^{(t-1)}(\mathcal{B}_i^{(t)}) - \mathbf{d}_i^{(t-1)}\right\|_2^2 \\
&\leq \frac{L^2}{\mathcal{S}_2}\left(\mathbb{E}\left\|\mathbf{x}_i^{(t)} - \mathbf{x}_i^{(t-1)}\right\|_2^2 + \mathbb{E}\left\|\mathbf{y}_i^{(t)} - \mathbf{y}_i^{(t-1)}\right\|_2^2\right) + \mathbb{E}\left\|\mathbf{r}_i^{(t-1)}\right\|_2^2,
\end{aligned}
\tag{120}
$$

where the inequality comes from Eqn. (A.4) in Fang et al. (2018). Recursively utilizing (120) for $n_t q < t < (n_t + 1)q$ and using (119) with $t = n_t q$ yields

$$
\mathbb{E}\left\|\mathbf{r}_i^{(t)}\right\|_2^2 \leq \frac{L^2}{\mathcal{S}_2}\sum_{r=n_t q}^{t-1}\left(\mathbb{E}\left\|\mathbf{x}_i^{(r+1)} - \mathbf{x}_i^{(r)}\right\|_2^2 + \mathbb{E}\left\|\mathbf{y}_i^{(r+1)} - \mathbf{y}_i^{(r)}\right\|_2^2\right) + \Upsilon.
\tag{121}
$$

Summing up (121) over $i = 1, \ldots, m$ proves (46).

For (45), we utilize the technique from Lemma C.7 in Mancino-Ball et al. (2023a) to have

$$
\left\|\mathbf{V}_{\perp,\mathbf{x}}^{(t+1)}\right\|_F^2 \leq \rho\left\|\mathbf{V}_{\perp,\mathbf{x}}^{(t)}\right\|_F^2 + \frac{1}{1-\rho}\left\|\mathbf{D}_{\mathbf{x}}^{(t+1)} - \mathbf{D}_{\mathbf{x}}^{(t)}\right\|_F^2.
\tag{122}
$$

By $\left\|\mathbf{d}_{\mathbf{x},i}^{(t+1)} - \mathbf{d}_{\mathbf{x},i}^{(t)}\right\|_2^2 \leq \left\|\mathbf{d}_i^{(t+1)} - \mathbf{d}_i^{(t)}\right\|_2^2, \forall i \in [m]$, we use Young's inequality to have

$$
\begin{aligned}
&\left\|\mathbf{D}_{\mathbf{x}}^{(t+1)} - \mathbf{D}_{\mathbf{x}}^{(t)}\right\|_F^2 \leq \left\|\mathbf{D}^{(t+1)} - \mathbf{D}^{(t)}\right\|_F^2 \\
&\leq 3\left\|\mathbf{D}^{(t+1)} - \nabla F(\mathbf{Z}^{(t+1)})\right\|_F^2 + 3\left\|\nabla F(\mathbf{Z}^{(t+1)}) - \nabla F(\mathbf{Z}^{(t)})\right\|_F^2 + 3\left\|\nabla F(\mathbf{Z}^{(t)}) - \mathbf{D}^{(t)}\right\|_F^2 \\
&\leq 3\left\|\mathbf{R}^{(t+1)}\right\|_F^2 + 3L^2\left\|\mathbf{Z}^{(t+1)} - \mathbf{Z}^{(t)}\right\|_F^2 + 3\left\|\mathbf{R}^{(t)}\right\|_F^2,
\end{aligned}
\tag{123}
$$

where the last inequality further uses Assumption 1. Take the expectation of (123) and utilize (46) to have

$$
\begin{aligned}
&\mathbb{E}\left\|\mathbf{D}_{\mathbf{x}}^{(t+1)} - \mathbf{D}_{\mathbf{x}}^{(t)}\right\|_F^2 \\
&\leq 3L^2\mathbb{E}\left\|\mathbf{Z}^{(t+1)} - \mathbf{Z}^{(t)}\right\|_F^2 + \frac{3L^2}{\mathcal{S}_2}\sum_{r=(n_{t+1})q}^{t}\mathbb{E}\left\|\mathbf{Z}^{(r+1)} - \mathbf{Z}^{(r)}\right\|_F^2 + 3m\Upsilon \\
&\quad + \frac{3L^2}{\mathcal{S}_2}\sum_{r=n_t q}^{t-1}\mathbb{E}\left\|\mathbf{Z}^{(r+1)} - \mathbf{Z}^{(r)}\right\|_F^2 + 3m\Upsilon.
\end{aligned}
\tag{124}
$$

Applying (124) to the expectation of (122) and further using the non-negativity of the 2-norm completes the proof. $\qquad\square$

**Proof of Lemma C.8** With constants in (49) and $L_P = L\sqrt{4\kappa^2 + 1} \leq L(2\kappa + 1)$, (42) implies

$$
\begin{aligned}
&\sum_{t=0}^{T-1}\mathbb{E}\left\|\mathbf{Y}^{(t+1)} - \mathbf{Y}^{(t)}\right\|_F^2 \\
&\leq \mathbb{E}\left[\frac{m}{L}(\hat{\delta}_0 - \hat{\delta}_T)\right] + \frac{1}{4L^2}\sum_{t=0}^{T-1}\mathbb{E}\left\|\mathbf{R}^{(t)}\right\|_F^2 + \frac{1}{16\kappa^2}\sum_{t=0}^{T-1}\mathbb{E}\left\|\widetilde{\mathbf{Y}}^{(t+1)} - \mathbf{Y}^{(t+1)}\right\|_F^2 \\
&\quad + (8\kappa^2 + \kappa + 1)\sum_{t=0}^{T-1}\mathbb{E}\left\|\mathbf{X}^{(t+1)} - \mathbf{X}^{(t)}\right\|_F^2 + m(8\kappa^2 + \kappa + 1)\sum_{t=0}^{T-1}\mathbb{E}\left\|\mathbf{\Lambda}^{(t+1)} - \mathbf{\Lambda}^{(t)}\right\|_F^2 \\
&\overset{(46),(48)}{\leq} \frac{m}{L}\hat{\delta}_0 + \frac{1}{16\kappa^2}\sum_{t=0}^{T-1}\mathbb{E}\left\|\widetilde{\mathbf{Y}}^{(t+1)} - \mathbf{Y}^{(t+1)}\right\|_F^2 + (8\kappa^2 + \kappa + 1)\sum_{t=0}^{T-1}\mathbb{E}\left\|\mathbf{X}^{(t+1)} - \mathbf{X}^{(t)}\right\|_F^2 \\
&\quad + m(8\kappa^2 + \kappa + 1)\sum_{t=0}^{T-1}\mathbb{E}\left\|\mathbf{\Lambda}^{(t+1)} - \mathbf{\Lambda}^{(t)}\right\|_F^2 + \frac{1}{4}\sum_{t=0}^{T-1}\mathbb{E}\left\|\mathbf{Z}^{(t+1)} - \mathbf{Z}^{(t)}\right\|_F^2 + \frac{m\Upsilon T}{4L^2},
\end{aligned}
$$

where we have used the fact $\hat{\delta}_T \geq 0$. Since $\mathbf{Z}^{(t)} = (\mathbf{X}^{(t)}, \mathbf{Y}^{(t)})$, the inequality above clearly indicates

$$\sum_{t=0}^{T-1} \mathbb{E} \left\| \mathbf{Y}^{(t+1)} - \mathbf{Y}^{(t)} \right\|_F^2 \leq \left( \frac{4}{3}(8\kappa^2 + \kappa + 1) + \frac{1}{3} \right) \sum_{t=0}^{T-1} \mathbb{E} \left\| \mathbf{X}^{(t+1)} - \mathbf{X}^{(t)} \right\|_F^2 \tag{125}$$

$$+ \frac{1}{12\kappa^2} \sum_{t=0}^{T-1} \mathbb{E} \left\| \widetilde{\mathbf{Y}}^{(t+1)} - \mathbf{Y}^{(t+1)} \right\|_F^2 + \frac{4}{3}m(8\kappa^2 + \kappa + 1) \sum_{t=0}^{T-1} \mathbb{E} \left\| \mathbf{\Lambda}^{(t+1)} - \mathbf{\Lambda}^{(t)} \right\|_F^2 + \frac{4m}{3L}\hat{\delta}_0 + \frac{m\Upsilon T}{3L^2}.$$

On the other hand, summing up (44) and using (46) and (48), we have

$$\frac{1}{4\kappa} \sum_{t=0}^{T-1} \mathbb{E} \left\| \widetilde{\mathbf{Y}}^{(t+1)} - \mathbf{Y}^{(t+1)} \right\|_F^2$$

$$\leq \left( 1 - \frac{1}{4\kappa} \right) \left\| \widetilde{\mathbf{Y}}^{(0)} - \mathbf{Y}^{(0)} \right\|_F^2 + \frac{Tm\Upsilon}{L\mu} \tag{126}$$

$$+ (16\kappa^3 + \kappa) \sum_{t=0}^{T-1} \mathbb{E} \left\| \mathbf{X}^{(t+1)} - \mathbf{X}^{(t)} \right\|_F^2 + 16m\kappa^3 \sum_{t=0}^{T-1} \mathbb{E} \left\| \mathbf{\Lambda}^{(t+1)} - \mathbf{\Lambda}^{(t)} \right\|_F^2$$

$$+ \kappa \sum_{t=0}^{T-1} \mathbb{E} \left\| \mathbf{Y}^{(t+1)} - \mathbf{Y}^{(t)} \right\|_F^2.$$

Utilize (125) within (126), solve it for $\sum_{t=0}^{T-1} \mathbb{E}\|\widetilde{\mathbf{Y}}^{(t+1)} - \mathbf{Y}^{(t+1)}\|_F^2$, and bound $\|\mathbf{X}^{(t+1)} - \mathbf{X}^{(t)}\|_F^2$ by (31). We obtain (50). Now substitute (50) into (125), use (31) again, and combine like terms to have (51) and complete the proof. $\qquad\square$

## H  Proofs of Lemmas in Section C.5

**Proof of Lemma C.9** For (72), we use (122). When VR-tag == STORM in Algorithm 1, it holds

$$\left\| \mathbf{d}_{\mathbf{x},i}^{(t+1)} - \mathbf{d}_{\mathbf{x},i}^{(t)} \right\|_2^2 \leq \left\| \mathbf{d}_i^{(t+1)} - \mathbf{d}_i^{(t)} \right\|_2^2$$

$$= \left\| G_i^{t+1}(\mathcal{B}_i^{(t+1)}) + (1 - \beta) \left( \mathbf{d}_i^{(t)} - G_i^{(t)}(\mathcal{B}_i^{(t+1)}) \right) - \mathbf{d}_i^{(t)} \right\|_2^2 \tag{127}$$

$$= \left\| G_i^{t+1}(\mathcal{B}_i^{(t+1)}) - G_i^{(t)}(\mathcal{B}_i^{(t+1)}) + \beta \left( G_i^{(t)}(\mathcal{B}_i^{(t+1)}) - \nabla f_i(\mathbf{x}_i^{(t)}, \mathbf{y}_i^{(t)}) \right) + \beta \left( \nabla f_i(\mathbf{x}_i^{(t)}, \mathbf{y}_i^{(t)}) - \mathbf{d}_i^{(t)} \right) \right\|_2^2.$$

Using Young's inequality and taking the expectation with respect to the samples $\mathcal{B}_i^{(t+1)}$ and then the full expectation yields

$$\mathbb{E} \left\| \mathbf{d}_{\mathbf{x},i}^{(t+1)} - \mathbf{d}_{\mathbf{x},i}^{(t)} \right\|_2^2 \leq 3L^2 \mathbb{E} \left\| \mathbf{x}_i^{(t+1)} - \mathbf{x}_i^{(t)} \right\|_2^2 + 3L^2 \mathbb{E} \left\| \mathbf{y}_i^{(t+1)} - \mathbf{y}_i^{(t)} \right\|_2^2 + 3\beta^2 \Upsilon_{t+1} + 3\beta^2 \mathbb{E} \left\| \mathbf{r}_i^{(t)} \right\|_2^2, \tag{128}$$

where $(\mathbf{r}_i^{(t)})^\top$ is defined as the $i$-th row of $\mathbf{R}^{(t)}$ for all $t \geq 0$ and we have used Assumption 1(ii). Taking the full expectation of (122) and applying (127) with (128) summed over $i = 1, \ldots, m$ yields (72). The proof of (73) follows from the same arguments of the proof of (Mancino-Ball et al., 2023a, Lemma C.9). $\qquad\square$

**Proof of Lemma C.10** We first take the expectation of (38), apply (41), and plug in the values of $c_1, c_2$, and $c_3$ specified in (74) to have

$$\mathbb{E} \left[ \phi(\bar{\mathbf{x}}^{(t+1)}, \mathbf{\Lambda}^{(t+1)}) - \phi(\bar{\mathbf{x}}^{(t)}, \mathbf{\Lambda}^{(t)}) \right] \leq - \frac{1}{4m\eta_{\mathbf{x}}} \mathbb{E} \left\| \mathbf{X}^{(t+1)} - \bar{\mathbf{X}}^{(t)} \right\|_F^2 - \frac{1}{2m\eta_{\mathbf{x}}} \mathbb{E} \left\| \mathbf{X}^{(t+1)} - \widetilde{\mathbf{X}}^{(t)} \right\|_F^2$$

$$- \left( \frac{1}{\eta_{\mathbf{\Lambda}}} - \frac{L_P}{2} - \frac{32\kappa^2 L}{\sqrt{\beta}} \right) \mathbb{E} \left\| \mathbf{\Lambda}^{(t+1)} - \mathbf{\Lambda}^{(t)} \right\|_F^2 + \frac{1}{2m} \left( L(\kappa + 1) + \frac{\sqrt{\beta}}{32} + \frac{\rho^2}{\eta_{\mathbf{x}}} \right) \mathbb{E} \left\| \mathbf{X}_\perp^{(t)} \right\|_F^2 \tag{129}$$

$$+ \frac{\sqrt{\beta}(L+1)}{64m\kappa^2} \mathbb{E} \left\| \widetilde{\mathbf{Y}}^{(t)} - \mathbf{Y}^{(t)} \right\|_F^2 + \frac{\sqrt{\beta}}{60mL_P} \mathbb{E} \left( \left\| \mathbf{R}^{(t)} \right\|_F^2 + \left\| \mathbf{V}_{\perp,\mathbf{x}}^{(t)} \right\|_F^2 \right),$$

where the coefficient of $\mathbb{E}\left\|\mathbf{X}^{(t+1)} - \bar{\mathbf{X}}^{(t)}\right\|_F^2$ is obtained by using $L_P = L\sqrt{4\kappa^2+1} \le L(2\kappa+1)$ and $\sqrt{\beta} < 1$ to have

$$L\left(\kappa + 1 + \frac{32\kappa^2}{\sqrt{\beta}}\right) + L_P\left(1 + \frac{60}{\sqrt{\beta}}\right) \le \frac{32\kappa^2 L + 123\kappa L + 62L}{\sqrt{\beta}} \le \frac{1}{2\eta_{\mathbf{x}}}.$$

Next, for any positive $\gamma_1, \gamma_2, \gamma_3,$ and $\gamma_4$, we add $\gamma_1 \mathbb{E}\left\|\mathbf{X}_\perp^{(t+1)}\right\|_F^2$, $\gamma_2 \mathbb{E}\left\|\mathbf{V}_{\perp,\mathbf{x}}^{(t+1)}\right\|_F^2$, $\gamma_3 \mathbb{E}\left\|\mathbf{R}^{(t+1)}\right\|_F^2$, and $\gamma_4 \mathbb{E}\left\|\widetilde{\mathbf{Y}}^{(t+1)} - \mathbf{Y}^{(t+1)}\right\|_F^2$ to both sides of (129) and upper bound the right-hand side using Lemmas C.3, C.6, and C.9 to obtain

$$
\begin{aligned}
&\mathbb{E}\left[\phi(\bar{\mathbf{x}}^{(t+1)}, \mathbf{\Lambda}^{(t+1)}) - \phi(\bar{\mathbf{x}}^{(t)}, \mathbf{\Lambda}^{(t)})\right] + \gamma_1 \mathbb{E}\left\|\mathbf{X}_\perp^{(t+1)}\right\|_F^2 + \gamma_2 \mathbb{E}\left\|\mathbf{V}_{\perp,\mathbf{x}}^{(t+1)}\right\|_F^2 \\
&\quad + \gamma_3 \mathbb{E}\left\|\mathbf{R}^{(t+1)}\right\|_F^2 + \gamma_4 \mathbb{E}\left\|\widetilde{\mathbf{Y}}^{(t+1)} - \mathbf{Y}^{(t+1)}\right\|_F^2 \\
&\le -\frac{1}{4m\eta_{\mathbf{x}}}\mathbb{E}\left\|\mathbf{X}^{(t+1)} - \bar{\mathbf{X}}^{(t)}\right\|_F^2 - \frac{1}{2m\eta_{\mathbf{x}}}\mathbb{E}\left\|\mathbf{X}^{(t+1)} - \widetilde{\mathbf{X}}^{(t)}\right\|_F^2 + \left(\frac{3\gamma_2}{1-\rho} + 2\gamma_3\right)m\beta^2\Upsilon_{t+1} \\
&\quad - \left(\frac{1}{\eta_{\mathbf{\Lambda}}} - \frac{L_P}{2} - \frac{32\kappa^2 L}{\sqrt{\beta}} - \frac{16\sqrt{2}\kappa^3 m\gamma_4}{\sqrt{\beta}}\right)\mathbb{E}\left\|\mathbf{\Lambda}^{(t+1)} - \mathbf{\Lambda}^{(t)}\right\|_F^2 \\
&\quad + \frac{1}{2m}\left(L(\kappa+1) + \frac{\sqrt{\beta}}{32} + \frac{\rho^2}{\eta_{\mathbf{x}}} + 2m\rho\gamma_1\right)\mathbb{E}\left\|\mathbf{X}_\perp^{(t)}\right\|_F^2 + \left(\rho\gamma_2 + \frac{\gamma_1\eta_{\mathbf{x}}^2}{1-\rho} + \frac{\sqrt{\beta}}{60mL_P}\right)\mathbb{E}\left\|\mathbf{V}_{\perp,\mathbf{x}}^{(t)}\right\|_F^2 \\
&\quad + \left(\gamma_4\left(1 - \frac{\sqrt{\beta}}{4\sqrt{2}\kappa}\right) + \frac{\sqrt{\beta}(L+1)}{64m\kappa^2}\right)\mathbb{E}\left\|\widetilde{\mathbf{Y}}^{(t)} - \mathbf{Y}^{(t)}\right\|_F^2 + \frac{16\sqrt{2}\kappa^3\gamma_4}{\sqrt{\beta}}\mathbb{E}\left\|\mathbf{X}^{(t+1)} - \mathbf{X}^{(t)}\right\|_F^2 \\
&\quad + \left(\frac{3L^2\gamma_2}{1-\rho} + 2L^2(1-\beta)^2\gamma_3\right)\mathbb{E}\left\|\mathbf{Z}^{(t+1)} - \mathbf{Z}^{(t)}\right\|_F^2 \\
&\quad + \left(\gamma_3(1-\beta)^2 + \frac{3\beta^2\gamma_2}{1-\rho} + \frac{\sqrt{\beta}}{60mL_P} + \frac{\sqrt{\beta}\gamma_4}{\sqrt{2}L\mu}\right)\mathbb{E}\left\|\mathbf{R}^{(t)}\right\|_F^2.
\end{aligned}
\tag{130}
$$

We apply $\left\|\mathbf{Z}^{(t+1)} - \mathbf{Z}^{(t)}\right\|_F^2 = \left\|\mathbf{X}^{(t+1)} - \mathbf{X}^{(t)}\right\|_F^2 + \left\|\mathbf{Y}^{(t+1)} - \mathbf{Y}^{(t)}\right\|_F^2$ and (42) to (130) to have

$$
\begin{aligned}
&\mathbb{E}\left[\phi(\bar{\mathbf{x}}^{(t+1)}, \mathbf{\Lambda}^{(t+1)}) - \phi(\bar{\mathbf{x}}^{(t)}, \mathbf{\Lambda}^{(t)})\right] + \gamma_1 \mathbb{E}\left\|\mathbf{X}_\perp^{(t+1)}\right\|_F^2 + \gamma_2 \mathbb{E}\left\|\mathbf{V}_{\perp,\mathbf{x}}^{(t+1)}\right\|_F^2 \\
&\quad + \gamma_3 \mathbb{E}\left\|\mathbf{R}^{(t+1)}\right\|_F^2 + \left(\gamma_4 - \frac{\beta}{60\kappa^2}\left(\frac{3L^2\gamma_2}{1-\rho} + 2L^2(1-\beta)^2\gamma_3\right)\right)\mathbb{E}\left\|\widetilde{\mathbf{Y}}^{(t+1)} - \mathbf{Y}^{(t+1)}\right\|_F^2 \\
&\le -\frac{1}{4m\eta_{\mathbf{x}}}\mathbb{E}\left\|\mathbf{X}^{(t+1)} - \bar{\mathbf{X}}^{(t)}\right\|_F^2 - \frac{1}{2m\eta_{\mathbf{x}}}\mathbb{E}\left\|\mathbf{X}^{(t+1)} - \widetilde{\mathbf{X}}^{(t)}\right\|_F^2 + \left(\frac{3\gamma_2}{1-\rho} + 2\gamma_3\right)m\beta^2\Upsilon_{t+1} \\
&\quad - \left(\frac{1}{\eta_{\mathbf{\Lambda}}} - \frac{L_P}{2} - \frac{32\kappa^2 L}{\sqrt{\beta}} - \frac{16\sqrt{2}\kappa^3 m\gamma_4}{\sqrt{\beta}}\right. \\
&\qquad\qquad \left. - \frac{m}{2}\left(\frac{\sqrt{\beta}}{\sqrt{2}}\sqrt{4\kappa^2+1} + 30\kappa^2\right)\left(\frac{3L^2\gamma_2}{1-\rho} + 2L^2(1-\beta)^2\gamma_3\right)\right)\mathbb{E}\left\|\mathbf{\Lambda}^{(t+1)} - \mathbf{\Lambda}^{(t)}\right\|_F^2 \\
&\quad + \frac{1}{2m}\left(L(\kappa+1) + \frac{\sqrt{\beta}}{32} + \frac{\rho^2}{\eta_{\mathbf{x}}} + 2m\rho\gamma_1\right)\mathbb{E}\left\|\mathbf{X}_\perp^{(t)}\right\|_F^2 + \left(\rho\gamma_2 + \frac{\gamma_1\eta_{\mathbf{x}}^2}{1-\rho} + \frac{\sqrt{\beta}}{60mL_P}\right)\mathbb{E}\left\|\mathbf{V}_{\perp,\mathbf{x}}^{(t)}\right\|_F^2 \\
&\quad + \left(\gamma_4\left(1 - \frac{\sqrt{\beta}}{4\sqrt{2}\kappa}\right) + \frac{\sqrt{\beta}(L+1)}{64m\kappa^2}\right)\mathbb{E}\left\|\widetilde{\mathbf{Y}}^{(t)} - \mathbf{Y}^{(t)}\right\|_F^2 + \frac{m\sqrt{\beta}}{\sqrt{2}L}\left(\frac{3L^2\gamma_2}{1-\rho} + 2L^2(1-\beta)^2\gamma_3\right)\mathbb{E}\left[\hat{\delta}_t - \hat{\delta}_{t+1}\right] \\
&\quad + \left(\frac{16\sqrt{2}\kappa^3\gamma_4}{\sqrt{\beta}} + \left(\frac{\sqrt{\beta}}{2\sqrt{2}}(\sqrt{4\kappa^2+1}+1) + 15\kappa^2+1\right)\left(\frac{3L^2\gamma_2}{1-\rho} + 2L^2(1-\beta)^2\gamma_3\right)\right)\mathbb{E}\left\|\mathbf{X}^{(t+1)} - \mathbf{X}^{(t)}\right\|_F^2 \\
&\quad + \left(\gamma_3(1-\beta)^2 + \frac{3\beta^2\gamma_2}{1-\rho} + \frac{\sqrt{\beta}}{60mL_P} + \frac{\sqrt{\beta}\gamma_4}{\sqrt{2}L\mu} + \frac{\beta}{8L^2}\left(\frac{3L^2\gamma_2}{1-\rho} + 2L^2(1-\beta)^2\gamma_3\right)\right)\mathbb{E}\left\|\mathbf{R}^{(t)}\right\|_F^2.
\end{aligned}
\tag{131}
$$

Next, we utilize (31) to bound $\mathbb{E}\left\|\mathbf{X}^{(t+1)} - \mathbf{X}^{(t)}\right\|_F^2$. Then we subtract $\gamma_1 \mathbb{E}\left\|\mathbf{X}_\perp^{(t)}\right\|_F^2$, $\gamma_2 \mathbb{E}\left\|\mathbf{V}_{\perp,\mathbf{x}}^{(t)}\right\|_F^2$, $\gamma_3 \mathbb{E}\left\|\mathbf{R}^{(t)}\right\|_F^2$ from both sides of (131) with

$$c_{\mathbf{x}} := \frac{16\sqrt{2}\kappa^3\gamma_4}{\sqrt{\beta}} + \left(\frac{\sqrt{\beta}}{2\sqrt{2}}(\sqrt{4\kappa^2+1}+1) + 15\kappa^2 + 1\right)\left(\frac{3L^2\gamma_2}{1-\rho} + 2L^2(1-\beta)^2\gamma_3\right) \tag{132}$$

to have

$$
\begin{aligned}
&\mathbb{E}\left[\phi(\bar{\mathbf{x}}^{(t+1)}, \mathbf{\Lambda}^{(t+1)}) - \phi(\bar{\mathbf{x}}^{(t)}, \mathbf{\Lambda}^{(t)})\right] + \gamma_1\mathbb{E}\left\|\mathbf{X}_\perp^{(t+1)}\right\|_F^2 - \gamma_1\mathbb{E}\left\|\mathbf{X}_\perp^{(t)}\right\|_F^2 \\
&\quad + \gamma_2\mathbb{E}\left\|\mathbf{V}_{\perp,\mathbf{x}}^{(t+1)}\right\|_F^2 - \gamma_2\mathbb{E}\left\|\mathbf{V}_{\perp,\mathbf{x}}^{(t)}\right\|_F^2 + \gamma_3\mathbb{E}\left\|\mathbf{R}^{(t+1)}\right\|_F^2 - \gamma_3\mathbb{E}\left\|\mathbf{R}^{(t)}\right\|_F^2 \\
&\quad + \left(\gamma_4 - \frac{\beta}{60\kappa^2}\left(\frac{3L^2\gamma_2}{1-\rho} + 2L^2(1-\beta)^2\gamma_3\right)\right)\mathbb{E}\left\|\widetilde{\mathbf{Y}}^{(t+1)} - \mathbf{Y}^{(t+1)}\right\|_F^2 \\
&\leq -\frac{1}{4m\eta_{\mathbf{x}}}\mathbb{E}\left\|\mathbf{X}^{(t+1)} - \bar{\mathbf{X}}^{(t)}\right\|_F^2 - \left(\frac{1}{2m\eta_{\mathbf{x}}} - 2c_{\mathbf{x}}\right)\mathbb{E}\left\|\mathbf{X}^{(t+1)} - \widetilde{\mathbf{X}}^{(t)}\right\|_F^2 + \left(\frac{3\gamma_2}{1-\rho} + 2\gamma_3\right)m\beta^2\Upsilon_{t+1} \\
&\quad - \left(\frac{1}{\eta_{\mathbf{\Lambda}}} - \frac{L_P}{2} - \frac{32\kappa^2 L}{\sqrt{\beta}} - \frac{16\sqrt{2}\kappa^3 m\gamma_4}{\sqrt{\beta}}\right. \\
&\qquad\qquad \left. - \frac{m}{2}\left(\frac{\sqrt{\beta}}{\sqrt{2}}\sqrt{4\kappa^2+1} + 30\kappa^2\right)\left(\frac{3L^2\gamma_2}{1-\rho} + 2L^2(1-\beta)^2\gamma_3\right)\right)\mathbb{E}\left\|\mathbf{\Lambda}^{(t+1)} - \mathbf{\Lambda}^{(t)}\right\|_F^2 \\
&\quad - \frac{1}{2m}\left(2m(1-\rho)\gamma_1 - L(\kappa+1) - \frac{\sqrt{\beta}}{32} - \frac{\rho^2}{\eta_{\mathbf{x}}} - 16mc_{\mathbf{x}}\right)\mathbb{E}\left\|\mathbf{X}_\perp^{(t)}\right\|_F^2 \\
&\quad - \left((1-\rho)\gamma_2 - \frac{\gamma_1\eta_{\mathbf{x}}^2}{1-\rho} - \frac{\sqrt{\beta}}{60mL_P}\right)\mathbb{E}\left\|\mathbf{V}_{\perp,\mathbf{x}}^{(t)}\right\|_F^2 \\
&\quad + \left(\gamma_4\left(1 - \frac{\sqrt{\beta}}{4\sqrt{2}\kappa}\right) + \frac{\sqrt{\beta}(L+1)}{64m\kappa^2}\right)\mathbb{E}\left\|\widetilde{\mathbf{Y}}^{(t)} - \mathbf{Y}^{(t)}\right\|_F^2 + \frac{m\sqrt{\beta}}{\sqrt{2}L}\left(\frac{3L^2\gamma_2}{1-\rho} + 2L^2(1-\beta)^2\gamma_3\right)\mathbb{E}\left[\hat{\delta}_t - \hat{\delta}_{t+1}\right] \\
&\quad - \left((1-(1-\beta)^2)\gamma_3 - \frac{3\beta^2\gamma_2}{1-\rho} - \frac{\sqrt{\beta}}{60mL_P} - \frac{\sqrt{\beta}\gamma_4}{\sqrt{2}L\mu} - \frac{\beta}{8L^2}\left(\frac{3L^2\gamma_2}{1-\rho} + 2L^2(1-\beta)^2\gamma_3\right)\right)\mathbb{E}\left\|\mathbf{R}^{(t)}\right\|_F^2.
\end{aligned} \tag{133}
$$

Set

$$\gamma_1 = \frac{2}{m(1-\rho)}\left((24\kappa^2 + 7\kappa + 4)\left(\frac{16(L+1)}{\sqrt{\beta}} + \frac{8L}{\kappa(1-\rho)^2}\right) + L(\kappa+1) + \frac{\sqrt{\beta}}{32} + \frac{1}{\eta_{\mathbf{x}}}\right), \tag{134a}$$

$$\gamma_2 = \frac{2}{1-\rho}\left(\frac{\sqrt{\beta}}{60mL_P} + \frac{\gamma_1\eta_{\mathbf{x}}^2}{1-\rho}\right), \tag{134b}$$

$$\gamma_3 = \frac{2}{\beta}\left(\frac{\sqrt{\beta}}{60mL_P} + \frac{\sqrt{\beta}\gamma_4}{\sqrt{2}L\mu} + \frac{3\gamma_2\beta(\beta+1/8)}{1-\rho}\right), \tag{134c}$$

$$\gamma_4 = \frac{\sqrt{2}(L+1)}{8m\kappa} + \frac{4L^2\sqrt{\beta}}{15\sqrt{2}\kappa}\left(\frac{9\gamma_2}{2(1-\rho)} + \frac{12\gamma_2\beta}{1-\rho} + \frac{1}{15\sqrt{\beta}mL_P}\right). \tag{134d}$$

By the definition of $\eta_{\mathbf{x}}$ and $\beta < 1$, we can easily have $(24\kappa^2 + 7\kappa + 4)\left(\frac{16(L+1)}{\sqrt{\beta}} + \frac{8L}{\kappa(1-\rho)^2}\right) + L(\kappa+1) + \frac{\sqrt{\beta}}{32} \leq \frac{1}{\eta_{\mathbf{x}}}$ and thus $\gamma_1 \leq \frac{4}{m(1-\rho)\eta_{\mathbf{x}}}$. Then by the choice of $\gamma_2$, $L_P \geq 2\kappa L$, and $\beta < 1$, it follows that

$$\gamma_2 \leq \frac{1}{60m(1-\rho)\kappa L} + \frac{8\eta_{\mathbf{x}}}{m(1-\rho)^3} \leq \frac{1}{40m(1-\rho)\kappa L}, \tag{135}$$

where the last inequality uses $\eta_{\mathbf{x}} \leq \frac{(1-\rho)^2}{960\kappa L}$. Hence by the choice of $\gamma_4$, $L_P \geq 2\kappa L$, $\beta < 1$, and (135), we additionally have

$$\gamma_4 \leq \frac{L+1}{5m\kappa} + \frac{\sqrt{\beta}L}{10m\kappa^2(1-\rho)^2}. \tag{136}$$

Moreover, applying $\beta < 1$, $L_P \geq 2\kappa L$, (135), and (136) to (134c) yields $\gamma_3 \leq \frac{1+1/L}{2\sqrt{\beta}mL} + \frac{2}{5m\kappa L(1-\rho)^2}$. Thus by the upper bounds of $\gamma_2$ and $\gamma_3$ together with $1 - \beta < 1$ yields

$$\frac{3L^2\gamma_2}{1-\rho} + 2L^2(1-\beta)^2\gamma_3 \leq \frac{L}{m(1-\rho)^2\kappa} + \frac{L+1}{\sqrt{\beta}m}. \tag{137}$$

Hence, from the definition of $c_{\mathbf{x}}$ in (132) and (136), (137), it follows

$$
\begin{aligned}
c_{\mathbf{x}} &\leq \frac{16\sqrt{2}\kappa^3}{\sqrt{\beta}}\left(\frac{L+1}{5m\kappa} + \frac{\sqrt{\beta}L}{10m\kappa^2(1-\rho)^2}\right) + \left(\frac{\sqrt{\beta}}{2\sqrt{2}}(\sqrt{4\kappa^2+1}+1) + 15\kappa^2 + 1\right)\left(\frac{L}{m(1-\rho)^2\kappa} + \frac{L+1}{\sqrt{\beta}m}\right) \\
&\leq \frac{L+1}{m\sqrt{\beta}}\left(\frac{16\sqrt{2}\kappa^2}{5} + 15\kappa^2 + 1 + \frac{\kappa+1}{\sqrt{2}}\right) + \frac{L}{m(1-\rho)^2\kappa}\left(\frac{16\sqrt{2}\kappa^2}{10} + 15\kappa^2 + 1 + \frac{\kappa+1}{\sqrt{2}}\right) \leq \frac{1}{8m\eta_{\mathbf{x}}},
\end{aligned} \tag{138}
$$

where the last inequality follows from the definition of $\eta_{\mathbf{x}}$. In addition, by $\beta < 1$, $\sqrt{4\kappa^2+1} \leq 2\kappa+1$, (136), and (137), we have

$$
\begin{aligned}
&\frac{16\sqrt{2}\kappa^3 m\gamma_4}{\sqrt{\beta}} + \frac{m}{2}\left(\frac{\sqrt{\beta}}{\sqrt{2}}\sqrt{4\kappa^2+1} + 30\kappa^2\right)\left(\frac{3L^2\gamma_2}{1-\rho} + 2L^2(1-\beta)^2\gamma_3\right) \\
&\leq \frac{16\sqrt{2}\kappa^2(L+1)}{5\sqrt{\beta}} + \frac{8\sqrt{2}\kappa L}{5(1-\rho)^2} + (15\kappa^2+\kappa+1)\left(\frac{L}{(1-\rho)^2\kappa} + \frac{L+1}{\sqrt{\beta}}\right) \leq \frac{(L+1)(20\kappa^2+\kappa+1)}{\sqrt{\beta}} + \frac{L(18\kappa+2)}{(1-\rho)^2}.
\end{aligned}
$$

By the above equation and the choice of $\eta_{\mathbf{\Lambda}}$, we have

$$\frac{1}{\eta_{\mathbf{\Lambda}}} - \frac{L_P}{2} - \frac{32\kappa^2 L}{\sqrt{\beta}} - \frac{16\sqrt{2}\kappa^3 m\gamma_4}{\sqrt{\beta}} - \frac{m}{2}\left(\frac{\sqrt{\beta}}{\sqrt{2}}\sqrt{4\kappa^2+1} + 30\kappa^2\right)\left(\frac{3L^2\gamma_2}{1-\rho} + 2L^2(1-\beta)^2\gamma_3\right) \geq \frac{1}{2\eta_{\mathbf{\Lambda}}}. \tag{139}$$

On the other hand, from (138) and the definition of $\gamma_1$, it follows

$$
\begin{aligned}
&\frac{1}{2m}\left(2m(1-\rho)\gamma_1 - L(\kappa+1) - \frac{\sqrt{\beta}}{32} - \frac{\rho^2}{\eta_{\mathbf{x}}} - 16mc_{\mathbf{x}}\right) \\
&\geq \frac{1}{2m}\left(2m(1-\rho)\gamma_1 - \frac{m(1-\rho)\gamma_1}{2} - \frac{2}{\eta_{\mathbf{x}}}\right) \geq \frac{1}{6m\eta_{\mathbf{x}}}.
\end{aligned} \tag{140}
$$

Additionally, by the definition of $\gamma_2$ and $\beta < 1$, (61) holds, and thus

$$\frac{(1-\rho)\gamma_2}{2} \geq \frac{\gamma_1\eta_{\mathbf{x}}^2}{1-\rho} \geq \frac{\eta_{\mathbf{x}}}{m(1-\rho)^2}. \tag{141}$$

Also, by the choice of $\gamma_3$ and $\gamma_4$ and $(1-\beta)^2 \leq 1$, it holds that

$$
\begin{aligned}
&\gamma_4 - \frac{\beta}{60\kappa^2}\left(\frac{3L^2\gamma_2}{1-\rho} + 2L^2(1-\beta)^2\gamma_3\right) - \gamma_4\left(1 - \frac{\sqrt{\beta}}{4\sqrt{2}\kappa}\right) - \frac{\sqrt{\beta}(L+1)}{64m\kappa^2} \\
&\geq \frac{\sqrt{\beta}}{4\sqrt{2}\kappa}\gamma_4 - \frac{2\beta L^2}{60\kappa^2}\frac{2}{\beta}\left(\frac{\sqrt{\beta}}{60mL_P} + \frac{\sqrt{\beta}\gamma_4}{\sqrt{2}L\mu} + \frac{3\gamma_2\beta(\beta+1/8)}{1-\rho}\right) - \frac{3\beta L^2\gamma_2}{60\kappa^2(1-\rho)} - \frac{\sqrt{\beta}(L+1)}{64m\kappa^2} \\
&= \frac{11\sqrt{\beta}}{60\sqrt{2}\kappa}\left(\frac{\sqrt{2}(L+1)}{8m\kappa} + \frac{4L^2\sqrt{\beta}}{15\sqrt{2}\kappa}\left(\frac{9\gamma_2}{2(1-\rho)} + \frac{12\gamma_2\beta}{1-\rho} + \frac{1}{15\sqrt{\beta}mL_P}\right)\right) \\
&\quad - \frac{2\beta L^2}{60\kappa^2}\frac{2}{\beta}\left(\frac{\sqrt{\beta}}{60mL_P} + \frac{3\gamma_2\beta(\beta+1/8)}{1-\rho}\right) - \frac{3\beta L^2\gamma_2}{60\kappa^2(1-\rho)} - \frac{\sqrt{\beta}(L+1)}{64m\kappa^2} \\
&\geq \frac{\sqrt{\beta}(L+1)}{160m\kappa^2}.
\end{aligned} \tag{142}
$$

Furthermore, from $1 - (1-\beta)^2 \geq \beta$ and the formula of $\gamma_3$, it holds that

$$
\begin{aligned}
&(1 - (1-\beta)^2)\gamma_3 - \frac{3\beta^2\gamma_2}{1-\rho} - \frac{\sqrt{\beta}}{60mL_P} - \frac{\sqrt{\beta}\gamma_4}{\sqrt{2}L\mu} - \frac{\beta}{8L^2}\left(\frac{3L^2\gamma_2}{1-\rho} + 2L^2(1-\beta)^2\gamma_3\right)\\
\geq& \frac{\sqrt{\beta}}{60mL_P} + \frac{\sqrt{\beta}\gamma_4}{\sqrt{2}L\mu} + \frac{6\gamma_2\beta(\beta+1/8)}{1-\rho} - \frac{\beta\gamma_3}{4} - \frac{3\gamma_2\beta(\beta+1/8)}{1-\rho}\\
=& \frac{\beta\gamma_3}{4} \geq \frac{\sqrt{\beta}\gamma_4}{2\sqrt{2}L\mu} \geq \frac{\sqrt{\beta}(L+1)}{16m\kappa L\mu} \geq \frac{\sqrt{\beta}}{16mL}.
\end{aligned}
\tag{143}
$$

Applying (139)-(143) to (133), summing over $t = 0$ to $T-1$, and finally using $\hat{\delta}_T \geq 0$ and the upper bounds on $\gamma_2, \gamma_3, \gamma_4$ completes the proof. $\qquad\square$

