# OpenReview forum: "Variance-reduced accelerated methods for decentralized stochastic double-regularized nonconvex strongly-concave minimax problems"
_TMLR — Accepted by TMLR_

### Review · Reviewer_PYRf · 2026-03-20

**Summary Of Contributions:**

The paper studies decentralized stochastic nonconvex–strongly-concave minimax problems with convex nonsmooth regularizers on both primal and dual variables, and proposes a single-loop variance-reduced method based on the Lagrangian reformulation of Xu (2024). The problem class is broader than several earlier decentralized minimax works, and the paper gives both SPIDER- and STORM-type variants together with convergence guarantees. That said, I am not fully convinced by the paper’s technical novelty or theoretical significance in its current form.

**Audience:**

Yes

**Audience Explanation:**

The problem class is broader than several earlier decentralized minimax works, and the paper gives both SPIDER- and STORM-type variants together with convergence guarantees.

**Claims And Evidence:**

No

**Claims Explanation:**

- A main selling point of the paper is the inclusion of convex nonsmooth regularizers on both x and y. This certainly enlarges the problem class. However, from the analysis side, the extension does not appear to introduce substantial new difficulty. In particular, in places such as Lemma B.1, it seems that the effect of the regularizers is handled mainly through the nonexpansiveness of the proximal operator.
- The paper claims to address open questions left by Xu (2024), especially by incorporating stochastic gradients and variance reduction. But it is not clear to me that this extension is technically very challenging. In stochastic minimization, moving from full gradients to SPIDER/STORM-type estimators is by now quite standard, and the paper does not clearly explain what makes the minimax case significantly harder here.
- The paper emphasizes that, unlike DREAM, it only uses one communication round per iteration. But if multiple rounds of communication can significantly reduce the overall communication complexity, as also acknowledged in the paper, then it is not obvious why single communication should be preferred.
- The sample and communication complexity results do not improve over existing work; they only match the best-known rates in special cases. The paper also does not reach a lower bound. As a result, it is difficult to view the contribution as a real complexity advance. At present, the theory looks more like an extension of known ideas to a broader setting than a fundamentally new result. I think the paper should position itself more modestly and more clearly along these lines.
- Assumption 1(iii) requires an expected Lipschitz condition on the stochastic gradients themselves. This is stronger than simply assuming that each f_i is smooth. I am not convinced this assumption is mild, and the paper does not explain clearly whether it is really necessary. Could the analysis go through under standard smoothness of f_i plus a variance condition? If not, where exactly is Assumption 1(iii) essential?
- The main theorem is presented in a form that is quite difficult to digest. There are too many constants and interacting terms, and the key message is not easy to extract. More importantly, I am confused by the dependence on $(1-\rho)^2$ in Corollary 4.1. From the way the initialization condition on Y enters (18) and (19), it seems that this factor may end up in the numerator rather than the denominator. This may be my misunderstanding, but I think the derivation should be checked carefully. Since this corollary is one of the main takeaways of the paper, any such inconsistency would be important.

**Requested Changes:**

- The experiments only use a small number of agents. Given that the paper is about decentralized methods and communication efficiency, the evaluation should be carried out at a larger scale. For datasets such as MNIST and CIFAR-10, using 32 or 64 workers should be feasible on 8 V100 GPUs. This would make the empirical claims more convincing. It would also be useful to test more graph topologies or different numbers of agents, rather than keeping the setup fixed.
- Please use “big batch” and “large batch” consistently
- There is an extra “=” near the end of p7
- $\hat{\delta}_0$ and $\mathbf{V}_{\bot}$ are used in Theorem 4.1 before being clearly defined

---

> ### Author Response · Authors · 2026-04-23
> **Response to Comments from Reviewer PYRf**
>
> We thank the reviewer for comments and suggestions. They are addressed below one by one.
> ## Comment 1
> Indeed, the inclusion of convex nonsmooth regularizers on both $x$ and $y$ makes the problem class larger than what considered in literature. To handle the regularizers, we do use nonexpansiveness of their proximal mapping. But we emphasize that these terms are carried over throughout the analysis. Though we do not have surprising techniques to address the challenges, it was unknown how to address them before our work. We believe and hope the reviewer can agree that in many cases, once a technique is presented to solve a hard problem, people may find it is not so challenging. In particular, we had to work on the reformulation (7) to design our algorithm and conduct our analysis instead of the original one in (3).
> ## Comment 2
> Though we work on the reformulation (7) that was introduced by (Xu, 2024), our algorithm is significantly different. (Xu, 2024) applies inner maximization update to dual variable. This is prohibited for the stochastic case. To address this issue, we use a single variance-reduction stochastic gradient update to both primal and dual variables. We agree that SPIDER and STORM are now standard techniques in accelerating stochastic methods. Utilizing them in solving stochastic Minimax Problems with double nonsmooth regularizers in a decentralized setting is highly non-trivial. The main reason is the interplay of several different factors including stochasticity, consensus error, nonconvexity, and nonsmoothness. We will highlight these in the revision.
> ## Comment 3
> Indeed multi-communication can improve the dependence on $\rho$. Our method can also benefit from using multi-communication technique as we explained in Remark 4.1. However, our complexity results still follow even if we do not perform multiple communications per update. This is the key difference from DREAM and is what we emphasized. Practically, we observed that doing one communication per update gives best performance. Hence, our theory gives support for the practically best implementation.
> ## Comment 4
> We emphasize that our complexity results are **optimal** in terms of the dependence on $\epsilon$ and they match the existing best results that are also optimal in $\epsilon$. Hence, the key difference lies in the broader applicability of our methods that can handle double-regularized problems. For dependence on $\rho$, our results can be made optimal by employing multi-communication techniques like DREAM, as we explained in Remark 4.1. Without using such a trick, our method cannot achieve the optimal dependence on $\rho$ unless $\kappa$ is big enough but we do not notice any other method can.
> ## Comment 5
> We acknowledge that the mean-squared smoothness in Assumption 1(iii) is stronger than smoothness of each $f_i$. However, this condition is necessary to achieve the optimal order of $\epsilon^{-3}$ (please see [1] - around equation (4) - and the references therein). We leverage this Assumption in the proof of Lemmas B.7 and B.9. We will make this more clear in our paper.
>
> [1] Arjevani, Yossi, et al. "Lower bounds for non-convex stochastic optimization." Math Programming, 2023.
>
> ## Comment 6
> We keep all constants in main theorems for rigor but simplify the complexity in Corollaries 4.1 and 4.2 for ease of understanding. We checked our proofs. Those are correct. We use the initialization condition on $V$ and $Y$ to simplify $C_0$, and $(1-\rho)^2$ appears in the denominator because of the choice of $\eta_x$ and $\eta_\Lambda$, which have $(1-\rho)^2$ in the numerator of their formula and there are terms $\frac{20}{\eta_x}$ in (18) and $\frac{4}{\eta_\Lambda}$ in (19).
>
> ## Change 1
> Unfortunately, the original machine with 8 GPUs cannot be used any more. We ran more tests on a machine with 10 CPU cores and 2 Quadro RTX 5000 GPUs. For DRO problem, we ran tests with heterogeneous data of MNIST and FashionMNIST (partitioned by label) on 10 agents by CPU, and tests with homogeneous data (randomly partitioned) on 20 agents by CPU. For Fair Classification, we tried to run tests with heterogeneous data of Cifar10 on training AllCNN network by using 2 GPUs as 5 agents. However, it would take more than a month to have one trial over four methods. Hence, we chose not to include new results on Cifar. The new figures with descriptions are included at https://anonymous.4open.science/r/NewPlots-7FE6/NewPlots.md
>
> ## Change 2
> We thank the reviewer for catching this inconsistency. We will change all occurrences to "large batch."
> ## Change 3
> Is the reviewer referring to VR-tag==SPIDER? If so, this is on purpose as lines 4 and 6 in Algorithm 1 use this notation as well.
> ## Change 4
> We thank the reviewer for pointing this out. Our analysis requires the careful construction of bounds related to the gradient estimator errors in order to guarantee convergence. We will move definitions of necessary terms in our Theorems to the main body of the paper.

---

> > ### Comment · Reviewer_PYRf · 2026-04-28
> >
> > Thank you for the authors’ response and the additional experiments. Some of my concerns have been addressed. I will provide my final recommendation accordingly.

---

### Review · Reviewer_PpxE · 2026-04-01

**Summary Of Contributions:**

## Summary

This paper studies distributed minimax optimization for smooth, nonconvex strongly concave objective functions with two nonsmooth regularizers. It integrates two widely used variance reduction (VR) techniques, i.e., SPIDER and STOREM, within a two-timescale update scheme, and establishes sample complexity results that are optimal with respect to the target accuracy $\epsilon$.

## Strengths
* The sample complexities reach the optimal with respect to $\epsilon$.
* The algorithm can handle the structured minimax problem with two regularizers.

## Weaknesses
* The dependence of total sample complexities (of two VR options) with respect to the graph-related parameter ($1-\rho$) is unclear.
* The communication complexity in Corollary 4.1 is worse than previous results with respect to the graph-related parameter.
* The dependence of communication and sample complexities on the graph-related parameters in Corollary 4.2 is unclear.
* The comparison with other algorithms for solving distributed minimax optimization is currently not straightforward. The authors may consider adding a table to better highlight the position of this paper in the literature.

**Additional Comments:**

* Could the authors clarify why different metrics are used in the theoretical analysis and in the experiments (Eq. 24, 26)?
* Could the authors further discuss why the stationarity metric in Eq. (11) excludes both the proximal gradient mapping and the consensus error term for $y$?

**Audience:**

Yes

**Audience Explanation:**

The structured problem minimax problem considered in this paper covers a wide range of applications.

**Broader Impact Concerns:**

None.

**Claims And Evidence:**

Yes

**Claims Explanation:**

The authors support their results by rigorous proofs and numerical experiments.

**Requested Changes:**

* (Critical) The paper would be better positioned within the distributed minimax literature if the authors can include a comparative table for distributed minimax methods with stochastic gradients. Suggested columns include problem setting (structured vs. unstructured), use of variance reduction, sample complexity, and communication complexity.

* (Critical) The treatment of data heterogeneity in the experiments should be clarified. If heterogeneity is considered, the paper should specify how it is quantified and at what levels. If it is not considered, the authors should add experiments under heterogeneous data settings.

* (Critical) Could the authors clarify the practical overhead introduced by the two VR techniques in distributed minimax optimization, including additional storage costs and extra hyperparameter tuning requirements?

* (Maybe a typo) Eq. (7) includes $g_i$ but Eq. (1) does not have $g_i$.

---

> ### Author Response · Authors · 2026-04-23
> **Response to Weaknesses and Requested Changes made by Reviewer PpxE**
>
> We thank the reviewer for comments and questions. They are addressed below one by one.
> ## Weaknesses 1-2
> In Corollary 4.1, we explicitly show the dependence of communication complexity $T_c$ on $1-\rho$ and the sample complexity $T_s$ on $1-\rho$ as well. In Remark 4.1, we discuss the scenario when $\kappa(1-\rho)^2 \ge1$ and our complexity will be independent of $1-\rho$. Otherwise, we can perform multiple communication at the initial step to make $\hat{\delta}_0 \le \kappa(1-\rho)^2$, where $\hat{\delta}_t$ is defined in (43); this way, the dependence will be $\kappa(1-\rho)^{-2}$ in communication complexity $T_c$ and the same in sample complexity $T_s$ for the finite-sum case and $\kappa(1-\rho)^{-4}$ in $T_s$ for the general stochastic case. Moreover, as our analysis only requires $\rho \in [0,1)$, Chebyshev acceleration can be applied for performing multiple communications per update, this way, the dependence on $\rho$ can be reduced to $(1-\rho)^{-\frac{1}{2}}$ in communication complexity and the sample complexity will be independent of $\rho$. These are explained in Remark 4.1.
> ## Weakness 3
> In Corollary 4.2, we assume $\epsilon \le \sigma (1-\rho)^2$. In this case, our communication and sample complexity are independent of $\rho$. This has been explicitly stated in Remark 4.2.
> ## Weakness 4 and Change 1
> Thanks for the suggestion. We plan to add the following table. Here, $\iota_Y$ is the indicator function on $Y$; we do not show the dependence of $\kappa$ and $\rho$ in the first two methods because it is not given in their papers. For VRLM-SPIDER, we assume $\kappa(1-\rho)^2 = \Omega(1)$; for VRLM-STORM, we assume $\epsilon\le (1-\rho)^2$.
> | Method | Structure | $(g, h)$ | single comm. | Samp. Comp. | Comm. Comp. |
> |--------|-------------|----------|------|-------------|-------------|
> | GT-DA (Tsaknakis et al. 2020) | Finite-sum | $(0, \iota_Y)$ | yes | $\tilde{O}\left(\frac{n}{\epsilon^{2}}\right)$ | $\tilde{O}\left(\frac{1}{\epsilon^{2}}\right)$ |
> | GT-SRVR (Liu et al. 2023) | Finite-sum | $(0, 0)$ | yes | $O\left(n+\frac{\sqrt{n}}{\epsilon^{2}}\right)$ | $O\left(\frac{1}{\epsilon^{2}}\right)$ |
> | DREAM (Chen et al. 2022) | Finite-sum | $(0, \iota_Y)$ | no | $O\left(n+\frac{\sqrt{n}\kappa^2}{\epsilon^{2}}\right)$ | $O\left(\frac{\kappa^2}{\sqrt{1-\rho}\epsilon^{2}}\right)$ |
> | DREAM (Chen et al. 2022) | Stoch. | $(0, \iota_Y)$ | no | $O\left(\frac{\kappa^3}{\epsilon^{3}}\right)$ | $O\left(\frac{\kappa^2}{\sqrt{1-\rho}\epsilon^{2}}\right)$ |
> | VRLM-STORM | Stoch. | (cvx, cvx) | yes | $O\left(\frac{\kappa^3}{\epsilon^{3}}\right)$ | $O\left(\frac{\kappa^3}{\epsilon^{3}}\right)$ |
> | VRLM-SPIDER | Finite-sum | (cvx, cvx) | yes | $O\left(n+\frac{\sqrt{n}\kappa^2}{\varepsilon^{2}}\right)$ | $O\left(\frac{\kappa^2}{\epsilon^{2}}\right)$ |
> | VRLM-SPIDER | Stoch. | (cvx, cvx) | yes | $O\left(\frac{\kappa^3}{\epsilon^{3}}\right)$ | $O\left(\frac{\kappa^2}{\epsilon^{2}}\right)$ |
>
> ## Change 2
> In our experiments, we partitioned the data uniformly at random, so the data are homogeneous across multiple agents. However, our theorems do not require data homogeneity. We conducted more experiments on heterogeneous data case, where data are partitioned based on label. The new results with descriptions are included at https://anonymous.4open.science/r/NewPlots-7FE6/NewPlots.md
> ## Change 3
> In the tests, we did grid search for the learning rate, the same as all other methods. For SPIDER, we tried two different $q$, the same as DM-HSGD and GT-SRVR. So compared to existing methods, we do not need extra overhead. In storage costs, our methods need to save two copies of models for variance reduction. We will further clarify these in the paper.
> ## Change 4
> We thank the reviewer for catching this; we will this typo.
>
> ## Additional Comment 1
>
> We use two different metrics for the numerical experiments for the following reasons:
>
> 1. Equation (24) uses the primal function of the original problem (3) evaluated at the average $x$. We use this condition because other methods do not solve the reformulated problem (7) and hence they do not have $\Lambda$-variables.
>
> 2. Equation (26) is slightly different. First, we do not have a regularizer in the primal variable (i.e. $g\equiv 0$) hence we do not need the proximal mapping. Second, as stated at the top of page 12 we are memory constrained in our compute environment so we cannot put all data on one machine to solve the global $\mathbf{y}$-maximization problem.
>
> ## Additional Comment 2
>
> Notice that $\nabla_\Lambda P(\bar{x},\Lambda)=-\frac{L}{2m}(W-I)Y$, hence if $\nabla_\Lambda P(\bar{x},\Lambda)=0$ we have $\frac{L}{2m}(W-I)Y=0$ which by Assumption 2 means $Y$ has reached consensus. Additionally, the maximization of the $Y$-variable is contained in the primal function; thus as we state in Remark 2.1, our notion is called optimization stationarity. We would be happy to further clarify these points in the revision.

---

> > ### Comment · Reviewer_PpxE · 2026-04-28
> >
> > I appreciate the authors' response. Most of my concerns have been addressed. My main remaining concern is the assumption $\kappa(1-\rho)^2 = \Omega(1)$. Since $\kappa$ reflects properties of the function class while $(1-\rho)$ captures the connectivity of the communication graph, these quantities arise from different aspects of the problem. Therefore, I encourage the authors to distinguish between them explicitly, rather than imposing a joint condition on both.

---

> > > ### Author Response · Authors · 2026-04-28
> > > **Response to more suggestions**
> > >
> > > We thank the reviewer for more suggestions. We agree that $\kappa$ and $1-\rho$ come from different aspects of the problem. We will distinguish them more explicitly in the revision by discussing the cases of $\kappa(1-\rho)^2 = \Omega(1)$ and $\kappa(1-\rho)^2 = O(1)$ separately.

---

### Review · Reviewer_jkab · 2026-04-01

**Summary Of Contributions:**

**Summary**

This paper studies decentralized nonconvex-strongly concave minimax problem with possible nonsmooth convex regularizer on primal and dual variables. To motivate the algorithm, the author reformulated the original decentralized minimax problem to a new one with a one-sided Lagrangian form for the dual variable. Building on this new formulation, dual variable can be updated using a primal-dual-type scheme to accommodate the consensus constraint while the primal variable is updated using standard gradient tracking technique. Furthermore, to enhance the convergence rate, the author further incorporate the acceleration schemes, such as momentum and variance reduction schemes SPIDER and STORM. This paper is well-written with solid theoretical results. As such, the contribution is good. Specifically, I read the finite-sum part of the main proof and find it plausible. Finally, I have a few comments that might help improve the clarity of the paper.

**Strength:**
1. The paper studies an interesting problem of decentralized minimax optimization problem with possibly nonconvex regularizer on primal dual variable, making the their methodology applicable to the constraint settings while prior works merely focus on unconstraint or one-sided constraint. A novel one-sided primal-dual reformulation on dual variable is proposed without being studied by prior works and the resulting algorithms are novel.

2. The theoretical results are rigorous, e.g., some of the theoretical results, e.g., $\mathcal{O}(\epsilon^{-3})$, matches the best-known rate in terms of accuracy tolerance factor $\epsilon$.

3. The proposed algorithms allow practitioners to switch between STORM and SPIDER strategies, giving great flexibility to gain advantage in either finite-sum or general stochastic scenarios.

4. The simulations are conducted under a real distributed environment using multiple-GPUs, which validates the performance and strengthens the applicability the proposed methods.

Weakness:

1. As the proof is dense, it is hard for reviewer to track all of them without a proof sketch. Also, more detailed discussion of the consequence on each Lemma should be added.


Other suggestions/questions:

1. In my humble opinion, gradient tracking is just a primal form of some bias-correction (primal-dual) approach, if the goal is to remove the gradient heterogeneity across users, then doing double gradient tracking on both variables should be simple enough? Noting that the primal-dual implementation on dual variable makes the algorithm harder to deploy in practice. Also, it is known that gradient tracking does not achieve the best performance over a sparsely connected network, i..e, when $\rho \rightarrow 1$. Other bias-correction (primal-dual) approach, e.g., EXTRA and exact diffusion, are known to be superior than gradient tracking, perhaps some discussions on these methods would be helpful for future extensions?

2. In my humble opinion, one of the benefit of the decentralized methods is achieving a linear speed-up on the number of users/clients $m$. Adding some discussion to explain what is the cause of the lose of such a linear speed-up would be helpful.

3. Could the author explains why a strong-concavity condition on local cost is needed? Distributed optimization usually needs condition on a global cost rather than local one as the local one, to the best of our knowledge, is stronger. More discussion on such an analytical need would be useful.

4. In remark 2.1, the author pointed out the regularizer $h(\cdot)$ needs to be smooth to show $X$ is optimization stationary point. But the author mentioned the indicator function which is not smooth? Then, how can the author reconcile the equivalence between the proposed convergence metrics and the optimization stationarity as the nonsmooth indicator function is a common choice?

5. In my humble opinion, more technical details that specify the derived bound to final communication complexity and sample complexity results would be useful; especially, those details that show how the parameters $1-\rho$ is being absorbed into the final results, seeing that many parameter, e.g., $C_0$, has a $1-\rho$ dependency.

Minor issues:

1. In (4),  the definition about $X_{\perp}$ should be $X - 1^\top \bar{x}$, without scaling by $1/m$ ?
2. In (7), users use a common regularizer, then what is $g_i$?
3. In (8), please makes the position of arguments inside $\Phi()$ consistent.
4. In proposition B.1 and B.2, the $L_Q$ and $L_p$ have the same value, should the notation be unified?
5. What is $n_t$ in remark B.1, is it $ floor [t/q]$?
6. In (114), be careful about the sign of $2\eta_y - \eta^2_y L$ before applying the strong concavity property.
7. Listing a table of notations would be useful.

**Audience:**

Yes

**Audience Explanation:**

Decentralized minimax optimization is an popular topic.

**Broader Impact Concerns:**

This paper may advance a subfield of machine learning problems

**Claims And Evidence:**

Yes

**Claims Explanation:**

Yes, the results are convincing.

**Requested Changes:**

See my comments above

---

> ### Author Response · Authors · 2026-04-23
> **Response to Comments and Questions from Reviewer jkab**
>
> We kindly thank the reviewer for the constructive comments. We address them below one by one.
>
> ## Weakness
> A proof sketch is available at the beginning of Appendix B. If the reviewer prefers we move this to the main body of the paper, we are happy to consider to do that. Also, we can provide more exposition after each Lemma that does not currently have one.
> ## Question 1
> We agree that gradient tracking can be applied to both $x$ and $y$ update. However, the reformulation in (7) with introduction of a variable $\Lambda$ incorporated the consensus constraint on $y$ into the objective, thus no need to do gradient tracking on $y$. Without working on the reformulation, one may directly do gradient descent ascent to (3) by gradient tracking to both $x$ and $y$. However, we are not sure if convergence can be shown for this way. We believe a primal-dual update is inevitable due to the problem structure, and our algorithm can be easily implemented by a single stochastic gradient update to both primal and dual variables without a subroutine to maximize the dual problem.
>
> Indeed, there are different ways to handle heterogeneity. We choose gradient tracking due to its simplicity and effectiveness in practice. We will add more discussions on other ways in the revision.
> ## Question 2
> We appreciate the reviewer's insightful question. Indeed, a linear speed up in decentralized optimization is an important benefit. However, except the paper (Chen et al, 2024) that utilizes a multi-communication trick, we were not aware of other work showing a linear speed up for decentralized nonconvex stochastic minimax problem. While preparing the paper, we did notice this drawback but could not figure out how to resolve it. We have acknowledged this drawback in Remark 4.3. It could be because our analysis is not tight or because of the difficulty of the double-regularized problem or because of the limitation of the algorithm.
> ## Question 3
> Indeed, the strong concavity on each $f_i$ is stronger than strong concavity of the global function. We need the assumption because of the reformulation in (7). Notice that in (6), $\Phi$ depends on $Y$ instead of the average of all $\mathbf{y}_i$'s. We essentially need strong concavity of $\Phi$ about $Y$ to ensure the primal functions $P$ and $Q$ defined in (8) to be smooth. The strong-concavity is leveraged in Lemma B.6 (equation (114)) and in the proof of Proposition B.1. We will add an explanation after Assumption 2 in the revision.
> ## Question 4
> To clarify, we do NOT need the smoothness of $h$ to show $X$ is an optimization stationary point. The second half in Remark 2.1, i.e., "In addition, ...", is used to relate the optimality condition of our reformulated problem (7) to that of the original formulation in (3); specifically, we maintain the order of $\epsilon$ error when solving (3) if $h$ is smooth, however, **a smooth $h$ is not required for our complexity results to hold on our reformulated problem (7)**. We will add these clarifications in Remark 2.1.
> ## Question 5
> Thanks for the suggestion. Indeed, a few terms in $C_0$ depend on $1-\rho$. To eliminate those dependence, we can choose appropriate initial points, as we state in Corollary 4.1 and Corollary 4.2. We will spell out the technical details on how to cancel the dependence in the revision.
> ## Minor issues
> 1-2: We thank the reviewer for careful reading and catching the typos. We will fix them.
>
> 3: We will make the arguments consistent.
>
> 4: We will unify $L_Q$ and $L_P$.
>
> 5: In Lemma B.7, we define $n_t$ as the unique integer such that $n_t q \le t < (n_t + 1) q$. So the reviewer is right. We will make the definition more visible in the revision.
>
> 6: Because $\eta_y \le \frac{1}{4L}$, it holds $2\eta_y - \eta_y^2 L > 0$. We will explicitly mention this before applying the strong concavity.
>
> 7: Thanks for the suggestion. We would happily add this to the revision.

---

> ### Comment · Reviewer_jkab · 2026-04-27
> **Feedback**
>
> We sincerely appreciate the author’s substantial effort on the detailed response. We acknowledge the significant challenges introduced by primal-dual reformulation on the $y$-variable, which possibly make it fundamentally harder to achieve clear linear speedup benefits or require the stronger assumption on local costs on the theoretical side. That said, in my humble opinion, the benefit over prior works could be limited. This also raises a question:  if the $x$-variable can be simply handled by gradient tracking, then what is the necessity of the primal-dual formulation on the y-variable? In fact, if both variables adopt the gradient tracking plus proximal mapping, would the resulting algorithm be easier to analyze and facilitate the investigation of the linear speed-up benefit? Finally, we suggest being cautious with the claim of completely removing the dependence on $1-\rho$. To our understanding, decentralized primal-dual algorithms generally do not eliminate this dependence entirely; it may still appear in non-dominant terms. Overall, we sincerely thank the author's effort to address my concerns, for which I would recommend weak acceptance for this paper.

---

> > ### Author Response · Authors · 2026-04-28
> > **Response to follow-up questions**
> >
> > Thanks for the feedback. The primal-dual reformulation for $y$-part is to incorporate the consensus constraint of $y$ into the new objective. This makes the analysis go through. We tend to believe adopting gradient tracking plus proximal update to both $x$ and $y$ variables, **without the reformulation by using $\Lambda$**, will be harder to analyze. This was actually what we tried in the very beginning, and we could not figure out how to have guaranteed convergence without performing multiple communication per update. We agree with the reviewer that the dependence on $\rho$ cannot be eliminated. It appears in non-dominant term. We will make these clearer in the revision.

---

### Decision · Action_Editor_UPQQ · 2026-05-08

**Recommendation:** Accept as is

**Additional Comments:**

Please incorporate the reviewers' suggestions into the camera-ready version. An additional minor suggestion is to provide a high-level overview of the proof strategy to help readers understand it more easily.

**Audience:**

Yes

**Audience Explanation:**

Decentralized minimax optimization has been actively studied in recent years. This paper could benefit the field because it investigates a new framework developed in Xu (2024), whose updating scheme differs from that used in the standard minimax framework.

**Claims And Evidence:**

Yes

**Claims Explanation:**

This paper focuses on decentralized minimax optimization. It developed variance-reduced decentralized minimax optimization algorithms based on STORM and SPIDER under both stochastic and finite-sum settings. By leveraging variance-reduced gradients, the proposed methods improve the sample and communication complexities of the most relevant work, Xu (2024). These complexities also match those of existing state-of-the-art algorithms under the standard minimax framework. Overall, the convergence analysis is rigorous, and the theoretical contributions are solid. The experimental results on two applications: distributionally robust optimization and fair classification, confirm the superiority of the proposed algorithms.